# Stable and oxidative charged Ru enhance the acidic oxygen evolution reaction activity in two-dimensional ruthenium-iridium oxide

Wenxiang Zhu[1,4], Xiangcong Song[1,4], Fan Liao [1], Hui Huang [1], Qi Shao [2], Kun Feng[1], Yunjie Zhou[1], Mengjie Ma[1], Jie Wu[1], Hao Yang [1], Haiwei Yang[1], Meng Wang[1], Jie Shi[1], Jun Zhong[1], Tao Cheng [1]✉, Mingwang Shao [1]✉, Yang Liu [1]✉ & Zhenhui Kang [1,3]✉

The oxygen evolution reactions in acid play an important role in multiple energy storage devices. The practical promising Ru-Ir based catalysts need both the stable high oxidation state of the Ru centers and the high stability of these Ru species. Here, we report stable and oxidative charged Ru in two-dimensional ruthenium-iridium oxide enhances the activity. The $Ru_{0.5}Ir_{0.5}O_2$ catalyst shows high activity in acid with a low overpotential of 151 mV at 10 mA cm$^{-2}$, a high turnover frequency of 6.84 s$^{-1}$ at 1.44 V versus reversible hydrogen electrode and good stability (618.3 h operation). $Ru_{0.5}Ir_{0.5}O_2$ catalysts can form more Ru active sites with high oxidation states at lower applied voltages after Ir incorporation, which is confirmed by the pulse voltage induced current method. Also, The X-ray absorption spectroscopy data shows that the Ru-O-Ir local structure in two-dimensional $Ru_{0.5}Ir_{0.5}O_2$ solid solution improved the stability of these Ru centers.

Anodic oxygen evolution reaction (OER) is the bottleneck in the hydrogen production process of water electrolysis[1–4]. Currently, only Ir-based oxides[5–8], Ru-based oxides[9–12] and their derivatives have sufficient corrosion resistance to withstand the harsh acid corrosion and oxidation environments of OER[13]. The scarcity and relatively low OER activity of Ir are insufficient to meet industrial requirements[13,14], while, Ru-based catalysts generally suffer from poor stability on account of the formation of soluble Ru oxides (such as $RuO_4$) during OER process[15,16]. Up to date, the OER catalysts of RuIr bimetallic oxides have been extended from their component-dependence[17–21], optimization of bimetallic oxide nanostructures (one-dimensional[22], three-dimensional[23] and core-shell structures[24], etc.) to modification of electronic properties[25–27]. Notably, the redox of Ru in RuIr bimetallic oxides could be affected by Ir species in RuIr bimetallic oxides system[13,19], in which Ru exhibits a strong oxidation state[19–21,25,26,28]. Two

sides of requirements must be met by a prospective RuIr oxide-based OER catalyst. One is the stable high oxidation state of the Ru active center, and the other one is to prevent the dissolution and inactivation of Ru species by excessive oxidation. Designing a RuIr oxide based OER catalyst that can be qualified for both requirements is quite challenging.

Here, a two-dimensional substitutional solid solution material, phase ruthenium-iridium oxide was successfully synthesized via a two-step molten-alkali process. We show that the stable and oxidative charged Ru two-dimensional RuIr oxides enhance the OER activity significantly. When the optimal $Ru_{0.5}Ir_{0.5}O_2$ used as an OER catalyst, it shows excellent OER performance in acidic media, providing an anodic current density of 10 mA cm$^{-2}$ at an overpotential of only 151 mV, and together with an activity retention time over a 618.3 h stability test at 10 mA cm$^{-2}$. In addition, $Ru_{0.5}Ir_{0.5}O_2$ also achieves a high mass activity

[1]Institute of Functional Nano & Soft Materials (FUNSOM), Jiangsu Key Laboratory for Carbon-Based Functional Materials & Devices, Soochow University, 199 Ren'ai Road, Suzhou 215123 Jiangsu, China. [2]College of Chemistry, Chemical Engineering and Materials Science, Soochow University, Jiangsu 215123, China. [3]Macao Institute of Materials Science and Engineering (MIMSE), MUST-SUDA Joint Research Center for Advanced Functional Materials, Macau University of Science and Technology, Taipa 999078 Macao, China. [4]These authors contributed equally: Wenxiang Zhu, Xiangcong Song. ✉e-mail: tcheng@suda.edu.cn; mwshao@suda.edu.cn; yangl@suda.edu.cn; zhkang@suda.edu.cn

of 730.4 A $g_{Ir + Ru}^{-1}$ at 1.44 V vs. RHE and a high turnover frequency (TOF) of 6.84 $s^{-1}$ at 1.44 V vs. RHE. The results of pulse voltage induced current (PVC), cyclic voltammetry (CV), density functional theory studies and transient light-induced voltage (TPV) tests showed that the origin of high activity in the $Ru_{0.5}Ir_{0.5}O_2$ catalyst is more Ru active sites with high oxidation states at low applied voltage were formed after Ir incorporation, while increasing the oxidative charge concentration on the surface of the catalyst during the OER process. The X-ray absorption spectroscopy (XAS) measurements of $Ru_{0.5}Ir_{0.5}O_2$ show that the stability was originated from the interaction within the local structure of Ru-O-Ir, and the characteristics of two-dimensional material structure.

## Results

### Synthesis strategy and morphological structural characterizations of $Ru_{0.5}Ir_{0.5}O_2$

The synthesis strategy is to uniformly disperse (at the atomic level) the catalytic active components (Ru/Ir atoms) in the catalysts. In $RuO_2$ and $IrO_2$, Ru/Ir atoms have the same valence state, the same crystal structure type and similar chemical properties, which are conducive to synthesis substitutional solid solution. In a typical experiment, $IrCl_3$ (0.5 M) and KOH were mixed and thoroughly ground and the mixture was heated again by a mechano-thermal method[29]. The cooled solid was thoroughly ground again with addition of $RuCl_3$ (0.5 M) and KOH and heated again by a mechano-thermal method to obtain $Ru_{0.5}Ir_{0.5}O_2$ product[29]. Both heating processes are heated to 800 °C for 2 h. One thing should be noted that when further annealing $Ru_{0.5}Ir_{0.5}O_2$ at 900 °C for 2 h, it obtains the mixture of thermally stable rutile phase $IrO_2$ and $RuO_2$ (Supplementary Fig. 1), which demonstrates that the synthesized $Ru_{0.5}Ir_{0.5}O_2$ is a solid solution. Ru species have higher OER activity than Ir species. In order to achieve higher OER activity, more Ru species are highly suggested to incorporate in the substitutional solid solution. Unfortunately, when the molar amount of raw material ($RuCl_3$) in the second synthetic step was further increased to more than 0.5 M (i.e., the molar ratio of Ru: Ir larger than 1), a large amount of soluble Ru/Ir complexes were generated in the obtained product (Supplementary Fig. 2). Therefore, the optimal molar ratio of Ru in two-dimensional RuIr oxides is 0.5.

As shown in the scanning electronic microscopy (SEM) image (Fig. 1a) and transmission electron microscopy (TEM) images (Fig. 1b and Supplementary Fig. 3a), $Ru_{0.5}Ir_{0.5}O_2$ exhibits a typical two-dimensional (2D) sheet-like shape structure with a large diameter of about 3–5 μm. Atomic force microscopy (AFM) image further confirms the two-dimensional structure where the thickness of the $Ru_{0.5}Ir_{0.5}O_2$ nanosheet is approximately 1.9 nm (Supplementary Fig. 3b, c). Brunauer-Emmett-Teller (BET) adsorption-desorption isotherm reveals that the $Ru_{0.5}Ir_{0.5}O_2$ nanosheets exhibit a relatively high surface area (25.8 $m^2$ $g^{-1}$) (Supplementary Fig. 4 and Supplementary Table 1). The TEM energy-dispersive X-ray spectroscopy (TEM-EDX) and inductively coupled plasma atomic emission spectra (ICP-AES) were used to determine the chemical composition ratio of $Ru_{0.5}Ir_{0.5}O_2$. The measured atomic ratio of Ru: Ir: O is about 0.46–0.50: 0.54–0.50: 2 (Supplementary Fig. 3d and Supplementary Table 2). High-angle annular dark-field scanning TEM (HAADF-STEM) image and scanning transmission electron microscopy energy dispersive X-ray spectroscopy (STEM-EDX) element mapping reveal that Ru, Ir and O are uniformly distributed throughout the nanosheet (Supplementary Fig. 3g). At the same time, commercial $IrO_2$ (C-$IrO_2$) and commercial $RuO_2$ (C-$RuO_2$) were characterized by X-ray diffraction (XRD), SEM, TEM, and HAADF-STEM. Both the structure characterizations showed the stable rutile phase C-$IrO_2$ and rutile phase C-$RuO_2$ (Supplementary Figs. 5 and 6).

The powder XRD was used to further confirm the crystal structure and the phase properties of the synthesized $Ru_{0.5}Ir_{0.5}O_2$. The sharp Bragg diffraction peaks in the XRD pattern illustrate the high crystallinity and ordered stacking of the 2D layers in $Ru_{0.5}Ir_{0.5}O_2$ (Fig. 1c). Meanwhile, the highly crystalline structures of $Ru_{0.5}Ir_{0.5}O_2$ were also confirmed by the lattice-resolved high-resolution TEM (HRTEM) image (Supplementary Fig. 3e) and the corresponding fast Fourier transform (FFT) pattern (Supplementary Fig. 3f). Selected area electron diffraction (SAED) pattern obtained at the $Ru_{0.5}Ir_{0.5}O_2$ nanosheets is compatible with the trigonal crystal structure (Fig. 1). The atomic structure of was further clarified by aberration-corrected HAADF-STEM image. As shown in Fig.1d, a highly ordered arrangement of metal atoms can be observed. Figure 1e (the high-magnification image of the region in Fig. 1d) reveals that the ordered arranged metal atoms have different

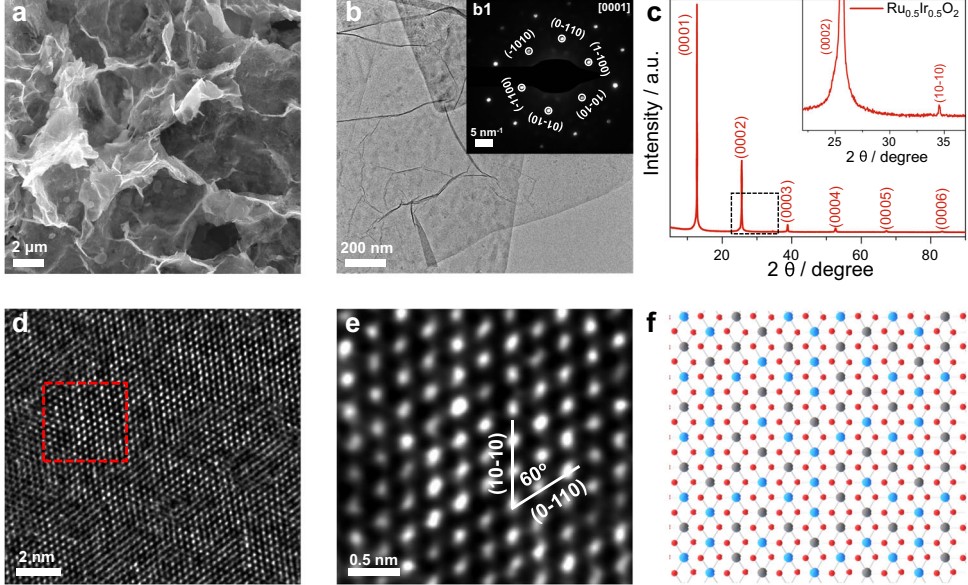

**Fig. 1 | Structural and phase characterizations of $Ru_{0.5}Ir_{0.5}O_2$. a** SEM and **b** TEM images of $Ru_{0.5}Ir_{0.5}O_2$. **b1** The SAED pattern of $Ru_{0.5}Ir_{0.5}O_2$, where the hexagonal pattern shows the [0001] projection. **c** XRD pattern of $Ru_{0.5}Ir_{0.5}O_2$. The inset is a larger view of the marked area in **c. d.** The aberration-corrected HAADF-STEM image for $Ru_{0.5}Ir_{0.5}O_2$. **e** High-magnification image of the region in **d. f** The schematic atom structure of $Ru_{0.5}Ir_{0.5}O_2$, the Ru, Ir, O atoms are represented by blue, gray and red spheres.

brightness, arising from a random and even distribution of Ir and Ru atoms of different brightness, which is also consistent with the substitutional solid solution structure of $Ru_{0.5}Ir_{0.5}O_2$. $Ru_{0.5}Ir_{0.5}O_2$ crystal were modeled as shown in Fig. 1f, where the blue, gray and red spheres schematically represent the arrangement of Ru, Ir and O atoms of $Ru_{0.5}Ir_{0.5}O_2$. Combining with the XRD result (Fig. 1c), the unit-cell parameters of $Ru_{0.5}Ir_{0.5}O_2$ can determined to be $a = b = 3.00$ Å, $c = 6.95$ Å; $\alpha = \beta = 90°$, $\gamma = 120°$.

The composition and chemical states of $Ru_{0.5}Ir_{0.5}O_2$ were then analyzed by X-ray photoelectron spectroscopy (XPS). As depicted in the XPS spectra, $Ru_{0.5}Ir_{0.5}O_2$ exhibits a meaningful negative-shift of Ru $3p_{3/2}$ and Ru $3p_{1/2}$ peaks (461.6 and 483.6 eV) in comparison with those for C-$RuO_2$ (462.0 and 484.0 eV) (Fig. 2a), which confirms the a lower valence state of Ru in $Ru_{0.5}Ir_{0.5}O_2$ than that in C-$RuO_2$ ($Ru^{IV+}$)[18,24]. Ir $4f$ XPS spectra (Fig. 2b) show the peaks located at 62.2 and 65.2 eV, which are assigned to Ir $4f_{7/2}$ and Ir $4f_{5/2}$ of $Ir^{IV+}$ [24]. Compared with those of C-$IrO_2$ (61.8 and 64.8 eV), $Ru_{0.5}Ir_{0.5}O_2$ shows a slight positive-shift. Which indicates that $Ru_{0.5}Ir_{0.5}O_2$ has the higher Ir valence state than

C-$IrO_2$[30]. The fitting parameters used of all peaks can be found in Supplementary Table 3.

The white line of X-ray absorption near-edge structure (XANES) was used to explore the electron transition behavior and electronic structure of $Ru_{0.5}Ir_{0.5}O_2$, and the intensity analysis of which can give the clearer valence electronic state information[31]. As depicted in Fig. 2c, the white-line region of Ru K-edge for $Ru_{0.5}Ir_{0.5}O_2$ shows the white-line adsorption energy of which is between that of C-$RuO_2$ and Ru foil, indicating that the valence state of Ru in $Ru_{0.5}Ir_{0.5}O_2$ is between 0 and +4[32,33]. Meanwhile, by characterizing the intensity of white-line intensity of Ir $L_3$-edge XANES spectra, the Ir valence state can be more easily observed. Figure 2d shows that the white line peak intensity of $Ru_{0.5}Ir_{0.5}O_2$ exhibits significantly higher than that of C-$IrO_2$ and Ir foil, demonstrating that the valence states of Ir show an order of $Ru_{0.5}Ir_{0.5}O_2 >$ C-$IrO_2 >$ Ir foil[34]. The valance state results of Ru and Ir species were consistent with those discussed earlier in the XPS results. The Fourier transforms of the extended X-ray absorption fine structure (EXAFS) spectra (Fig. 2e, f) at the Ru K-edge and Ir $L_3$-edge were

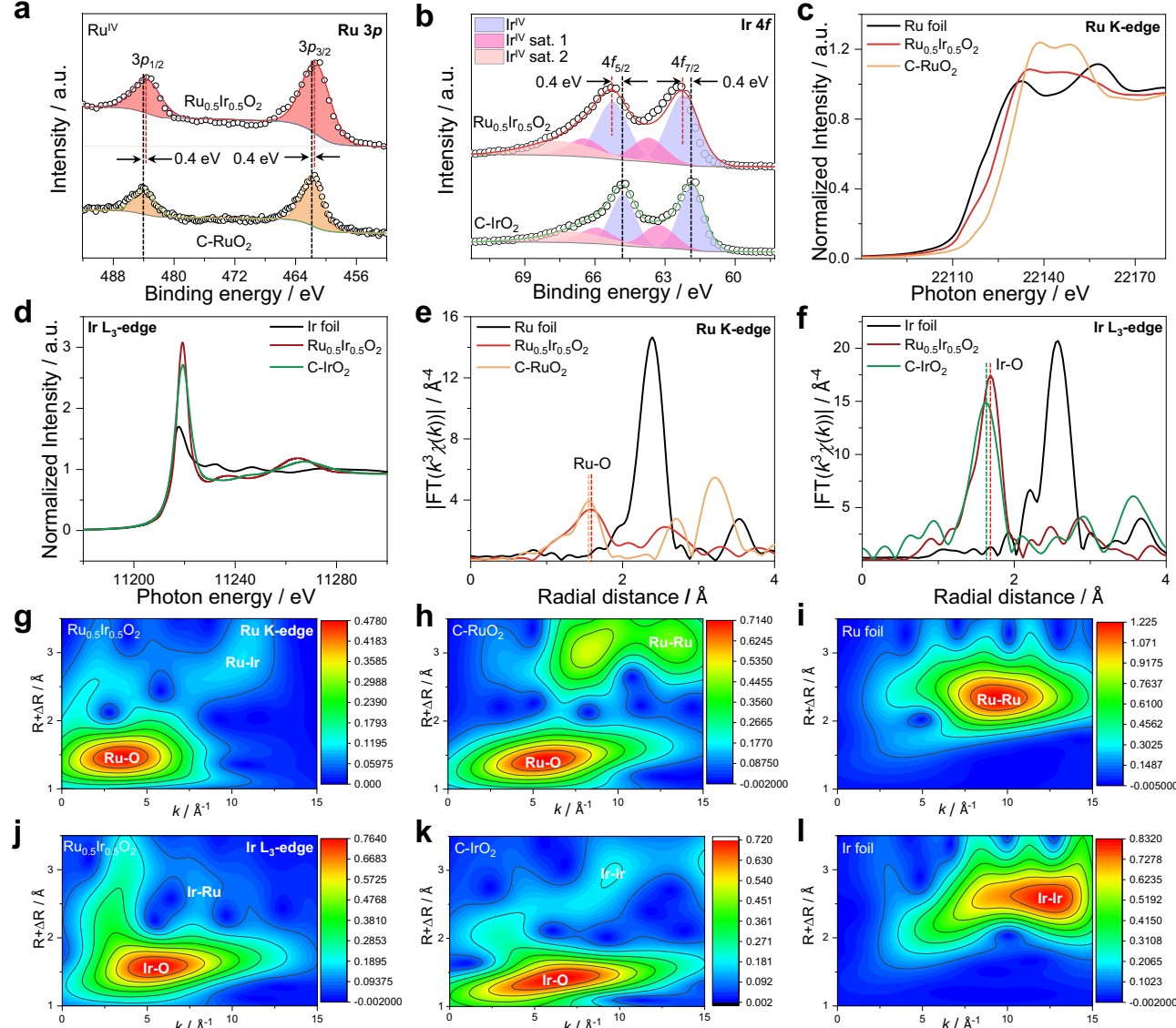

**Fig. 2 | XPS and XAS characterizations of electrocatalysts. a** Ru $3p$ XPS spectra of $Ru_{0.5}Ir_{0.5}O_2$ and C-$RuO_2$. **b** Ir $4f$ XPS spectra of $Ru_{0.5}Ir_{0.5}O_2$ and C-$IrO_2$. **c** Ru K-edge XANES spectra of $Ru_{0.5}Ir_{0.5}O_2$, C-$RuO_2$ and Ru foil. **d** Ir $L_3$-edge XANES spectra for $Ru_{0.5}Ir_{0.5}O_2$, C-$IrO_2$ and Ir foil. **e** Fourier-transformed EXAFS spectra of Ru K-edge

spectra for $Ru_{0.5}Ir_{0.5}O_2$, C-$RuO_2$ and Ru foil. **f** Fourier-transformed EXAFS spectra at Ir $L_3$-edge collected for $Ru_{0.5}Ir_{0.5}O_2$, C-$IrO_2$ and Ir foil. **g**–**i** Ru K-edge WT-EXAFS of $Ru_{0.5}Ir_{0.5}O_2$, C-$RuO_2$ and Ru foil. **j**–**l** Ir $L_3$-edge WT-EXAFS of $Ru_{0.5}Ir_{0.5}O_2$, C-$IrO_2$ and Ir foil.

conducted to investigate the local chemical environment of $Ru_{0.5}Ir_{0.5}O_2$[24]. The FT-EXAFS of Ru K-edge and Ir $L_3$-edge reveal that the bond length of Ru-O in $Ru_{0.5}Ir_{0.5}O_2$ is slightly increased compared to that of C-RuO$_2$, and the length of Ir-O bonds is also increased compared to that of C-IrO$_2$, which may be due to different crystal structure as the metal-oxygen bonds in 1 T phase are larger than those in rutile. (Fig. 2e, f, Supplementary Figs. 7–9 and Supplementary Table 4). This suggests the interaction between Ru, O and Ir[13]. Figure 2e, f shows that the peaks in the R-space at around 1.5 Å and 1.6 Å corresponds to the coordination Ru-O shell and Ir-O shell, indicating that $Ru_{0.5}Ir_{0.5}O_2$ has the typical $RuO_6$ and $IrO_6$ coordination octahedron as shown in Supplementary Table 4. As shown in Supplementary Figs. 8, 9 and Supplementary Table 4, The fitting curves in $Ru_{0.5}Ir_{0.5}O_2$ and reference samples also coincided well with the experimental spectra, which also reveals that there is an interaction between the Ru and Ir. The local structure of Ru-O-Ir was further observed using the wavelet transform of EXAFS (WT-EXAFS)[35]. Compared with the reference samples, Ru-Ir, Ir-Ru scattering signals appeared in Ru K-edge (Fig. 2g–i) and Ir $L_3$-edge (Fig. 2j–l) WT-EXAFS, respectively, which indicated that Ru and Ir had a strong interaction in $Ru_{0.5}Ir_{0.5}O_2$[13]. The presence of strong interactions in these Ru-O-Ir local structures may avoid the formation of more

soluble Ru/Ir high-valent complexes and thus significantly improve the stability of electrocatalysts[13,36]. All structure and characterization information conclude that the $Ru_{0.5}Ir_{0.5}O_2$ belongs to the space group of P-3m1.

## Electrochemical performances

The OER performance of $Ru_{0.5}Ir_{0.5}O_2$ was evaluated in $O_2$-saturated 0.5 M $H_2SO_4$ electrolyte and compared with the advanced C-IrO$_2$ and C-RuO$_2$ electrocatalysts. The saturated calomel electrode (SCE) was used as a reference electrode and was calibrated prior to electrocatalytic testing (Supplementary Fig. 10). The linear sweep voltammetry (LSV) curves of all catalysts are normalized by the geometric area of glassy carbon electrode (GCE) (with the mass loading of 283 μg cm$^{-2}$) with $iR$-correction (Supplementary Figs. 11 and 12). As can be seen from Fig. 3a, Supplementary Fig. 13, $Ru_{0.5}Ir_{0.5}O_2$ exhibits the lowest onset potential ($\eta_{0.3}$) of 1.30 V vs. reversible hydrogen electrode (RHE) at 0.3 mA cm$^{-2}$. And to deliver a current density of 10 mA cm$^{-2}$, $Ru_{0.5}Ir_{0.5}O_2$ only requires a minimum overpotential ($\eta_{10}$) of 151 mV, while the overpotentials ($\eta_{10}$) of C-IrO$_2$ and C-RuO$_2$ were 321 and 297 mV, respectively. Besides, the lowest Tafel slope for $Ru_{0.5}Ir_{0.5}O_2$ (45 mV dec$^{-1}$) indicates that the fastest kinetic velocity compared to

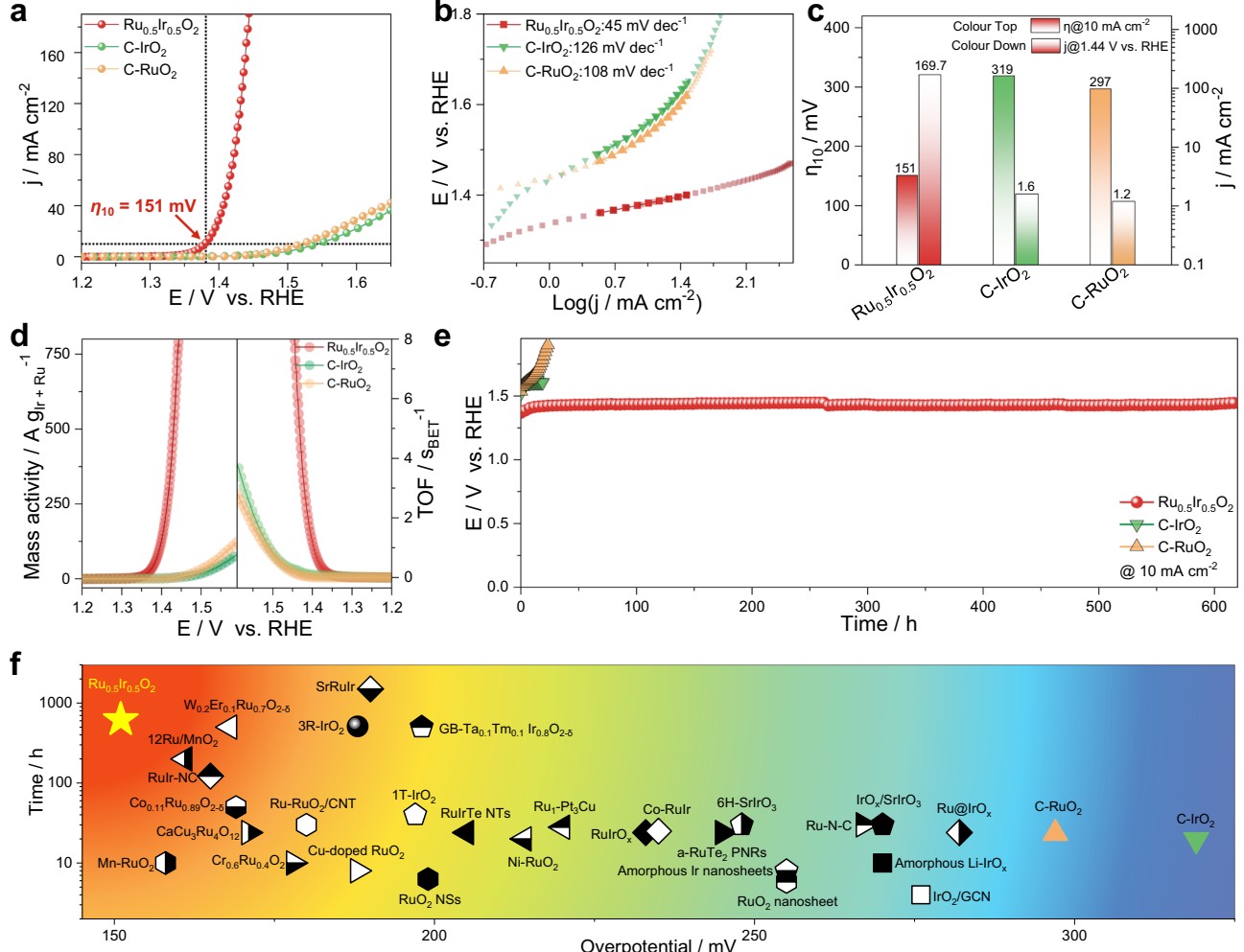

**Fig. 3 | OER performance of $Ru_{0.5}Ir_{0.5}O_2$ electrocatalyst and the reference samples. a** The OER polarization curves of $Ru_{0.5}Ir_{0.5}O_2$, C-IrO$_2$ and C-RuO$_2$ in $O_2$-saturated 0.5 M $H_2SO_4$ electrolyte with $iR$-correction (mass loading ~283 'μg cm$^{-2}$). **b** Tafel plots of $Ru_{0.5}Ir_{0.5}O_2$, C-IrO$_2$ and C-RuO$_2$ collected in 0.5 M $H_2SO_4$ electrolyte were calculated from the corresponding LSV curves (**a**). **c** The comparison of overpotentials at 10 mA cm$^{-2}$ and current densities at 1.44 V vs. RHE for different

catalysts. **d** Mass activities and TOFs of $Ru_{0.5}Ir_{0.5}O_2$, C-IrO$_2$ and C-RuO$_2$. **e** The comparison of chronopotentiometric measurements for different catalysts. **f** The Comparison of the required overpotential at 10 mA cm$^{-2}$ and chronopotentiometry durability in acidic media for various reported electrocatalysts (Supplementary Table 7).

C-IrO$_2$ (126 mV dec$^{-1}$) and C-RuO$_2$ (108 mV dec$^{-1}$) (Fig. 3b). In addition, Supplementary Fig. 14 shows that the OER performance achieved by Ru$_{0.5}$Ir$_{0.5}$O$_2$ has good reproducibility.

Figure 3c shows the comparison of the overpotential ($\eta_{10}$) at 10 mA cm$^{-2}$ and the current density at 1.440 V vs. RHE for different electrocatalysts. It is noteworthy that the current density of Ru$_{0.5}$Ir$_{0.5}$O$_2$ reaches 169.7 mA cm$^{-2}$ at 1.440 V vs. RHE, which is 106.1 times and 141.4 times of C-IrO$_2$ (1.6 mA cm$^{-2}$) and C-RuO$_2$ (1.2 mA cm$^{-2}$), respectively. The gas chromatographic diagram analysis was further used to confirm the formation of O$_2$ gas and evaluate the Faraday efficiency of O$_2$. The data of the produced oxygen by Ru$_{0.5}$Ir$_{0.5}$O$_2$ at 20 mA cm$^{-2}$, 40 mA cm$^{-2}$, 100 mA cm$^{-2}$ in 0.5 M H$_2$SO$_4$ electrolyte were collected (Supplementary Fig. 15). Supplementary Fig. 16 shows that the O$_2$ Faraday efficiencies of Ru$_{0.5}$Ir$_{0.5}$O$_2$ at different current densities are in close proximity to 100%.

In order to compare the intrinsic activity of the prepared electrocatalysts, the mass activity and the turnover frequency (TOF) of the prepared electrocatalysts were calculated based on the total loaded mass of the noble metal (Ir + Ru). Figure 3d manifests the mass activities of Ru$_{0.5}$Ir$_{0.5}$O$_2$ reach 730.4 A g$_{Ir + Ru}$$^{-1}$ at 1.440 V vs. RHE, which are 110.7 and 130.4 times higher than those of C-IrO$_2$ (6.6 A g$_{Ir + Ru}$$^{-1}$ at 1.440 V vs. RHE) and C-RuO$_2$ (5.6 A g$_{Ir + Ru}$$^{-1}$ at 1.440 V vs. RHE). The turnover frequency (TOF) of noble metal site was also calculated. Figure 3d shows that the TOF of Ru$_{0.5}$Ir$_{0.5}$O$_2$ (6.84 s$^{-1}$ at 1.440 V vs. RHE) was nearly 136.8 and 171 times higher than that of C-IrO$_2$ (0.05 s$^{-1}$ at 1.440 V vs. RHE) and C-RuO$_2$ (0.04 s$^{-1}$ at 1.440 V vs. RHE).

To better understand the origin of the high OER performance of Ru$_{0.5}$Ir$_{0.5}$O$_2$, we explored the electrochemical double-layer capacitance (C$_{dl}$) test and the electrochemically active surface area (ECSA) was calculated by C$_{dl}$ for activity normalization (Supplementary Figs. 17, 18 and Supplementary Fig. Table 5). The prepared Ru$_{0.5}$Ir$_{0.5}$O$_2$ with typical two-dimensional (2D) sheet-like shape structure exhibited higher C$_{dl}$ and ECSA than those of C-IrO$_2$ and C-RuO$_2$ (Supplementary Fig. 17 and Supplementary Table 5), suggesting that 2D shape can significantly improve the density of active sites. The OER activity normalized to the ECSA and BET of the catalysts were also calculated (Supplementary Figs. 18 and 19). The specific activity of Ru$_{0.5}$Ir$_{0.5}$O$_2$ was better than those of all the other samples. These results show that Ru$_{0.5}$Ir$_{0.5}$O$_2$ has superior intrinsic OER catalytic activity.

The operational durability is the key to evaluate the application potential of OER catalyst. Thus, the stability of the Ru$_{0.5}$Ir$_{0.5}$O$_2$ electrocatalyst was performed by using chronopotentiometric measurement at a constant current density of 10 mA cm$^{-2}$, showing that the catalyst exhibits a stable overpotential for 618.3 h (Fig. 3e). The overpotential of the Ru$_{0.5}$Ir$_{0.5}$O$_2$ at 10 mA cm$^{-2}$ increased from 151 mV to 204 mV in first 100 h test, to 214 mV at the end of the test (Supplementary Fig. 20). And the mean increase of overpotential was 0.102 mV h$^{-1}$. This is much slower than that of the contrast catalysts. For instance, at a constant current density of 10 mA cm$^{-2}$, C-IrO$_2$ was kept stable for only 19.5 h (2.67 mV h$^{-1}$), and C-RuO$_2$ was inactivated after 23.5 h test (15.83 mV h$^{-1}$). The amount of dissolved Ru and Ir ions in the electrolyte at different times during the stability test of Ru$_{0.5}$Ir$_{0.5}$O$_2$ was determined by ICP-AES analysis (Supplementary Table 6). The concentration of Ru ion is higher than that of Ir ion, which may be attributed to the dissolution of Ru in an acidic medium. Nevertheless, the dissolution of Ru and Ir ions remained at a very low level (less than 30.0 ppb) and varied slightly over time. The chemical state of Ru$_{0.5}$Ir$_{0.5}$O$_2$ after the stability test was also confirmed by XPS analysis. The Ru 3$p_{3/2}$ (462.2 eV) and Ru 3$p_{1/2}$ (484.2 eV) peaks of Ru$_{0.5}$Ir$_{0.5}$O$_2$ after the stability test were positive-shifted compared with before the stability test (Supplementary Fig. 21a). And Ir 4$f_{7/2}$ (61.6 eV) and Ir 4$f_{5/2}$ (64.6 eV) peaks showed that there was is a slight negative-shift compared with before stability tests (Supplementary Fig. 21b). The XPS analysis shows that the surface oxidation state of Ru increased and the surface oxidation state of Ir decreased compared with before

the test during the OER stability test. Furthermore, the XANES analyses also reveal that the valence state of Ru increased slightly and the oxidation state of Ir decreased slightly after the stability test compared with before the test (Supplementary Fig. 22). These may be due to the strong interaction between Ir-Ru and the electron transition behavior that limits the redox elasticity of Ru[13,19–21], avoiding the generation of more soluble Ru high-valent complexes and also the origin of high stability of the Ru$_{0.5}$Ir$_{0.5}$O$_2$[25–27].

The morphology and crystal structure of Ru$_{0.5}$Ir$_{0.5}$O$_2$ after acid OER stability test were further confirmed by morphology characterization. The Bragg diffraction peaks in the XRD pattern of Ru$_{0.5}$Ir$_{0.5}$O$_2$ were basically unchanged, and no other peaks corresponding to rutile C-IrO$_2$ and rutile C-RuO$_2$ were observed (Supplementary Fig. 23a). As shown in Supplementary Fig. 23b, the original two-dimensional sheet structure of Ru$_{0.5}$Ir$_{0.5}$O$_2$ basically remained after stable operation test under harsh conditions. The SAED pattern (Supplementary Fig. 23c), HRTEM image and the corresponding fast Fourier transform (FFT) pattern (Supplementary Fig. 23d) reveal that the crystal structure change of Ru$_{0.5}$Ir$_{0.5}$O$_2$ is negligible. Furthermore, the STEM-EDX element mapping (Supplementary Fig. 23e) shows the uniform distribution of Ir and Ru elements. And the TEM-EDX shows the Ru, Ir and O elements of Ru$_{0.5}$Ir$_{0.5}$O$_2$ after acid OER stability test with an atomic ratio of about 0.50: 0.51: 2.17 (Supplementary Fig. 23f). Due to long-term stability tests, part of the metal cation dissolved in the electrolyte.

The stability of Ru$_{0.5}$Ir$_{0.5}$O$_2$ was further demonstrated by the accelerated durability test-cyclic voltammetry (ADT-CV). Supplementary Fig. 24a showed the cyclic voltammogram (CV) test of Ru$_{0.5}$Ir$_{0.5}$O$_2$ for 1000 cycles with a scan rate of 100 mV s$^{-1}$. Supplementary Fig. 24b showed the OER polarization curves of Ru$_{0.5}$Ir$_{0.5}$O$_2$ before and after 1000 cycles. After the ADT test, the overpotential ($\eta$ @10 mA cm$^{-2}$) of Ru$_{0.5}$Ir$_{0.5}$O$_2$ was increased by 11 mV (Supplementary Fig. 24b) and The Tafel slopes of Ru$_{0.5}$Ir$_{0.5}$O$_2$ before and after ADT are 44 mV dec$^{-1}$ and 49 mV dec$^{-1}$ respectively (Supplementary Fig. 24c), indicating that Ru$_{0.5}$Ir$_{0.5}$O$_2$ has good stability. And the characterization of Ru$_{0.5}$Ir$_{0.5}$O$_2$ after 1000 cycles of ADT-CV showed that the change in crystallinity of Ru$_{0.5}$Ir$_{0.5}$O$_2$ before and after ADT was negligible (Supplementary Fig. 25).

In addition, Ru$_{0.5}$Ir$_{0.5}$O$_2$ (mass loading ~1.0 mg cm$^{-2}$) was used as an anode catalyst in acidic PEM electrolyte (0.5 M H$_2$SO$_4$) at room temperature (Supplementary Fig. 26). The PEM electrolyzers can achieve current densities of ~200 mA cm$^{-2}$ at least 255 h by using Ru$_{0.5}$Ir$_{0.5}$O$_2$ catalyst, and the performance of PEM electrolyzer without significant performance degradation. We compared the OER performance of Ru$_{0.5}$Ir$_{0.5}$O$_2$ with the reported OER electrocatalysts in terms of activity, TOF, mass activity, and stability. As shown in Fig. 3f and Supplementary Table 7, the performance of Ru$_{0.5}$Ir$_{0.5}$O$_2$ is one of the most active electrocatalysts of the state-of-the-art Ru/Ir-based OER electrocatalysts reported in the reported literature.

## Effect of charge on the OER performance

The electrochemical cyclic voltammetry (CV) tests of Ru$_{0.5}$Ir$_{0.5}$O$_2$, C-IrO$_2$, and C-RuO$_2$ were measured in the anhydrous acetonitrile to explore the intrinsic redox capacity of the catalysts. As shown in Supplementary Fig. 27, the prepared catalysts exhibit a series of significant redox peaks in the range of −1.2 to 2.0 V vs. NHE in anhydrous acetonitrile. As depicted in Fig. 4a, compared to C-IrO$_2$ and C-RuO$_2$, Ru$_{0.5}$Ir$_{0.5}$O$_2$ shows an increase in the oxidation state under a low applied voltage. Combined with the electron structure characterization (XPS, XPS simulation, Bader charges and XAS results before and after the long-term OER testing), CV results indicate that Ru species in Ru$_{0.5}$Ir$_{0.5}$O$_2$ material are more easily oxidized than Ir species under the same conditions. It suggests that Ru$_{0.5}$Ir$_{0.5}$O$_2$ is more likely to generate more high valence Ru active sites than C-IrO$_2$ and C-RuO$_2$ at low applied voltage, which may mainly be responsible for the good acidic oxygen evolution reaction (OER) activity over Ru$_{0.5}$Ir$_{0.5}$O$_2$[13,37–39].

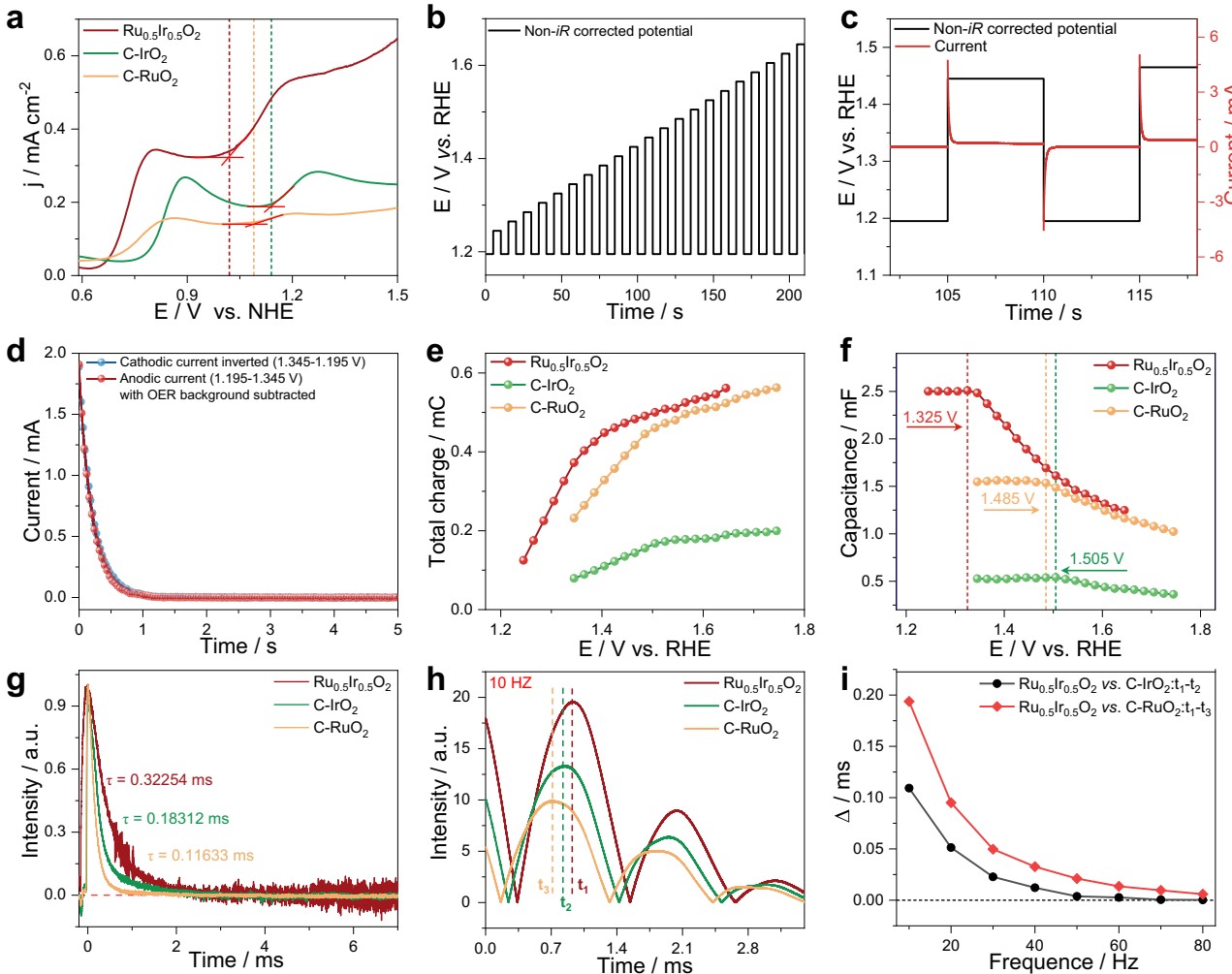

**Fig. 4 | The CV, PVC response and TPV curves of different electrocatalysts. a** CV curves of $Ru_{0.5}Ir_{0.5}O_2$, $C$-$IrO_2$ and $C$-$RuO_2$ in anhydrous acetonitrile. **b** PVC protocol of $Ru_{0.5}Ir_{0.5}O_2$, $C$-$IrO_2$ and $C$-$RuO_2$ between 1.195 V vs. RHE cathodic and 1.245 to 1.645 V vs. RHE anodic non-$iR$ corrected potentials in $O_2$-saturated 0.5 M $H_2SO_4$ electrolyte. **c** PVC protocol (black) and showing oxidation and reduction with current response (red). **d** The anodic and inverted cathodic current decay of $Ru_{0.5}Ir_{0.5}O_2$. **e** Total charge (integral anodic charge) of $Ru_{0.5}Ir_{0.5}O_2$, $C$-$IrO_2$ and $C$-$RuO_2$ versus potential from PVC. **f** Anodic capacitance derived from normalized anodic charge to potential step from the PVC. **g** The TPV curves of $Ru_{0.5}Ir_{0.5}O_2$, $C$-$IrO_2$ and $C$-$RuO_2$. **h** Intensity-Time curves of $Ru_{0.5}Ir_{0.5}O_2$, $C$-$IrO_2$ and $C$-$RuO_2$ at 10 Hz. ($t_1$, $t_2$ and $t_3$ are the peak time of $Ru_{0.5}Ir_{0.5}O_2$, $C$-$IrO_2$ and $C$-$RuO_2$, respectively). **i** Comparison of peak occurrence time of $Ru_{0.5}Ir_{0.5}O_2$ with peak occurrence time of $C$-$IrO_2$ and $C$-$RuO_2$ at different frequencies (10, 20, 30 40, 50, 60, 70 and 80 Hz).

The degree of oxidation of the active site in the catalyst and the concentration of the oxidative charge on the surface directly respond to the OER rate[40,41]. Therefore, in order to determine the instantaneous oxidation state of the Ru active site during the OER process, we used the pulse voltage induced current (PVC) test method to compare potential, charge and quantify the capacitance of the catalysts under different biases. The bias applied in the PVC test preferentially acted on the oxidation process of the metal, and the resulting oxidative charge on the metal surface is then involved in the OER reaction process, and the change in the oxidation state of the metal causes the capacitance of the catalyst to change[40,42,43]. Thus, the instantaneous oxidation state of the active site during OER can be determined, and the sensitivity can be adjusted according to the size of the applied bias interval. According to XANES data and CV data, Ru is easier to be oxidized in the $Ru_{0.5}Ir_{0.5}O_2$ solid solution, so the changes in solid solution capacitance can be regarded as the change of Ru pseudo-capacitance, and the existence of pseudo-capacitance in Ir/Ru based materials is well known[42,44]. Figure 4b shows a typical PVC protocol between the 1.195 V vs. RHE cathode and the 1.245 to 1.645 V vs. RHE anode non-$iR$ correction potential. As shown in Fig. 4c, the part of the PVC (black) reveals oxidation and reduction pulses with current response (red). Figure 4d shows the anodic and inverted cathodic current decay of $Ru_{0.5}Ir_{0.5}O_2$. By integrating the current (OER current is deducted as background current) response of the anodic voltage pulses, the charge (the total charge with respect to the anodic bias) stored in the catalyst at a given potential can be quantified. As highlighted in Fig. 4e, this approach can obtain the relationship between stored charge and potential. The change of charge in potential is bilinear for pre-catalysts, and the slope change indicates that the capacitance was changed. Figure 4f shows the relationship between capacitance and potential, which shows the capacitance of $Ru_{0.5}Ir_{0.5}O_2$ begins to drop from about 2.45 mF at 1.325 V vs. RHE. The capacitance of $C$-$IrO_2$ and $C$-$RuO_2$ begin to change at 1.485 V, 1.505 V vs. RHE, respectively. And the charge-log (current densities) profile (Supplementary Fig. 28) shows that the accumulation of oxidation charge may affect the OER reaction[40]. These results show that the voltages required for the increased the accumulation of Ru oxidation charge in $Ru_{0.5}Ir_{0.5}O_2$ is much lower than those of other reference samples and the applied voltages generate more Ru active sites with high oxidation states in $Ru_{0.5}Ir_{0.5}O_2$.

In addition, we carried out in situ Ru K-edge XAS to further observe the trend of Ru oxidation state change with applied voltage

increasing. As depicted in Supplementary Fig. 29, we observe that the oxidation state of Ru increases with the increase of applied voltage. The oxidation state of Ru increased significantly within 1.41 V vs. RHE. The above results support that the applied voltage has a key role in the promotion of high-valence Ru sites, which are known to be more active than $Ru^{IV+}$ species[13,38,39]. When the applied voltage is increased to 1.51 V vs. RHE, further oxidation of Ru is limited and stability is improved mainly due to the stable Ru-O-Ir local structure. We surmise that the high performance of $Ru_{0.5}Ir_{0.5}O_2$ may mainly be mainly due to the fact that the applied voltage promotes the accumulation of oxidation charge in $Ru_{0.5}Ir_{0.5}O_2$ (that is, more Ru active species with high oxidation state are generated), thus improving the OER activity of the catalyst.

The presence of the local structure of Ru-O-Ir and large amount of capacitance in Ir/Ru based materials may impede electron transmission. Based on this, the transient photo-induced voltage (TPV) test was measured as an effective means to explore the electron transfer behavior of different catalysts[45,46]. The schematic diagram of the test device is shown in Supplementary Fig. 30. Figure 4g shows typical TPV attenuation curves of $Ru_{0.5}Ir_{0.5}O_2$, C-$IrO_2$ and C-$RuO_2$. The decay rate is usually described by fitting the time decay constant ($\tau$) value (the lower $\tau$, the faster the charge transfer rate)[47,48]. The $\tau$ values of $Ru_{0.5}Ir_{0.5}O_2$, C-$IrO_2$ and C-$RuO_2$ were calculated to be 0.32254 ms, 0.18312 ms and 0.11633 ms, respectively. Because the Ru pseudo-capacitance in $Ru_{0.5}Ir_{0.5}O_2$ is easy to change and the oxidation state of Ru is easy to increase. And before and after laser the irradiation of $Ru_{0.5}Ir_{0.5}O_2$ can be regarded as the charge-discharge process of pseudo-capacitance in the material. Simultaneously, there are strong interactions in the local structure of Ru-O-Ir. These results cause that the electron transfer rate of $Ru_{0.5}Ir_{0.5}O_2$ solid solution decreases significantly.

The fast Fourier transform (FFT) technique based on TPV data was used to further analyze the electron transfer behavior of catalysts[46]. Supplementary Fig. 31 shows the FFT curves of $Ru_{0.5}Ir_{0.5}O_2$, C-$IrO_2$ and C-$RuO_2$. None of these curves has obvious peak values, which means that there are no obvious static frequency and periodic frequency components in the TPV attenuation curve[48]. FFT can only identify the frequency component of TPV signal, but not the relationship between frequency and time. In order to study the non-static characteristics of TPV curves, Continuous Wavelet Transform (CWT) was applied to TPV data. 2D and 3D CWT spectra of $Ru_{0.5}Ir_{0.5}O_2$, C-$IrO_2$ and C-$RuO_2$ are shown in Supplementary Fig. 32, which involve three parameters: time, frequency and intensity.

Low frequency represents slow electron transport, and high frequency means fast electron transport[45]. In order to further analyze the dynamics of the electron transport process, the intensities of peak positions at different frequencies (10−80 Hz) were compared with the relationship between time[45-47]. It can be seen from Fig. 4h that in the relatively slow electron transfer process of 10 Hz, the peak position of $Ru_{0.5}Ir_{0.5}O_{22}$ moves backwards at 10 Hz on the time scale, indicating that the charge transfer rate is slowing down. Similarly, comparisons at other higher different frequencies confirm this same conclusion (Supplementary Fig. 33). In addition, at different frequencies, the time difference ($\Delta t$) of peak values between C-$IrO_2$ (black line, $t_1$−$t_2$), C-$RuO_2$ (red line, $t_1$−$t_3$), and $Ru_{0.5}Ir_{0.5}O_2$ was calculated at different frequencies, respectively, using the time of the $Ru_{0.5}Ir_{0.5}O_2$ ($t_1$) as the peak reference value. As shown in Fig. 4i, the values of $\Delta t$ from 10 Hz to 80 Hz indicates that $Ru_{0.5}Ir_{0.5}O_2$ has the slowest transfer rate during the charge transfer process[46].

In summary, $Ru_{0.5}Ir_{0.5}O_2$ exhibits record OER activity in sulfuric acid electrolyte may mainly be due to more Ru active sites with high oxidation states generated at low applied voltage. And the local structure of Ru-O-Ir in $Ru_{0.5}Ir_{0.5}O_2$ has strong interaction and high stability, which prevents excessive oxidation and dissolution of the active site.

## Theoretical insights of OER activity and stability on $Ru_{0.5}Ir_{0.5}O_2$

We used density functional theory (DFT) simulations to rationalize the observed OER activity and understand the effect of Ru on the OER performance in $Ru_{0.5}Ir_{0.5}O_2$ catalyst. Consistent with the XRD pattern (Fig. 1c), $Ru_{0.5}Ir_{0.5}O_2$ crystal were modeled to have a 1T-crystal structure with a unit cell of $a = b = 3.00$ Å and $c = 6.95$ Å (Fig. 5a). Both C-$IrO_2$ and C-$RuO_2$ crystals were modeled as rutile crystal structures (Fig. 5b, c). The lattice parameters of bulk rutile $IrO_2$ were determined to be $a = b = 4.45$ Å and $c = 3.19$ Å. The lattice parameters of bulk rutile $RuO_2$ were determined to be $a = b = 4.54$ Å and $c = 3.14$ Å. In our DFT models for $Ru_{0.5}Ir_{0.5}O_2$, we approached the problem by substituting Ir atoms of 2D $IrO_2$ with Ru atoms to reach an atomic ratio of Ru/Ir = 1:1, which aligns closely with the experimental value. In order to obtain relatively stable $Ru_{0.5}Ir_{0.5}O_2$ structure, we constructed $Ru_{0.5}Ir_{0.5}O_2$ with different permutations of Ir atoms and Ru atoms, and performed optimization calculations on these models (Fig. 5a and Supplementary Fig. 34), the calculation results of the most stable structure are shown in Fig. 5a.

We carried out additional XPS predictions and Bader charge analysis to verify the oxidation states of Ru and Ir species in $Ru_{0.5}Ir_{0.5}O_2$. As shown in Supplementary Fig. 35a, the Ru 3p binding energy exhibits a decrease from C-$RuO_2$ to $Ru_{0.5}Ir_{0.5}O_2$, indicating a lower Ru oxidation state in $Ru_{0.5}Ir_{0.5}O_2$, consistent with experiment. Meanwhile, as shown in Supplementary Fig. 35b, the Ir 4f binding energy exhibits an increase from C-$RuO_2$ to $Ru_{0.5}Ir_{0.5}O_2$, indicating a higher Ir oxidation state in $Ru_{0.5}Ir_{0.5}O_2$, consistent with experiment. These trends are also supported by the Bader charge analysis, as shown in Supplementary Table 8. Thus, both the XPS prediction and Bader charge analysis support the consistent oxidation state change compared with experiment. As shown in Supplementary Table 9, the bond length of Ru-O in $Ru_{0.5}Ir_{0.5}O_2$ increased compared to that of C-$RuO_2$ and the length of Ir-O bonds also increases compared to that of C-$IrO_2$, which is consistent with experimental observations. For solid solutions, there are two factors that affect the bond length of metals-oxygen. One is the crystal structure, and the other is the valence state. In the 1 T phase structure, the metal-oxygen bond length is longer than that of the rutile phase;[29] For valence states, the higher valence state corresponds to the shorter bond length, while the lower valence state corresponds to the longer bond length. In this work, the crystal structure has larger impact on to the bond length of Ir-O, so the bond length of the Ir-O in $Ru_{0.5}Ir_{0.5}O_2$ is longer than that of the corresponding rutile phase.

We chose to study the catalytic performance of $Ru_{0.5}Ir_{0.5}O_2$ (01-10) edge-site[49,50], C-$IrO_2$ (110) surface and C-$RuO_2$ (110) surface[29,51,52](Fig. 5a). We calculated the Gibbs free energy of all intermediates (*OH, *O, and *OOH) in each reaction step to obtain the theoretical OER overpotentials of $Ru_{0.5}Ir_{0.5}O_2$, C-$IrO_2$, and C-$RuO_2$ surfaces, according to the four-step mechanism[1]. Figure 5d and Supplementary Figs. 36a, 37a depicted the reaction cycles of intermediates on the representative catalysts. As shown in Fig. 5e and Supplementary Figs. 36b, 37b, we found that $O_2$ formation is the rate-determining step (RDS) in C-$IrO_2$, and C-$RuO_2$, and the corresponding limiting overpotential is 1.00 V and 0.81 V, respectively. For $Ru_{0.5}Ir_{0.5}O_2$, generation of $O_2$ is the PDS with an overpotential of 0.36 V. Thus, our DFT simulations showed that the $Ru_{0.5}Ir_{0.5}O_2$ has the lowest OER overpotential ($\eta_{OER} = 0.36$ V) compared to C-$IrO_2$ ($\eta_{OER} = 1.00$ V) and C-$RuO_2$ ($\eta_{OER} = 0.81$ V), coincident with our experimental observation that OER stability and catalytic activity of $Ru_{0.5}Ir_{0.5}O_2$ catalyst were significantly improved.

Indeed, our prediction of overpotentials of catalysts are higher than the experiment, but still within the range of reported DFT results, as shown in Supplementary Tables 10 and 11. After systematically comparing the available published results (Supplementary Table 10), we realize that most of the existing DFT calculations overestimate the OER overpotentials of various reported catalysts. Among many factors affecting the overpotential predictions, we found the type of surface

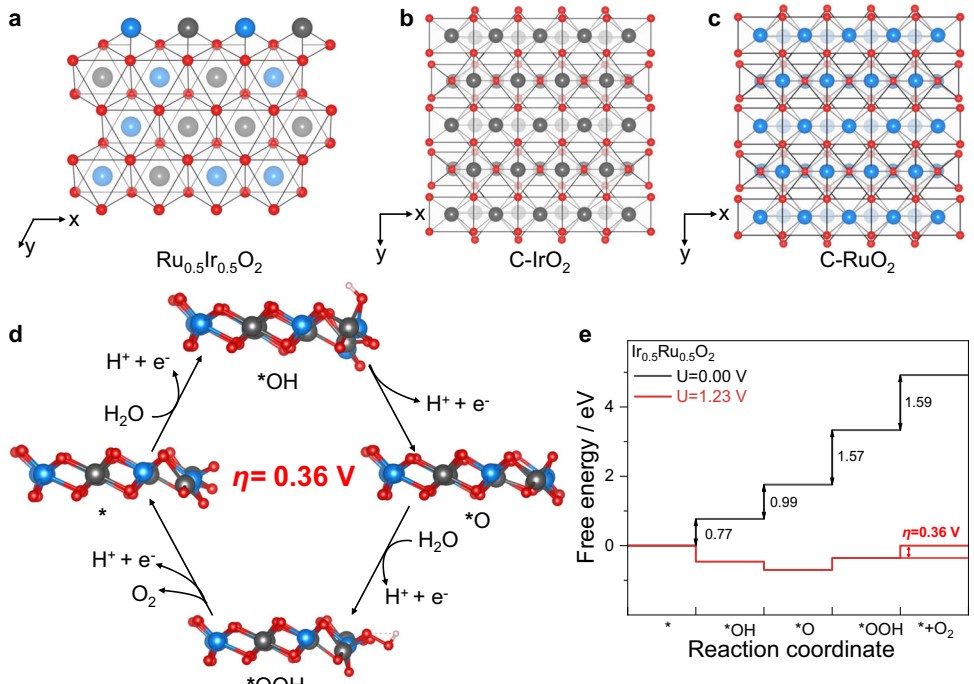

**Fig. 5 | DFT simulation findings of Ru$_{0.5}$Ir$_{0.5}$O$_2$. a** Atomistic structure and $E_{dft}$ of the Ru$_{0.5}$Ir$_{0.5}$O$_2$ (−331.4 eV). Atomistic structures of C-IrO$_2$ (**b**) and C-RuO$_2$ (**c**) (blue, Ru; gray, Ir; red, O). **d** Schematic illustration of OER mechanism on the Ru$_{0.5}$Ir$_{0.5}$O$_2$ (blue, Ru; gray, Ir; red, O; white, H). **e** The reaction paths on Ru$_{0.5}$Ir$_{0.5}$O$_2$ catalyst with the set potential of 0 and 1.23 V. The overpotential (η) is labeled for viewing convenience.

site is most important. We considered the edges of Ru$_{0.5}$Ir$_{0.5}$O$_2$ as active sites. These simulations yielded a considerably lower overpotential of 360 mV, which aligns much more closely with the experimentally observed overpotential of 151 mV. We believe that this now adequately explained the difference between the predicted overpotential of DFT calculation and the experiment results. Remaining differences can be attributed to various sources of uncertainty inherent in both theoretical and experimental methods. Moreover, most of the existing DFT calculations exhibit a systematic overestimation of the OER overpotentials for IrO$_2$ and RuO$_2$ (Supplementary Table 11). These discrepancies can be reasonably attributed to the limitations in current DFT calculations.

In general, Ru$_{0.5}$Ir$_{0.5}$O$_2$ can form more Ru active sites with high-valence at low applied voltage and the amount of oxidative charge stored in the Ru$_{0.5}$Ir$_{0.5}$O$_2$ significantly affect the OER performance. The interactions within the Ru-O-Ir local structure, and two-dimensional structure characteristics play significant roles in stability of Ru$_{0.5}$Ir$_{0.5}$O$_2$ electrocatalyst. DFT results also support that Ru$_{0.5}$Ir$_{0.5}$O$_2$ is more active in OER than rutile phase C-IrO$_2$ and C-RuO$_2$. Because of these advantages, Ru$_{0.5}$Ir$_{0.5}$O$_2$ showed significant superiority over the long-regarded state-of-the-art commercial OER electrocatalysts (IrO$_2$ and RuO$_2$).

## Discussion

In conclusion, we demonstrate that the well-defined 2D Ru$_{0.5}$Ir$_{0.5}$O$_2$ substitutional solid solution prepared by a 2-step mechano-thermal method is highly OER active and acid-stable material. Ru$_{0.5}$Ir$_{0.5}$O$_2$ exhibits a low overpotential of only 151 mV at 10 mA cm$^{-2}$ and attributes its enhanced catalytic activity to the more Ru active sites with high oxidation states generated at low applied voltage, and the surface oxidative charge concentrations are increased. Ru$_{0.5}$Ir$_{0.5}$O$_2$ also shows good stability after 618.3 h operation at 10 mA cm$^{-2}$. In addition to the 2D structural characteristics of Ru$_{0.5}$Ir$_{0.5}$O$_2$, a strong interaction in the local structure of Ru-O-Ir, namely, dominated its high stability, preventing the excessive oxidative dissolution of the active site.

## Methods
### Chemicals
Ruthenium chloride (RuCl$_3$, 99.9%) and Iridium chloride (IrCl$_3$, 99.9%) were purchased from Alfa Aesar Co. Commercial ruthenium (IV) oxide (C-RuO$_2$, 99.9%) and commercial iridium (IV) oxide (C-IrO$_2$, 99%) were purchased from Aladdin Chemical Regent Co. Potassium hydroxide (KOH, 99%) was purchased from Sinopharm Chemical Reagent Co. Nafion solution (5 wt%) was obtained from Sigma-Aldddrich Co. Isopropanol was purchased from Sinopharm Chemical Reagent Co. The continuous ultrathin carbon layer on a holey carbon/formvar support film were purchased from Zhongjingkeyi (Beijing) Film Technology Co., LTD. All other experimental reagents were analytical reagent grade, and all the aqueous solutions used in the experiment were prepared by double distilled water.

### Synthesis of Ru$_{0.5}$Ir$_{0.5}$O$_2$
The Ru$_{0.5}$Ir$_{0.5}$O$_2$ layered structure was prepared via a two-step molten-alkali process[29]. At the first step, the raw materials (149 mg (0.5 M) IrCl$_3$ and 2 g KOH) were mixed and heated by a mechano-thermal method[29]. Then the furnace was cooled down to room temperature. At the second step, the gotten solid production and other raw materials (104 mg (0.5 M) RuCl$_3$ and 2 g KOH) were mixed and thoroughly ground and heated again by a mechano-thermal method[29]. Both heating processes are heated to 800 °C for 2 h. The as-prepared solid production was washed by double distilled water and dried by lyophilization to obtain Ru$_{0.5}$Ir$_{0.5}$O$_2$.

### Characterizations
X-ray powder diffraction (XRD, Philips X'pert PRO MPD diffractometer, Holland) with Cu Kα radiation source (λ = 0.15406 nm) was used to characterize the phase and crystallization of the samples. Transmission electron microscopy (TEM) image, high resolution transmission electron microscopy (HRTEM) image and high-angle annular dark-field scanning TEM (HAADF-STEM) images were recorded via a Talos F200X transmission electron microscope

(USA) under an accelerating voltage of 200 kV. Scanning transmission electron microscopy (STEM) results were collected on a fifth order aberration-corrected transmission electron microscope (JEOL ARM200CF) at 80 kV. Scanning electron microscopy (SEM) was conducted by using a Zeiss G500. X-ray photoelectron spectroscopy (XPS) was conducted on a Thermo Scientific™ K-Alpha™⁺ spectrometer equipped with a monochromatic Al Kα X-ray source (1486.6 eV) operating at 100 W. Samples were analyzed under vacuum ($P < 10^{-8}$ mbar) with a pass energy of 150 eV (survey scans) or 25 eV (high-resolution scans). The experimental peaks were fitted with Avantage software. The topographic height of surface was measured by Atomic force microscopic (AFM, Bruker Dimension Icon). The BET specific surface areas were characterized by American Micromeritics ASAP−2020 porosimeter. Inductively coupled plasma atomic emission spectra (ICP-AES, PerkinElmer Optima 7300 DV) was used to measure the dissolved Ru in electrolyte (0.5 M $H_2SO_4$) after the OER stability testing.

### PVC measurements

PVC was performed by using a RRDE-3A Rotating Ring Disk Electrode Apparatus (ALS Co, Ltd, Japan) when the current was tracked over a period of time (1600 rpm RDE). An GC RDE (3 mm in diameter) loaded with catalysts was employed as the working electrode (mass loading ~283 μg cm$^{-2}$). A saturated calomel electrode (SCE) and a platinum wire were used as the reference electrode and counter electrode. The electrochemical pulse voltammetry tests were measured in $O_2$-saturated 0.5 M $H_2SO_4$. Keep the potential at low potential ($E_{low}$ = 1.195 V vs. RHE) for 5 s, then switch to high potential ($E_{high}$) for 5 s and back to low potential for 5 s. The anodic current is a convolution of capacitive charge, catalyst oxidation and OER currents. The $E_{high}$ was increased from 1.245 to 1.645 V vs. RHE ($Ru_{0.5}Ir_{0.5}O_2$) or 1.345 to 1.745 V vs. RHE (C-$IrO_2$ and C-$RuO_2$) at a speed of 20 mV/step, and the $E_{low}$ remained unchanged. The charge integration scheme applied in the main text is to integrate the current after deducting the background current (OER current is deducted as background current) in the PVC test[40].

### Electrochemical characterization in acetonitrile

The electrolyte was prepared from 0.4 g of tetrabutylammonium perchlorate dispersed in 10 mL anhydrous acetonitrile and thoroughly deoxygenated with pure $N_2$/Ar for 20 min and maintained the positive pressure of the gas during the electrochemical test. A platinum wire was used as the counter electrode and non-aqueous Ag$^+$ electrode was used as reference electrode. The non-aqueous Ag$^+$ reference electrode has been calibrated according to the oxidation peak of ferrocene[53] (Supplementary Fig. 27). The CV of $Ru_{0.5}Ir_{0.5}O_2$, C-$IrO_2$ and C-$RuO_2$ were tested in anhydrous acetonitrile with the scan rate of 30 mV s$^{-1}$ (mass loading ~283 μg cm$^{-2}$).

### TPV measurements

The TPV test is performed with a platinum network covered by the operating electrode of the power sample (1 × 1 cm) and a platinum wire as the counter electrode. The TPV was excited with a nanosecond laser radiation pulse (wavelength of 355 nm and the repetition rate was 5 Hz) from a third harmonic Nd:YAG laser (Beamtech Optronics Co., Ltd.). The TPV signals were amplified by an amplifier and were recorded by an oscilloscope. All measurements were performed at room temperature and under ambient pressure.

### Electrochemical measurements of OER

All electrochemical measurements of OER were carried out in a conventional three-electrode system performed at room temperature with a CHI 760E electrochemical workstation. The synthetic $Ru_{0.5}Ir_{0.5}O_2$ and commercial catalyst (C-$IrO_2$ and C-$RuO_2$) were dispersed 5 mg in 0.9 mL isopropanol solution and 0.1 mL Nafion solution (0.5 wt%), and the ink was formed by ultrasonically ultrasonic action for 1 h to form a homogeneous ink. Then 4 μL of suspension was loaded on glass carbon electrode (GCE, 3 mm in diameter) to prepared the working electrode (mass loading ~283 μg cm$^{-2}$). Finally, the prepared working electrode was dried in room temperature. And the working electrode for stability test was prepared by 28.3 μL of suspension was loaded onto the carbon paper 0.5 × 1.0 cm (283 μg cm$^{-2}$). A platinum wire was used as the counter electrode and SCE as the reference electrode. The SCE electrode were also calibrated in highly pure $H_2$-saturated 0.5 M $H_2SO_4$ electrolytes bubbled with pure hydrogen gas with a scan rate of 5 mV s$^{-1}$ (Supplementary Fig. 10). All electrochemical measurements were carried out in $O_2$-saturated 0.5 M $H_2SO_4$ solution (pH = 0.4 ± 0.1). All linear sweep voltammetry (LSV) polarization curves were obtained with 95% *iR*-compensation at a scan rate of 5 mV s$^{-1}$. The solution resistances were measured by electrochemical impedance spectra (EIS). The EIS test was performed at a frequency range of 100 kHz to 10 mHz with an amplitude of 5 mV. And measured on GCE (3 mm in diameter) with a mass loading of 283 μg cm$^{-2}$ at 1.38 V vs. RHE.

### PEM device

For PEM electrolyzer test, a self-made cell was used as the PEM device and a cation exchange membrane (Nafion 117) as the membrane electrolyte. $Ru_{0.5}Ir_{0.5}O_2$ catalyst (1.0 mg) was used as the anode catalyst and Pt/C (20 wt%, 0.7 mg) was used as the cathode catalyst. To prepare the cathode or anode catalyst ink, the catalyst was suspended into a 0.9 mL mixture of isopropanol and water ($V_{isopropanol}$: $V_{water}$ = 1: 3,) and 100 μL of 5 wt% Nafion solution. The suspension was then ultrasonicated for 1 h until a well-dispersed catalyst ink was formed. The ink of $Ru_{0.5}Ir_{0.5}O_2$ and Pt/C catalysts were sprayed onto Ti wafers (1 × 1 cm$^2$), respectively. And pretreated Ti wafers (1 × 1 cm$^2$) were used as cathode and anode gas diffusion layers (GDLs), respectively. The membrane electrode assembly (MEA) was constructed by placing Nafion 117 membrane in the middle of the Pt/C cathode catalyst-supported Ti wafer and $Ru_{0.5}Ir_{0.5}O_2$ anode catalyst-supported Ti foam GDLs, followed by hot pressing at 130 °C for 5 min under a pressure of 2 MPa. The constructed MEAs were finally applied in the PEM electrolyzer. The PEM electrolyzer was performed by CHI 660E with a high-current amplifier (CHI 680 C) using 0.5 M $H_2SO_4$ as the electrolyte with the flowing rate of 10 mL min$^{-1}$ at room temperature. The performance of the PEM electrolyzer was evaluated by measuring polarization curves from 10 to 2000 mA cm$^{-2}$ at room temperature and ambient pressure. The stability of the PEM electrolyzer using $Ru_{0.5}Ir_{0.5}O_2$ as the anode catalyst was evaluated by measuring the chronoamperometry test at the cell voltages of 1.43 V.

All potentials were referenced to a reversible hydrogen electrode (RHE).

The conversion between the measured potential and the reversible hydrogen electrode (RHE) was calculated by Eq. (1):

$$E(vs. RHE) = E\left(vs. \frac{Hg}{Hg_2Cl_2}\right) + 0.245V \qquad (1)$$

### Calculation of *iR*-correction

Equation (2) determines the 95% *iR*-compensation of the polarization curves of $Ru_{0.5}Ir_{0.5}O_2$, C-$IrO_2$ and C-$RuO_2$.

$$E(95\% \; iR \; compensation) = E(non \; iR \; compensation) - i \times R \times 95\% \qquad (2)$$

Where *i* is the tested current, and *R* is the solution resistance.

## Calculation of turnover frequency (TOF)

Equations (3–6) determine the average active surface atoms per square centimeter of $Ru_{0.5}Ir_{0.5}O_2$, $C-IrO_2$ and $C-RuO_2$[29].

$$\#\text{active sites}_{Ru_xIr_{1-x}O_2} = \frac{0.5\,\text{atom}}{(2.99 \times 2.99)\text{Å}^2 \times \sin 60°} = 6.46 \times 10^{14} \frac{\text{atoms}}{\text{cm}^2} \quad (3)$$

$$\#\text{active sites}_{C-IrO_2} = \frac{2\,\text{atom}}{(4.50 \times 3.15)\text{Å}^2 \times \sqrt{2}} = 9.98 \times 10^{14} \frac{\text{atoms}}{\text{cm}^2} \quad (4)$$

$$\#\text{active sites}_{C-RuO_2} = \frac{2\,\text{atom}}{(4.50 \times 3.11)\text{Å}^2 \times \sqrt{2}} = 10.11 \times 10^{14} \frac{\text{atoms}}{\text{cm}^2} \quad (5)$$

TOF of $Ru_{0.5}Ir_{0.5}O_2$ is calculated as follows Eq. (6).

$$TOF_{BET} = \frac{\left(1.56 \times 10^{15} \frac{O_2}{s} \text{ per } \frac{mA}{cm^2}\right) \times j}{(\text{active sites}) \times A_{EBET}} \quad (6)$$

The TOFs of $C-IrO_2$ and $C-RuO_2$ are calculated according to the same procedure.

## The calculation of ECSA

The ECSA can be qualified based on the following Eq. (7):

$$ECSA = \frac{C_{dl}}{C_s} \quad (7)$$

The electrochemical double-layer capacitance ($C_{dl}$) at non-Faradaic potential range was obtained by measuring the capacitance of double layer at solid-liquid interface employing cyclic voltammetry (CV) with different scan rates (5, 10, 15, 20, 25, 30 and 35 mV s$^{-1}$) in a range from 0.92 and 1.18 V vs. RHE. The $C_s$ values were 0.035 mF cm$^{-2}$.

The $C_{dl}$ was calculated by the Eq. (8):

$$C_{dl} = \frac{\Delta j/2}{\nu} \quad (8)$$

where $\Delta j$ is the difference of the anodic and cathodic currents ($j_a$-$j_c$) at 1.05 V vs. RHE and $\nu$ is the scan rate.

## Density functional theory (DFT) calculations

In this experiment, the Vienna ab initio Simulation Package (VASP)[54,55] version 5.4.4, was used to simulate the density function theory (DFT). To account for the exchange-correlation interaction, a generalized gradient approximation (GGA) in the form of Perdew–Burke–Ernzerhof (PBE) functional was applied[56]. The Projector augmented-wave (PAW) method and the plane wave basis set were used to complete the calculations. In the settings of the computational simulation parameters, the cut-off energy of the plane wave of the PAW is set to 500 eV, and a vacuum layer of 15 Å thickness is introduced for all surfaces in the vertical direction. The Brillouin grid points are used as $3 \times 3 \times 1$ Monkhorst-Pack K points. The convergence condition for the force is 0.03 eV/Å, and the convergence accuracy for the energy is 10$^{-5}$ eV. Note that we constructed a single-layer $Ru_{0.5}Ir_{0.5}O_2$ structure both in stability calculations and reactivity predictions. When these two convergence criteria are met simultaneously, the structure is considered optimal at this point. Based on the VASP calculation, the Gibbs free energy was calculated, and the zero-point energy and contributions of enthalpy and entropy were computed using the VASPKIT program[57]. The solvation effect and applied potential were considered using the implicit solvation model implemented in VASPsol[58,59], and jDFTx[60] was employed to validate the constant potential calculation (Supplementary Fig. 38).

The XPS simulations were conducted to predict the core-level energies of 1s orbital for $Ru_{0.5}Ir_{0.5}O_2$, $C-IrO_2$ and $C-RuO_2$ catalyst implemented in VASP at the PBE-D3 level. In VASP, there are two approaches for the calculation of the core-level shift (CLS), the initial and final state approximation, respectively. The initial state approximation is based on the Kohn–Sham (K.S.) eigenvalues of the core states after a self-consistent calculation of the valence charge density, while the final state approximation requires removing the electron from the core and placing it into the valence[61]. We selected the initial state method to predict CLS because previous studies have shown that such an approach is reliable in reproducing the relative binding energy change as measured experimentally[62]. The work function's effect was not considered during the calculations. For directly compared to the experimental observation, all the calculated core-level binding energies were transformed to absolute values.

## Data availability

The data generated in this study are provided in the Supplementary Information/Source Data file. Source data are provided with this paper.

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

## Acknowledgements

This work is supported by National MCF Energy R&D Program of China (2018YFE0306105), National Key R&D Program of China (2020YFA0406104, 2020YFA0406101), Innovative Research Group Project of the National Natural Science Foundation of China (51821002), National Natural Science Foundation of China (51972216, 52272043, 52271223, 52202107, 52201269), Natural Science Foundation of Jiangsu Province (BK20220028, BK20210735), Key-Area Research and Development Program of Guang Dong Province (2019B010933001), Collaborative Innovation Center of Suzhou Nano Science & Technology, the 111 Project, and Suzhou Key Laboratory of Functional Nano & Soft Materials. We acknowledge Peiping Yu for her help in jDFTx calculation. We acknowledge the support from SSRF (14 W) for the XAS experiments.

## Author contributions

T.C., M.W.S., Y.L., and Z.H.K. supervised the research. W.X.Z., M.W.S., and Z.H.K. conceived the project design. W.X.Z. prepared the samples and measured their electrochemical properties. W.X.Z., M.J.M., H.W.Y., M.W., and J.S. performed the XPS, SEM, AFM, and TEM characterizations. W.X.Z., Y.J.Z., and J.W. performed the CV, PVC, and TPV measurements. W.X.Z., F.L., H.H., Q.S., T.C., M.W.S., Y.L., and Z.H.K. performed most of the data analysis. W.X.Z., K.F., and J.Z. performed the XAS experiment. X.C.S., H.Y., and T.C. performed and analysed the DFT simulations. W.X.Z. and X.C.S. wrote and revised the paper with help from all authors. All authors discussed the results and commented on the manuscript. W.X.Z. and X.C.S. equally contributed to this work.

## Competing interests

The authors declare no competing interests.
