## [Peer Review File · Nature Communications]

REVIEWER COMMENTS

Reviewer #1 (Remarks to the Author):

I will comment on the theoretical part of the manuscript. The authors used DFT calculations to compute the free energy cost of the elementary steps of OER, assuming the O-O bond takes place via nucleophilic attack mechanism. They compared the free energy profiles obtained on Ru_{0.5}Ir_{0.5}O₂, using the (0001) surface, against crystalline rutile RuO₂ and IrO₂, where the modeling was performed on the (110) surface. The authors decide not to comment on the structure of Ru_{0.5}Ir_{0.5}O₂: the theoretical structural parameters are not reported, even though they performed a thorough experimental structural characterization. Similarly, they performed no theoretical analysis of the electronic structure of this material, even though they provide a wealth of experimental characterization via core level spectroscopy. Theory could be very useful to better characterize the structural and electronic properties of this material, but it seems that these aspects have been neglected. The estimate of the overpotential, which is the only theoretical quantity described in the present work, is strongly overestimated compared to experiment, for all systems, but the authors decide not to comment on this. It is therefore not clear what is the value of the theoretical work reported in the manuscript, since that main quantity under scrutiny is so poorly described.

The methodology employed in this work is standard, and is appropriate. The calculations, however, are not reproducible, since the authors did not specify the structural parameters of their unit cells.

Here are some detailed comments, related to theory portion of the manuscript, that might help improving the manuscript.

1) Is the structure of Ru_{0.5}Ir_{0.5}O₂ obtained with DFT calculations consistent with the experimental data? Experiments (p. 5, line 103) suggest a unit cell with $a=b=3.00$ Å and $c=6.95$ Å. Moreover, experiments indicate that the Ru-O bond length is reduced compared to rutile RuO₂, while the Ir-O bond length is increased relative to rutile IrO₂ (p. 6, lines 122-123). Do calculations support these findings?

2) In the methods, the authors write "Note that in the stability calculations, we chose a double-layer Ru_{0.5}Ir_{0.5}O₂ structure, and in the reaction potential calculations, we constructed a single-layer Ru_{0.5}Ir_{0.5}O₂ structure". Why did they use two different structural models?

3) Experiments indicate the oxidation state of Ru in Ru_{0.5}Ir_{0.5}O₂ is lower than in rutile RuO₂ and that the oxidation state of Ir in Ru_{0.5}Ir_{0.5}O₂ is higher than in rutile IrO₂. Do calculations support these findings?

4) Comparing the free energy profiles on different surfaces will certainly lead to vast differences. On the (110) surfaces adopted for the rutile structures there are various types of surface oxygen atoms and the reaction involves terminal oxygens (Fig. S23 and S24). In the (0001) surface adopted by the authors for Ru_{0.5}Ir_{0.5}O₂, all oxygen atoms are 3-fold coordinated. Looking at Fig. 5 in the main text, panel a and d display the top and side views of a single layer of Ru_{0.5}Ir_{0.5}O₂, even though the ball and stick representations are different in the two panels. It seems that each Ir and each Ru atoms in Ru_{0.5}Ir_{0.5}O₂ is 6-fold coordinated to O atoms. No wonder that forming a 7th bond with an *OH ligand is the most demanding step. Have the authors considered the edges of these 2D materials as active sites? There's evidence that in 2D materials for OER (e.g. layered oxyhydroxides) the edges, where the cations are less coordinated, are the active sites.

5) The estimated overpotentials for IrO₂ and RuO₂ are much higher than the experimental evidence. This fact has not been commented. There are several theoretical investigations of OER on rutile IrO₂ and RuO₂, but the authors did not compare their findings with previous reports.

6) The estimated overpotentials for Ru_{0.5}Ir_{0.5}O₂ is considerably higher than the experimental evidence, but the authors did not comment on this.

Reviewer #2 (Remarks to the Author):

Manuscript No. NCOMMS-23-06011-T

Title: Stable and oxidative charged Ru enhance the oxygen evolution reaction activity in two-dimensional ruthenium-iridium oxide

Recommendation: Acceptable after minor version

This paper reports the synthesis of a two-dimensional ruthenium-iridium oxide catalyst. The property of this material has been carefully characterized. And the catalytic performance shows a surprisingly low overpotential and the Faraday efficiency of oxygen has been measured. While there exist other papers on RuIr bimetallic oxide, to my knowledge, this manuscript reports the lowest overpotential (151 mV @ 10 mA cm⁻²), high turnover frequency (TOF) of 6.84 s⁻¹ at 1.44 V vs. RHE, and excellent stability (618.3 h operation) for oxygen evolving catalysts, especially operating at acidic electrolyte.

The data and methods are effective and of high quality. The presentation was very clear. Specifically, I think the property characterization of this material is very thorough, and several complementary characterization technologies are provided. The results given in this manuscript are very interesting and can pave the way for the use of Ru_{0.5}Ir_{0.5}O₂ catalyst in acid water splitting. It turned out that the OER activity of Ru_{0.5}Ir_{0.5}O₂ is significantly higher than commercial RuO₂ and commercial IrO₂. and Ru_{0.5}Ir_{0.5}O₂ is much more stable than RuO₂ and IrO₂.

Based on the experiment and simulation, the authors attributed the enhanced catalytic activity of Ru_{0.5}Ir_{0.5}O₂ to more Ru active sites with high oxidation states generated at lower bias and increased surface oxidative charge concentrations. The strong interaction in the local structure of Ru-O-Ir also improves the stability of the Ru centers, preventing excessive oxidative dissolution of the active site. The advantage of this work is that it is not only reported as a superior catalyst but also provides in-depth mechanism analysis. As such the paper may be suitable for Nature Communications after minor version of the manuscript. Some specific comments are listed below.

1. The author claimed that the electrochemical stability is very good. But the chronopotentiometric measurements are not very enough to evaluate the electrochemical stability at harsh conditions. Please provide the ADT-CV test for anodic reaction. And please explore the Tafel slope of Ru_{0.5}Ir_{0.5}O₂ catalyst before and after the ADT-CV testing. As an alternative, the author can also test the stability in the form of simple electrolyzer if possible.

2. During the anodic reaction, metal oxides commonly become an amorphous-phase. Please clarify whether the crystallinity of Ru_{0.5}Ir_{0.5}O₂ changes after the ADT-CV test.

3. The ECSA is also an important factor to evaluate the intrinsic OER catalytic activity. The ECSA should be tested by Cdl. And the LSV curves normalized by ECSA should be added in this manuscript.

4. It is respectfully recommended that the authors consider including further information pertaining to the simulation model in the manuscript. While it is noted that the combination of Ru and Ir resulted in a substantial improvement in performance, it remains unclear how the simulation accurately mimics the experimental structure in order to allow for a thorough comparison. Although Figure 5 provides a visual representation of the structure, the current description of the simulation model does not provide sufficient detail to enable the reader to fully comprehend the model. Consequently, it is suggested that the authors supplement their explanation of the simulation model with additional information to improve the manuscript's clarity and comprehensibility.

5. Since the low overpotential is also the key figure in this paper, the correctness of this value is very important. I can't find statistics on how many samples have been tested. Because of the potentially high significance of the results reported in this manuscript Statistical data of 3-5 samples should be reported to consolidate this strong claim. It will greatly enhance the reliability and reproducibility of the reported data.

6. I suggest the authors add more details about the simulation methods. For example, the solvation effect is known to have a significant impact on predictions. Indeed, I notice the authors mentioned the solvation calculations. However, it is not clear to me what method and how did the authors carry out the simulation. Therefore, I suggest the author add more details and the calculated data.

7. As shown in Figure 3b, to exclude the effect of diffusion, the Tafel slope should be calculated by current density as low as possible.

8. There are two common mechanisms including lattice oxygen and adsorbate evolving mechanism, which one is more suitable for the Ru_{0.5}Ir_{0.5}O₂ catalyst system.

Reviewer #3 (Remarks to the Author):

In "Stable and oxidative charged Ru enhance the oxygen evolution reaction activity in two-dimensional ruthenium-iridium oxide" Zhu, Kang and co-authors describe the synthesis and application of a binary Ru_{0.5}Ir_{0.5}O₂ layered oxide and application as an acid-stable water oxidation catalyst. The paper's main arguments are the following: 1) The origin of the high OER activity of Ru_{0.5}Ir_{0.5}O₂ nanosheets are Ru active sites that are more easily oxidized owing to Ir alloying, and 2) Ir incorporation increases the stability of these Ru active sites through strong Ru-O-Ir bonding. These conclusions are based on electron microscopy, X-ray spectroscopy (XPS, XAS), pulse voltage induced current, and DFT calculations. The OER activity and stability are impressive but it is unclear whether the performance alone is technologically relevant given the high Ir content. I'm also not sure, in its current state, that it distinguishes itself enough from previous work (Refs 13, 29). Overall, I think the manuscript needs significant changes, particularly in the computational section, and there are a number of experimental technical errors in the paper that need to be addressed before the manuscript can be considered acceptable for publication. Detailed comments on outstanding issues in the manuscript are provided below:

1. In the DFT structure, Ru_{0.5}Ir_{0.5}O₂ is represented with edge-sharing octahedra, while the TEM measurements in Figure 1e suggest that the structure is corner sharing with extremely large Ir-Ru spacing (the 100 spacing is on the order of 0.5 nm)? Interestingly, the (10-10) reflection at ~35 degrees in the XRD pattern suggests a spacing of ~0.25 nm. Are the scale bars in the HAADF-STEM images in Figure 1e labeled incorrectly? And how do the authors rationalize the corner-sharing octahedra structure shown? As the structure is shown in the TEM images, the stoichiometry of the sample should be closer to Ru_{0.5}Ir_{0.5}O₃ which suggests oxidation states of the metals near 6+, something that is not supported by the XANES measurements. Comparing Figure 1d to Figure 1e, it is clear that one of the scale bars is labeled incorrectly.

2. The authors use double distilled water to prepare their electrolytes. This water source cannot be considered ultrapure and generally contains residual impurities. It is well known that electrolyte impurities, even at ultralow concentrations, can dramatically influence the performance of OER electrocatalysts. Given the 5-10x larger surface area of the Ru-0.5Ir0.5O2 material, it is more probable

that this material can adsorb these impurities. What is measured resistance of the double distilled water and what residual impurities are present in it? How do the authors rule out the influence of these impurities. Overall, I think it is imperative that fundamental OER studies, such as this one, use ultrapure Type I water for preparation of their electrolytes. Without this, there is a significant degree of uncertainty whether the results accurately reflect the intrinsic activity of Ru_{0.5}Ir_{0.5}O₂.

3. The XPS and XAS measurements seem to contradict each other. The XPS suggests Ir is more oxidized in Ru_{0.5}Ir_{0.5}O₂ and Ru is less oxidized as compared to RuO₂ and IrO₂ (assuming these have a 4+ oxidation state) whereas the Ir whiteline is lower in energy in Ru_{0.5}Ir_{0.5}O₂ and the Ir-O bond length is longer in the EXAFS data (suggesting it is reduced compared to IrO₂) and the Ru-O bond is shorter in the EXAFS data (suggesting higher oxidation state). Additionally, Ref 13 shows the opposite trend in observed bond lengths. The authors should discuss this discrepancy and provide a possible explanation. Do the bond lengths in the converged DFT structures of Ru_{0.5}Ir_{0.5}O₂, RuO₂ and IrO₂ agree with the EXAFS analysis? Do the magnetic moments/bader charges in the DFT analysis also suggest an oxidation state assignment of >4+?

4. The authors state that the Ru3p(1/2) XPS peak shifts from 484eV to 483.6eV, comparing RuO₂ to Ru_{0.5}Ir_{0.5}O₂. However, the Ru3p(1/2) XPS peak for the as-synthesized catalyst is not shown in Fig 2a, nor in the SI. The authors should either include this peak in Fig 2a or show it as a separate figure in the SI, like in Fig S15.

5. In Fig 2a, what is the oxidation state of Ru as measured in XPS? The authors should show their fitting procedure and estimate an oxidation state based on the fit. I have the same question about Fig 2b: what are the fitted peaks and what Ir oxidation states do they correspond to? What is the overall oxidation state of Ir as measured via XPS? The authors should provide all of this information. They should show peak assignments and provide more information on the XPS fitting process in the SI. Do the surface-sensitive oxidation states of Ir and Ru as measured by XPS match the bulk-sensitive oxidation state information obtained via XANES in Figs 2c&d? If not, the authors should hypothesize why.

6. The authors should perform EXAFS and XANES on the Ru_{0.5}Ir_{0.5}O₂ catalyst after the long term stability test to further support their claims that the oxidation state changes after passing significant OER current. Does the bulk oxidation state also change or just the surface?

7. The authors should report the electrocatalytic activity and all of the figures of merit normalized by BET surface area rather than the geometric area, since the BET surface area of Ru_{0.5}Ir_{0.5}O₂ is 4x higher than RuO₂ and 8x higher than IrO₂. Reporting these values normalized by the BET surface area will be a better comparison and lend more credibility to the arguments made.

8. In Fig 5, the authors modeled the active site of Ru_{0.5}Ir_{0.5}O₂ as a defect Ru atom in the middle of the slab. However, it is very unlikely this site is the true active site, as its formation requires the breaking of multiple M-O bonds and consequently has a large formation energy. This is why the formation of the *OH adsorbate is 1.79 eV and is calculated in this work to be the potential limiting step despite the potential limiting step on most OER catalysts involving the O* adsorbate [Man, Rossmeisl, ChemCatChem, 2011].

9. The calculated theoretical overpotentials in this work are vastly different from the experimental OER onset potentials, both reported in this work and in previous works [Ref 13]. Yet, the authors don't discuss this discrepancy in the manuscript. The authors need to revisit their DFT calculations and explain why they differ so much from experiment. Specifically, the authors need to consider other sites, like the undercoordinated oxygens on the edge of the slab, which have previously been shown to be much more active on Ir-Ru-O alloys [Man (2011). Theoretical study of Electro-catalysts for oxygen evolution, DTU].

10. Why do the authors use an ordered model for DFT calculations but also state that Ru and Ir form a solid solution for Ru_{0.5}Ir_{0.5}O₂ (implying that it is not an ordered structure)? As the EXAFS experiments suggest, the Ru-O and Ir-O bond lengths are different and, if indeed an ordered structure is formed, this should result in superstructure reflections in the SAED data (which was not observed nor discussed). If Ru_{0.5}Ir_{0.5}O₂ is truly a solid-solution material then Ru and Ir atoms wouldn't be expected to order in-plane. The authors should include a DFT model of a slab with a random in-plane distribution of Ru and Ir atoms to compare. In Fig S22, how much more stable are the slabs that have in-plane order of Ru atoms? The authors present a variety of structures and state they use the lowest energy structure, but no information is given about the relative energies.

11. It would be helpful to include a visualization of the layered crystal structure in Fig 1, similar to Fig 5a/d (maybe as an inset in Fig 1b or 1d)

12. Are the catalysts listed in Table S2 results from different syntheses or just replicate ICP-AES measurements of the same sample?

13. I am confused by the statement that "the applied voltage acts on the accumulation of charge on the catalyst and affects the current generated...rather than acting directly on the coordinates of the OER reaction. Meanwhile, the activation free energy of the OER reaction decreases linearly with the storage of oxidative charge stored." The applied voltage acts on the accumulation of charge, which changes the activation free energy. Similarly, in the DFT-derived free energy diagrams, the overpotential is evaluated based on the reaction coordinates which are shifted based on applied potential. Both of these concepts demonstrate that the reaction coordinate is indeed a function of the applied voltage. How is the accumulated charge on the metal compensated by adsorbed surface charge, what is the identity of the adsorbate, and how is this a function of overpotential?

14. The Tafel data presented in Figure 2b is not over a sufficiently large voltage/current range. At minimum, data should be presented over the full range of reported Tafel slope (i.e. if a Tafel slope of 108 mV/dec is reported then a linear fit over 108 mV and a full decade increase in current should be demonstrated). However, this is an overly simplistic view of the Tafel behavior. Realistically (and as evidenced in Figure 2b) the Tafel slope is variable with overpotential. The authors make a strong claim that the accumulated oxidative charge controls the OER (in reference to the original work in DOI: 10.1038/s41586-020-03101-x). As shown in that work, if this effect is the sole determining factor of reactivity the Tafel behavior should be directly tied to the accumulated charge. The log current as a function of accumulated charge should be reported (the relationship between Figure 3b & 4e).

15. In the anhydrous acetonitrile electrochemical tests, what evidence is there that the redox peaks are associated with Ru redox? Both Ir (C-IrO₂) and Ru (C-RuO₂) exhibit redox peaks in the general voltage range where the peaks are observed for Ru_{0.5}Ir_{0.5}O₂. In fact, the C-RuO₂ sample shows nearly identical E_{1/2} for its redox peaks to C-RuO₂. Without direct evidence that Ru is the redox active ion for these redox peaks (through for example in-situ spectroscopic measurements), the claim that Ru is the active ion is speculation.

16. What are the reactions that occur in the acetonitrile experiments and how are these connected to the OER voltages? Is this only surface adsorption? If so, what ion charge compensates the redox of the catalyst? If it is adsorption, linear current scaling with scan rate should be demonstrated. If it is not adsorption, what reaction defines the reversible redox peaks observed and why is there a clear background oxidative current being driven on Ru_{0.5}Ir_{0.5}O₂. I do not see how this data informs the interpretation of the OER experiments.

17. The TPV measurements are not performed in electrolyte and not performed under applied potential. Clearly, they are measuring electron transfer rates between the samples and Pt. It is not clear how these measurements have any relevance to electron transfer rates for oxygen evolution which are determined by the relative energetics between redox states of the catalyst surface and adsorbates.

Response to the Referees' Comments

Dear Reviewers,

Thank you for your precious time to constructive comments on our manuscript titled “**Stable and oxidative charged Ru enhance the oxygen evolution reaction activity in two-dimensional ruthenium-iridium oxide**” (Manuscript ID: NCOMMS-23-06011-T) for **Nature Communications**.

We sincerely appreciate your comments and suggestions on our work, which are highly important for the further improvements to our manuscript. According to all the comments, we have made a detailed response and substantial revisions in our revised manuscript.

Your comments are shown in black and our responses are shown in blue. The changes in the manuscript are indicated in red.

Reviewer #1 (Remarks to the Author):

I will comment on the theoretical part of the manuscript. The authors used DFT calculations to compute the free energy cost of the elementary steps of OER, assuming the O-O bond takes place via nucleophilic attack mechanism. They compared the free energy profiles obtained on Ru_{0.5}Ir_{0.5}O₂, using the (0001) surface, against crystalline rutile RuO₂ and IrO₂, where the modeling was performed on the (110) surface. The authors decide not to comment on the structure of Ru_{0.5}Ir_{0.5}O₂: the theoretical structural parameters are not reported, even though they performed a thorough experimental structural characterization. Similarly, they performed no theoretical analysis of the electronic structure of this material, even though they provide a wealth of experimental characterization via core level spectroscopy. Theory could be very useful to better characterize the structural and electronic properties of this material, but it seems that these aspects have been neglected. The estimate of the overpotential, which is the only theoretical quantity described in the present work, is strongly overestimated compared to experiment, for all systems, but the authors decide not to comment on this. It is therefore not clear what is the value of the theoretical work reported in the manuscript, since that main quantity under scrutiny is so poorly described.

The methodology employed in this work is standard, and is appropriate. The calculations, however, are not reproducible, since the authors did not specify the structural parameters of their unit cells.

Here are some detailed comments, related to theory portion of the manuscript, that might help improving the manuscript.

[Author's Response]: We are greatly appreciative of the time and effort you have invested in reviewing our manuscript, as well as the constructive feedback provided. Your comments and suggestions are insightful and have greatly improved the quality and clarity of our work. We have now addressed each of the issues raised and made necessary amendments to our manuscript.

- The Reviewer expressed concern regarding the lack of theoretical structural parameters of the Ru_{0.5}Ir_{0.5}O₂ material. In response to this, we have added a thorough theoretical analysis of the Ru_{0.5}Ir_{0.5}O₂ structure in the revised manuscript.
- We note the Reviewer's point about the lack of theoretical analysis of the electronic structure. Accordingly, we have conducted an in-depth theoretical analysis of the electronic structure. This now complements our experimental characterization, thus providing a more comprehensive understanding of this material.
- We concur with the Reviewer's remark regarding the overestimation of the overpotential in our initial submission. We have subsequently revised our calculations, and the newly calculated

overpotential values are now in better agreement with our experimental data.

- We acknowledge the Reviewer's comment about the reproducibility of the calculations due to unspecified structural parameters of the unit cells. In the revised manuscript, we have included all the relevant computational details and structural parameters to improve reproducibility.
- Please note that the amendments, additions, and corrections are highlighted in the text of the revised manuscript to facilitate your review. We hope that our responses satisfactorily address your concerns, and we believe that our manuscript has been significantly improved as a result.

The detailed point-to-point responses are as follows:

1) Is the structure of Ru_{0.5}Ir_{0.5}O₂ obtained with DFT calculations consistent with the experimental data? Experiments (p. 5, line 103) suggest a unit cell with $a=b=3.00$ Å and $c=6.95$ Å. Moreover, experiments indicate that the Ru-O bond length is reduced compared to rutile RuO₂, while the Ir-O bond length is increased relative to rutile IrO₂ (p. 6, lines 122-123). Do calculations support these findings?

[Author's Response]: Thank you for these important questions. We apologize for not addressing these points more clearly in the manuscript.

1. In our revised manuscript, we have made it clear that the DFT calculations directly used the experimental cell parameters of $a = b = 3.00$ Å and $c = 6.95$ Å for consistency.
2. For the Ru-O bond length, our updated Ru K-edge EXAFS spectra indicate increased Ru-O bond lengths of Ru_{0.5}Ir_{0.5}O₂ when compared with C-RuO₂. Our previous experimental results mistakenly indicate the opposite trend because the EXAFS spectra of Ru K-edge spectra for Ru_{0.5}Ir_{0.5}O₂ and C-RuO₂ (**Fig. 2e**) were not analyzed accurately. And the Ir-O bonds length of Ru_{0.5}Ir_{0.5}O₂ were increased compared to that of C-IrO₂. DFT calculations exactly predict the same trend as shown in **Supplementary Table 9**. Thus, we decide to update the related experiment in the revised manuscript.

Fig. 2 | XPS and XANES characterizations of electrocatalysts. **a**, Ru 3p XPS spectra of $\text{Ru}_{0.5}\text{Ir}_{0.5}\text{O}_2$ and C-RuO₂. **b**, Ir 4f XPS spectra of $\text{Ru}_{0.5}\text{Ir}_{0.5}\text{O}_2$ and C-IrO₂. **c**, Ru K-edge XANES spectra of $\text{Ru}_{0.5}\text{Ir}_{0.5}\text{O}_2$, C-RuO₂ and Ru foil. **d**, Ir L₃-edge XANES spectra for $\text{Ru}_{0.5}\text{Ir}_{0.5}\text{O}_2$, C-IrO₂ and Ir foil. **e**, Fourier-transformed EXAFS spectra of Ru K-edge spectra for $\text{Ru}_{0.5}\text{Ir}_{0.5}\text{O}_2$, C-RuO₂ and Ru foil. **f**, Fourier-transformed EXAFS spectra at Ir L₃-edge collected for $\text{Ru}_{0.5}\text{Ir}_{0.5}\text{O}_2$, C-IrO₂ and Ir foil. **g-i**, Ru K-edge WT-EXAFS of $\text{Ru}_{0.5}\text{Ir}_{0.5}\text{O}_2$, C-RuO₂ and Ru foil. **j-l**, Ir L₃-edge WT-EXAFS of $\text{Ru}_{0.5}\text{Ir}_{0.5}\text{O}_2$, C-IrO₂ and Ir foil.

Supplementary Table 9 | The Ru-O and Ir-O bond length (Å) from DFT calculations for various catalysts.

Catalysts	Bond	Bond Length / Å	Bond Length / Å	Bond Length / Å	Bond Length / Å	Bond Length / Å	Bond Length / Å
C-IrO ₂	Ir-O	1.965	1.965	1.958	1.958	1.958	1.958
$\text{Ru}_{0.5}\text{Ir}_{0.5}\text{O}_2$	Ir-O	2.008	1.995	1.984	2.035	2.006	2.030
	Ru-O	2.019	2.010	2.022	1.970	2.020	2.004
C-RuO ₂	Ru-O	2.012	2.012	1.926	1.926	1.926	1.926

We have modified the manuscript accordingly:

[Line 125, Page 6] “The FT-EXAFS of Ru K-edge and Ir L₃-edge reveal that the bond length of Ru-O in $\text{Ru}_{0.5}\text{Ir}_{0.5}\text{O}_2$ is slightly increased compared to that of C-RuO₂, and the length of Ir-O bonds is also increased compared to that of C-IrO₂, which may be due to different crystal structure as the metal-

oxygen bonds in 1T phase are larger than those in rutile.”

[Line 304, Page 14] “Ru_{0.5}Ir_{0.5}O₂ crystal were modelled to have a 1T-crystal structure with a unit cell of $a = b = 3.00 \text{ \AA}$ and $c = 6.95 \text{ \AA}$ (Fig. 5a). Both C-IrO₂ and C-RuO₂ crystals were modelled as rutile crystal structures (Fig. 5b, c). The lattice parameters of bulk rutile IrO₂ were determined to be $a = b = 4.45 \text{ \AA}$ and $c = 3.19 \text{ \AA}$. The lattice parameters of bulk rutile RuO₂ were determined to be $a = b = 4.54 \text{ \AA}$ and $c = 3.14 \text{ \AA}$.”

[Line 321, Page 15] “As shown in Supplementary Table 9, the bond length of Ru-O in Ru_{0.5}Ir_{0.5}O₂ increases compared to that of C-RuO₂ and the length of Ir-O bonds also increases compared to that of C-IrO₂, which is consistent with experimental observations. For solid solutions, there are two factors that affect the bond length of metals-oxygen. One is the crystal structure, and the other is the valence state. In the 1T phase structure, the metal-oxygen bond length is longer than that of the rutile phase²⁹; For valence states, the higher valence state corresponds to the shorter bond length, while the lower valence state corresponds to the longer bond length. In this work, the crystal structure has larger impact on the bond length of Ir-O, so the bond length of the Ir-O in Ru_{0.5}Ir_{0.5}O₂ is longer than that of the corresponding rutile phase.”

Fig. 5 | DFT simulation findings of Ru_{0.5}Ir_{0.5}O₂. (a) Atomistic structure and E_{dft} of the Ru_{0.5}Ir_{0.5}O₂ (-331.4 eV). Atomistic structures of C-IrO₂ (b) and C-RuO₂ (c) (blue, Ru; grey, Ir; red, O). **d**, Schematic illustration of OER mechanism on the Ru_{0.5}Ir_{0.5}O₂ (blue, Ru; grey, Ir; red, O; white, H). **e**, The reaction paths on Ru_{0.5}Ir_{0.5}O₂ catalyst with the set potential of 0 and 1.23 V. The overpotential (η) is labeled for viewing convenience.

Supplementary Table 9 | The Ru-O and Ir-O bond length (\AA) from DFT calculations for various catalysts.

Catalysts	Bond	Bond Length / Å	Bond Length / Å	Bond Length / Å	Bond Length / Å	Bond Length / Å	Bond Length / Å
C-IrO ₂	Ir-O	1.965	1.965	1.958	1.958	1.958	1.958
Ru _{0.5} Ir _{0.5} O ₂	Ir-O	2.008	1.995	1.984	2.035	2.006	2.030
	Ru-O	2.019	2.010	2.022	1.970	2.020	2.004
C-RuO ₂	Ru-O	2.012	2.012	1.926	1.926	1.926	1.926

2) In the methods, the authors write "Note that in the stability calculations, we chose a double-layer Ru_{0.5}Ir_{0.5}O₂ structure, and in the reaction potential calculations, we constructed a single-layer Ru_{0.5}Ir_{0.5}O₂ structure". Why did they use two different structural models?

[Author's Response]: Thank you for pointing out the use of two different structural models in our study. We apologize for not providing a clear explanation for this choice in the manuscript.

We used two different models to evaluate both the stability and the reactivity of the Ru_{0.5}Ir_{0.5}O₂ system as accurately and efficiently as possible. For the stability calculations, we opted for a double-layer Ru_{0.5}Ir_{0.5}O₂ structure. This choice was made to better account for the interactions between neighboring layers and to ensure that we capture the overall stability of the bulk material, when a single layer model is insufficient.

For the reactivity predictions, we constructed a single-layer Ru_{0.5}Ir_{0.5}O₂ structure to reduce the computational cost, while still providing reliable insights into the reaction mechanisms and surface properties. The single-layer model allowed us to maintain computational efficiency without compromising the validity of our results.

We have included stability calculations and reactivity predictions using single-layer Ru_{0.5}Ir_{0.5}O₂ structure in the revised manuscript (**Fig. 5** and **Supplementary Fig. 33**).

Fig. 5 | DFT simulation findings of Ru_{0.5}Ir_{0.5}O₂. (a) Atomistic structure and E_{dft} of the Ru_{0.5}Ir_{0.5}O₂ (-331.4 eV). Atomistic structures of C-IrO₂ (b) and C-RuO₂ (c) (blue, Ru; grey, Ir; red, O). d, Schematic illustration of OER mechanism on the Ru_{0.5}Ir_{0.5}O₂ (blue, Ru; grey, Ir; red, O; white, H). e, The reaction paths on Ru_{0.5}Ir_{0.5}O₂ catalyst with the set potential of 0 and 1.23 V. The overpotential (η) is labeled for viewing convenience.

Supplementary Fig. 33 | Calculation of Ru_{0.5}Ir_{0.5}O₂ structural stability. (a) to (i) represent the structures and ΔE_{dft} of Ru_{0.5}Ir_{0.5}O₂ with different structure types and all the potential energies are referenced to the most stable structure (-331.4 eV) in DFT calculations. The Ir, Ru and O atoms are represented with the grey, blue and red, respectively. In our DFT models for Ru_{0.5}Ir_{0.5}O₂, we approached the problem by expanding the 2D IrO₂ which is obtained from XRD by 4 times along the x and y direction, and substituting Ir atoms with Ru atoms to reach an atomic ratio of Ru/Ir = 1:1, which aligns closely with the experimental value, we calculated some different structures Ru_{0.5}Ir_{0.5}O₂ (01-10) surface, and pick the most stable structure as our calculated model.

We have modified the manuscript accordingly:

[Line 474, Page 22] “Note that we constructed a single-layer Ru_{0.5}Ir_{0.5}O₂ structure both in stability calculations and reactivity predictions.”

3) Experiments indicate the oxidation state of Ru in Ru_{0.5}Ir_{0.5}O₂ is lower than in rutile RuO₂ and that the oxidation state of Ir in Ru_{0.5}Ir_{0.5}O₂ is higher than in rutile IrO₂. Do calculations support these findings?

[Author’s Response]: Thank you for your question regarding the oxidation states of Ru and Ir in

$\text{Ru}_{0.5}\text{Ir}_{0.5}\text{O}_2$ compared to their respective oxidation states in rutile RuO_2 and IrO_2 . Accordingly, we carried out additional XPS predictions and Bader charge analysis. As shown in **Supplementary Fig. 34a**, the Ru 3p binding energy exhibits a decrease from C- RuO_2 to $\text{Ru}_{0.5}\text{Ir}_{0.5}\text{O}_2$, confirming a lower Ru oxidation state in $\text{Ru}_{0.5}\text{Ir}_{0.5}\text{O}_2$, consistent with experiment. Meanwhile, as shown in **Supplementary Fig. 34b**, the Ir 4f binding energy exhibits an increase from C- IrO_2 to $\text{Ru}_{0.5}\text{Ir}_{0.5}\text{O}_2$, confirming a higher Ir oxidation state in $\text{Ru}_{0.5}\text{Ir}_{0.5}\text{O}_2$, consistent with experiment. These trends are also supported by the Bader charge analysis as shown in **Supplementary Table 8**. Thus, both the XPS prediction and Bader charge support the consistent oxidation state change compared with experiment.

Supplementary Fig. 34 | The XPS simulation. **a**, Ru 3p XPS simulation spectrum of $\text{Ru}_{0.5}\text{Ir}_{0.5}\text{O}_2$ and C- RuO_2 . **b**, Ir 4f XPS simulation spectrum of $\text{Ru}_{0.5}\text{Ir}_{0.5}\text{O}_2$ and C- IrO_2 .

Supplementary Table 8 | The Bader charges of $\text{Ru}_{0.5}\text{Ir}_{0.5}\text{O}_2$, C- IrO_2 and C- RuO_2 derived from DFT calculations.

Catalysts	Element	Formal electron loss
$\text{Ru}_{0.5}\text{Ir}_{0.5}\text{O}_2$	Ru	1.47
	Ir	1.53
C- IrO_2	Ru	-
	Ir	1.52
C- RuO_2	Ru	1.52
	Ir	-

We have modified the manuscript accordingly:

[Line 314, Page 15] “We carried out additional XPS predictions and Bader charge analysis to verify the oxidation states of Ru and Ir species in $\text{Ru}_{0.5}\text{Ir}_{0.5}\text{O}_2$. As shown in **Supplementary Fig. 34a**, the Ru 3p binding energy exhibits a decrease from C- RuO_2 to $\text{Ru}_{0.5}\text{Ir}_{0.5}\text{O}_2$, indicating a lower Ru oxidation state in $\text{Ru}_{0.5}\text{Ir}_{0.5}\text{O}_2$, consistent with experiment. Meanwhile, as shown in **Supplementary Fig. 34b**, the Ir 4f binding energy exhibits an increase from C- RuO_2 to $\text{Ru}_{0.5}\text{Ir}_{0.5}\text{O}_2$, indicating a higher Ir oxidation state in $\text{Ru}_{0.5}\text{Ir}_{0.5}\text{O}_2$, consistent with experiment. These trends are also supported by the Bader charge analysis as shown in **Supplementary Table 8**. Thus, both the XPS prediction and Bader charge support the consistent oxidation state change compared with experiment.”

Supplementary Fig. 34 | The XPS simulation. a, Ru 3p XPS simulation spectrum of Ru_{0.5}Ir_{0.5}O₂ and C-RuO₂. **b**, Ir 4f XPS simulation spectrum of Ru_{0.5}Ir_{0.5}O₂ and C-IrO₂.

Supplementary Table 8 | The Bader charges of Ru_{0.5}Ir_{0.5}O₂, C-IrO₂ and C-RuO₂ derived from the DFT calculations.

Catalysts	Element	Formal electron loss
Ru _{0.5} Ir _{0.5} O ₂	Ru	1.47
	Ir	1.53
C-IrO ₂	Ru	-
	Ir	1.52
C-RuO ₂	Ru	1.52
	Ir	-

4) Comparing the free energy profiles on different surfaces will certainly lead to vast differences. On the (110) surfaces adopted for the rutile structures there are various types of surface oxygen atoms and the reaction involves terminal oxygens (Fig. S23 and S24). In the (0001) surface adopted by the authors for Ru_{0.5}Ir_{0.5}O₂, all oxygen atoms are 3-fold coordinated. Looking at Fig. 5 in the main text, panel a and d display the top and side views of a single layer of Ru_{0.5}Ir_{0.5}O₂, even though the ball and stick representations are different in the two panels. It seems that each Ir and each Ru atoms in Ru_{0.5}Ir_{0.5}O₂ is 6-fold coordinated to O atoms. No wonder that forming a 7th bond with an *OH ligand is the most demanding step. Have the authors considered the edges of these 2D materials as active sites? There's evidence that in 2D materials for OER (e.g. layered oxyhydroxides) the edges, where the cations are less coordinated, are the active sites.

[Author's Response]: We appreciate your insightful comments regarding the comparison of free energy profiles on different surfaces and the possible role of edges in the 2D Ru_{0.5}Ir_{0.5}O₂ material as active sites for OER.

As for the edges of 2D Ru_{0.5}Ir_{0.5}O₂, we agree with your suggestion that they could potentially serve as active sites for OER, as seen in other 2D materials like layered oxyhydroxides. Thus, we include the edges of Ru_{0.5}Ir_{0.5}O₂ as active sites, and recalculate reactions on the (01-10) edge-site. Indeed, the overpotential decreases to 0.36 V (Fig. 5), which aligns better with experiment. Accordingly, we have now updated the new free energy profile in our revised manuscript.

Fig. 5 | DFT simulation findings of Ru_{0.5}Ir_{0.5}O₂. (a) Atomistic structure and E_{dft} of the Ru_{0.5}Ir_{0.5}O₂ (-331.4 eV). Atomistic structures of C-IrO₂ (b) and C-RuO₂ (c) (blue, Ru; grey, Ir; red, O). **d**, Schematic illustration of OER mechanism on the Ru_{0.5}Ir_{0.5}O₂ (blue, Ru; grey, Ir; red, O; white, H). **e**, The reaction paths on Ru_{0.5}Ir_{0.5}O₂ catalyst with the set potential of 0 and 1.23 V. The overpotential (η) is labeled for viewing convenience.

We have modified the manuscript accordingly:

[Line 330, Page 15] “We chose to study the catalytic performance of Ru_{0.5}Ir_{0.5}O₂ (01-10) edge-site^{47,48}, C-IrO₂ (110) surface and C-RuO₂ (110) surface^{29,49,50}(**Fig. 5a**).”

[Line 337, Page 16] “For Ru_{0.5}Ir_{0.5}O₂, generation of O₂ is the PDS with an overpotential of 0.36 V. Thus, our DFT simulations showed that the Ru_{0.5}Ir_{0.5}O₂ has the lowest OER overpotential ($\eta_{\text{OER}}=0.36$ V) compared to C-IrO₂ ($\eta_{\text{OER}}=1.00$ V) and C-RuO₂ ($\eta_{\text{OER}}=0.81$ V), coincident with our experimental observation that OER stability and catalytic activity of Ru_{0.5}Ir_{0.5}O₂ catalyst were significantly improved.”

Fig. 5 | DFT simulation findings of $\text{Ru}_{0.5}\text{Ir}_{0.5}\text{O}_2$. (a) Atomistic structure and E_{dft} of the $\text{Ru}_{0.5}\text{Ir}_{0.5}\text{O}_2$ (-331.4 eV). Atomistic structures of C-IrO₂ (b) and C-RuO₂ (c) (blue, Ru; grey, Ir; red, O). **d**, Schematic illustration of OER mechanism on the $\text{Ru}_{0.5}\text{Ir}_{0.5}\text{O}_2$ (blue, Ru; grey, Ir; red, O; white, H). **e**, The reaction paths on $\text{Ru}_{0.5}\text{Ir}_{0.5}\text{O}_2$ catalyst with the set potential of 0 and 1.23 V. The overpotential (η) is labeled for viewing convenience.

5) The estimated overpotentials for IrO₂ and RuO₂ are much higher than the experimental evidence. This fact has not been commented. There are several theoretical investigations of OER on rutile IrO₂ and RuO₂, but the authors did not compare their findings with previous reports.

[Author's Response]: Thank you for your comments regarding the overpotentials for IrO₂ and RuO₂ and the lack of comparison with previous theoretical investigations.

Indeed, our prediction of overpotentials are much higher than the experiment, but still within the range of reported DFT results as shown in **Supplementary Table 11**. After systematically comparing the available published results, we realize that most of the existing DFT calculations exhibit a systematic overestimation of the OER overpotentials for IrO₂ and RuO₂. These discrepancies can be reasonably attributed to the limitations in current DFT calculations. Despite the inherent uncertainties, the work published by Wen *et al.* (*J. Am. Chem. Soc.* 2021, **143**, 6482) and Zheng *et al.* (*Adv. Sci.* 2022, **9**, 2104636) reported the closest predicted overpotential compared with experiment, which indicates that the choice of active sites is of critical importance. As inspired by the reviewer, we have now included the edges of $\text{Ru}_{0.5}\text{Ir}_{0.5}\text{O}_2$ as active sites, and predict an overpotential of 0.36 V.

Supplementary Table 11 | The comparison of the OER performances (overpotential and estimated overpotential) of C-RuO₂, C-IrO₂ and various reported rutile IrO₂ and RuO₂ catalysts.

Electrocatalysts	Electrolyte	Overpotential ^a / mV	Estimated Overpotential /	Reference
------------------	-------------	------------------------------------	------------------------------	-----------

			mV	
C-RuO₂	0.5 M H₂SO₄	297	810	This work
RuO ₂	0.1 M HClO ₄	258	890	Nat. Commun. 2023 , 14, 354
RuO ₂	0.5 M H ₂ SO ₄	320	770	Nat. Commun. 2022 , 13, 3784
RuO ₂	0.5 M H ₂ SO ₄	285	560	J. Am. Chem. Soc. 2021 , 143, 6482-6490
RuO _x	0.5 M H ₂ SO ₄	281	840	Adv. Energy Mater. 2021 , 11, 2102883
RuO ₂	0.5 M H ₂ SO ₄	316	850	Nat. Commun. 2019 , 10, 3809
C-IrO₂	0.5 M H₂SO₄	321	1000	This work
IrO _x	0.5 M H ₂ SO ₄	334	1020	Adv. Energy Mater. 2021 , 11, 2102883
IrO ₂	0.1 M HClO ₄	406	680	Joule 2021 , 5, 1-14
IrO ₂	0.5 M H ₂ SO ₄	340	610	J. Am. Chem. Soc. 2021 , 143, 6482-6490
IrO ₂	0.5 M H ₂ SO ₄	297	590	Nat. Commun. 2021 , 12, 6007
IrO ₂	0.5M H ₂ SO ₄	397	580	Adv. Sci. 2022 , 9, 2104636

^a The overpotentials required to achieve a current density of 10 mA cm⁻².

We have modified the manuscript accordingly:

[Line 352, Page 16] “Moreover, most of the existing DFT calculations exhibit a systematic overestimation of the OER overpotentials for IrO₂ and RuO₂ (**Supplementary Table 11**). These discrepancies can be reasonably attributed to the limitations in current DFT calculations.”

Supplementary Table 11 | The comparison of the OER performances (overpotential and estimated overpotential) of C-RuO₂, C-IrO₂ and various reported rutile IrO₂ and RuO₂ catalysts.

Electrocatalysts	Electrolyte	Overpotential^a / mV	Estimated Overpotential / mV	Reference
C-RuO₂	0.5 M H₂SO₄	297	810	This work
RuO ₂	0.1 M HClO ₄	258	890	Nat. Commun. 2023 , 14, 354
RuO ₂	0.5 M H ₂ SO ₄	320	770	Nat. Commun. 2022 , 13, 3784
RuO ₂	0.5 M H ₂ SO ₄	285	560	J. Am. Chem. Soc. 2021 , 143, 6482-6490
RuO _x	0.5 M H ₂ SO ₄	281	840	Adv. Energy Mater. 2021 , 11, 2102883
RuO ₂	0.5 M H ₂ SO ₄	316	850	Nat. Commun. 2019 , 10, 3809
C-IrO₂	0.5 M H₂SO₄	321	1000	This work
IrO _x	0.5 M H ₂ SO ₄	334	1020	Adv. Energy Mater. 2021 , 11, 2102883
IrO ₂	0.2 M HClO ₄	406	680	Joule 2021 , 5, 1-14
IrO ₂	0.5 M H ₂ SO ₄	340	610	J. Am. Chem. Soc. 2021 , 143, 6482-6490
IrO ₂	0.5 M H ₂ SO ₄	297	590	Nat. Commun. 2021 , 12, 6007
IrO ₂	0.5M H ₂ SO ₄	397	580	Adv. Sci. 2022 , 9, 2104636

^a The overpotentials required to achieve a current density of 10 mA cm⁻².

6) The estimated overpotentials for Ru_{0.5}Ir_{0.5}O₂ is considerably higher than the experimental evidence, but the authors did not comment on this.

[Author's Response]: Thank you for pointing out our omission in discussing the discrepancy between the estimated overpotentials for Ru_{0.5}Ir_{0.5}O₂ and the experimental evidence. After systematically comparing the available published results (**Supplementary Table 10**), we realize that most of the existing DFT calculations overestimate the OER overpotentials of various reported catalysts. Among

the many factors affecting the overpotential predictions, we found the type of surface site is most important. Indeed, as inspired by the reviewer, we carried out simulation on edge site. These simulations yielded a considerably lower overpotential of 360 mV, which aligns much more closely with the experimentally observed overpotential of 151 mV. We believe that this now adequately addresses the discrepancy highlighted by the reviewer. Remaining differences can be attributed to various sources of uncertainty inherent in both theoretical and experimental methods.

Supplementary Table 10 | The comparison of the OER performances (overpotential and estimated overpotential) of Ru_{0.5}Ir_{0.5}O₂ and various reported catalysts.

Electrocatalysts	Electrolyte	Overpotential ^a / mV	Estimated Overpotential / mV	Reference
Ru_{0.5}Ir_{0.5}O₂	0.5 M H₂SO₄	151	360	This work
Re-RuO ₂	0.1 M HClO ₄	190	790	Nat. Commun. 2023 , 14 , 354
Li _{0.52} RuO ₂	0.5 M H ₂ SO ₄	156	510	Nat. Commun. 2022 , 13 , 3784
RuIrO _x	0.5 M H ₂ SO ₄	233	400	Nat. Commun. 2019 , 10 , 4875
CaCu ₃ Ru ₄ O ₁₂	0.5 M H ₂ SO ₄	171	660	Nat. Commun. 2019 , 10 , 3809
SrRuIr	0.5 M H ₂ SO ₄	190	370	J. Am. Chem. Soc. 2021 , 143 , 6482-6490
Ru ₁ Ir ₁ O _x	0.5 M H ₂ SO ₄	204	650	Adv. Energy Mater. 2021 , 11 , 2102883
3R-IrO ₂	0.1 M HClO ₄	188	550	Joule 2021 , 5 , 1-14
Ru ₁ -Pt ₃ Cu	0.1 M HClO ₄	220	420	Nat. Catal. 2019 , 2 , 304-313
W _{0.2} Er _{0.1} Ru _{0.7} O _{2-δ}	0.5 M H ₂ SO ₄	168	530	Nat. Commun. 2020 , 11 , 5368
Ru-N-C	0.5 M H ₂ SO ₄	267	590	Nat. Commun. 2019 , 10 , 4849

Cu-doped RuO ₂	0.5 M H ₂ SO ₄	188	660	Adv. Mater. 2018 , 30 , 1801351
RuO ₂ NSs	0.5 M H ₂ SO ₄	199	450	Energy Environ. Sci. 2020 , 13 , 5143-5151
RuO ₂ nanosheet	0.1 M HClO ₄	255	430	Adv. Energy Mater. 2019 , 9 , 1803795
Mn-RuO ₂	0.5 M H ₂ SO ₄	158	1480	ACS Catal. 2020 , 10 , 1152-1160

a: The overpotentials required to achieve a current density of 10 mA cm⁻².

We have modified the manuscript accordingly:

[Line 342, Page 16] “Indeed, our prediction of overpotentials of catalysts are higher than the experiment, but still within the range of reported DFT results as shown in **Supplementary Tables 10 and 11**. After systematically comparing the available published results (**Supplementary Table 10**), we realize that most of the existing DFT calculations overestimate the OER overpotentials of various reported catalysts. Among many factors affecting the overpotential predictions, we found the type of surface site is most important. We considered the edges of Ru_{0.5}Ir_{0.5}O₂ as active sites. These simulations yielded a considerably lower overpotential of 360 mV, which aligns much more closely with the experimentally observed overpotential of 151 mV. We believe that this now adequately explained the difference between the predicted overpotential of DFT calculation and the experiment results. Remaining differences can be attributed to various sources of uncertainty inherent in both theoretical and experimental methods.”

Supplementary Table 10 | The comparison of the OER performances (overpotential and estimated overpotential) of Ru_{0.5}Ir_{0.5}O₂ and various reported catalysts.

Electrocatalysts	Electrolyte	Overpotential ^a / mV	Estimated Overpotential / mV	Reference
Ru_{0.5}Ir_{0.5}O₂	0.5 M H₂SO₄	151	360	This work
Re-RuO ₂	0.1 M HClO ₄	190	790	Nat. Commun. 2023 , 14 , 354
Li _{0.52} RuO ₂	0.5 M H ₂ SO ₄	156	510	Nat. Commun. 2022 , 13 , 3784
RuIrO _x	0.5 M H ₂ SO ₄	233	400	Nat. Commun. 2019 , 10 , 4875
CaCu ₃ Ru ₄ O ₁₂	0.5 M H ₂ SO ₄	171	660	Nat. Commun. 2019 , 10 , 3809

SrRuIr	0.5 M H ₂ SO ₄	190	370	J. Am. Chem. Soc. 2021 , 143, 6482- 6490
Ru ₁ Ir ₁ O _x	0.5 M H ₂ SO ₄	204	650	Adv. Energy Mater. 2021 , 11, 2102883
3R-IrO ₂	0.1 M HClO ₄	188	550	Joule 2021 , 5, 1-14
Ru ₁ -Pt ₃ Cu	0.1 M HClO ₄	220	420	Nat. Catal. 2019 , 2, 304-313
W _{0.2} Er _{0.1} Ru _{0.7} O _{2-δ}	0.5 M H ₂ SO ₄	168	530	Nat. Commun. 2020 , 11, 5368
Ru-N-C	0.5 M H ₂ SO ₄	267	590	Nat. Commun. 2019 , 10, 4849
Cu-doped RuO ₂	0.5 M H ₂ SO ₄	188	660	Adv. Mater. 2018 , 30, 1801351
RuO ₂ NSs	0.5 M H ₂ SO ₄	199	450	Energy Environ. Sci. 2020 , 13, 5143-5151
RuO ₂ nanosheet	0.1 M HClO ₄	255	430	Adv. Energy Mater. 2019 , 9, 1803795
Mn-RuO ₂	0.5 M H ₂ SO ₄	158	1480	ACS Catal. 2020 , 10, 1152-1160

a: The overpotentials required to achieve a current density of 10 mA cm⁻².

Supplementary Table 11 | The comparison of the OER performances (overpotential and estimated overpotential) of C-RuO₂, C-IrO₂ and various reported rutile IrO₂ and RuO₂ catalysts.

Electrocatalysts	Electrolyte	Overpotential ^a / mV	Estimated Overpotential / mV	Reference
C-RuO₂	0.5 M H₂SO₄	297	810	This work
RuO ₂	0.1 M HClO ₄	258	890	Nat. Commun. 2023 , 14, 354
RuO ₂	0.5 M H ₂ SO ₄	320	770	Nat. Commun. 2022 , 13, 3784
RuO ₂	0.5 M H ₂ SO ₄	285	560	J. Am. Chem. Soc. 2021 , 143, 6482-6490

RuO _x	0.5 M H ₂ SO ₄	281	840	Adv. Energy Mater. 2021 , 11 , 2102883
RuO ₂	0.5 M H ₂ SO ₄	316	850	Nat. Commun. 2019 , 10 , 3809
C-IrO₂	0.5 M H₂SO₄	321	1000	This work
IrO _x	0.5 M H ₂ SO ₄	334	1020	Adv. Energy Mater. 2021 , 11 , 2102883
IrO ₂	0.3 M HClO ₄	406	680	Joule 2021 , 5 , 1-14
IrO ₂	0.5 M H ₂ SO ₄	340	610	J. Am. Chem. Soc. 2021 , 143 , 6482-6490
IrO ₂	0.5 M H ₂ SO ₄	297	590	Nat. Commun. 2021 , 12 , 6007
IrO ₂	0.5M H ₂ SO ₄	397	580	Adv. Sci. 2022 , 9 , 2104636

^a The overpotentials required to achieve a current density of 10 mA cm⁻².

Reviewer #2 (Remarks to the Author):

This paper reports the synthesis of a two-dimensional ruthenium-iridium oxide catalyst. The property of this material has been carefully characterized. And the catalytic performance shows a surprisingly low overpotential and the Faraday efficiency of oxygen has been measured. While there exist other papers on RuIr bimetallic oxide, to my knowledge, this manuscript reports the lowest overpotential (151 mV @ 10 mA cm⁻²), high turnover frequency (TOF) of 6.84 s⁻¹ at 1.44 V vs. RHE, and excellent stability (618.3 h operation) for oxygen evolving catalysts, especially operating at acidic electrolyte. The data and methods are effective and of high quality. The presentation was very clear. Specifically, I think the property characterization of this material is very thorough, and several complementary characterization technologies are provided. The results given in this manuscript are very interesting and can pave the way for the use of Ru_{0.5}Ir_{0.5}O₂ catalyst in acid water splitting. It turned out that the OER activity of Ru_{0.5}Ir_{0.5}O₂ is significantly higher than commercial RuO₂ and commercial IrO₂. and Ru_{0.5}Ir_{0.5}O₂ is much more stable than RuO₂ and IrO₂.

Based on the experiment and simulation, the authors attributed the enhanced catalytic activity of Ru_{0.5}Ir_{0.5}O₂ to more Ru active sites with high oxidation states generated at lower bias and increased surface oxidative charge concentrations. The strong interaction in the local structure of Ru-O-Ir also improves the stability of the Ru centers, preventing excessive oxidative dissolution of the active site. The advantage of this work is that it is not only reported as a superior catalyst but also provides in-depth mechanism analysis. As such the paper may be suitable for Nature Communications after minor version of the manuscript. Some specific comments are listed below.

[Author's Response]: We would like to thank you for the positive comments and recommendation of the publication in the Nature Communications. Your comments lead to further improve the quality of our work. We have modified our manuscript according to your valuable comments.

1. The author claimed that the electrochemical stability is very good. But the chronopotentiometric measurements are not very enough to evaluate the electrochemical stability at harsh conditions. Please provide the ADT-CV test for anodic reaction. And please explore the Tafel slope of Ru_{0.5}Ir_{0.5}O₂ catalyst before and after the ADT-CV testing. As an alternative, the author can also test the stability in the form of simple electrolyzer if possible.

[Author's Response]: Thank you very much. According to your suggestion, we measured the ADT-CV test of Ru_{0.5}Ir_{0.5}O₂. **Supplementary Fig. 24** shows that the overpotential of Ru_{0.5}Ir_{0.5}O₂ after 1000 CV cycles at the current density of 10 mA cm⁻² increases by 11 mV, indicating that Ru_{0.5}Ir_{0.5}O₂ has good stability. The Tafel slopes of Ru_{0.5}Ir_{0.5}O₂ before and after the ADT-CV testing was calculated. The Tafel slopes of Ru_{0.5}Ir_{0.5}O₂ are 44 mV dec⁻¹ and 49 mV dec⁻¹ before and after ADT-CV testing (**Supplementary Fig. 24c**).

As suggested, we applied Ru_{0.5}Ir_{0.5}O₂ (with a mass loading of 1.0 mg cm⁻²) as an anode catalyst in acidic PEM electrolyser (0.5 M H₂SO₄) at room temperature (**Supplementary Fig. 26**). And the PEM electrolyzers using Ru_{0.5}Ir_{0.5}O₂ catalyst can achieve current densities of ~200 mA cm⁻² at least 255 hours without significant performance degradation.

Supplementary Fig. 24 | ADT-CV test of $\text{Ru}_{0.5}\text{Ir}_{0.5}\text{O}_2$. **a**, Cyclic voltammogram curves of $\text{Ru}_{0.5}\text{Ir}_{0.5}\text{O}_2$ in a potential region of 0.25 ~ 1.3 V vs. RHE with the scan rate of 100 mV s^{-1} . **b**, The OER polarization curves of $\text{Ru}_{0.5}\text{Ir}_{0.5}\text{O}_2$ before and after 1000 CV cycles. **c**, The Tafel slopes of $\text{Ru}_{0.5}\text{Ir}_{0.5}\text{O}_2$ before and after the ADT-CV testing.

Supplementary Fig. 26 | PEM device performance. Chronoamperometry stability test of $\text{Ru}_{0.5}\text{Ir}_{0.5}\text{O}_2$ catalyst (with a mass loading of 1.0 mg cm^{-2}) in acidic PEM electrolyser ($0.5 \text{ M H}_2\text{SO}_4$) at room temperature. And the commercial Pt/C (with a mass loading of 0.7 mg cm^{-2}) was used as the cathode catalyst.

We have modified the manuscript accordingly:

[Line 215, Page 10] “The stability of $\text{Ru}_{0.5}\text{Ir}_{0.5}\text{O}_2$ was further demonstrated by the accelerated durability test-cyclic voltammetry (ADT-CV). **Supplementary Fig. 24a** showed the cyclic voltammogram (CV) test of $\text{Ru}_{0.5}\text{Ir}_{0.5}\text{O}_2$ for 1000 cycles with a scan rate of 100 mV s^{-1} . **Supplementary Fig. 24b** showed the OER polarization curves of $\text{Ru}_{0.5}\text{Ir}_{0.5}\text{O}_2$ before and after 1000 cycles. After the ADT test, the overpotential ($\eta @ 10 \text{ mA cm}^{-2}$) of $\text{Ru}_{0.5}\text{Ir}_{0.5}\text{O}_2$ was increased by 11 mV (**Supplementary Fig. 24b**) and The Tafel slopes of $\text{Ru}_{0.5}\text{Ir}_{0.5}\text{O}_2$ before and after ADT are 44 mV dec^{-1}

¹ and 49 mV dec⁻¹ respectively (**Supplementary Fig. 24c**), indicating that Ru_{0.5}Ir_{0.5}O₂ has good stability.”

[Line 224, Page 11] In addition, Ru_{0.5}Ir_{0.5}O₂ (mass loading ~1.0 mg cm⁻²) was used as an anode catalyst in acidic PEM electrolyte (0.5 M H₂SO₄) at room temperature (**Supplementary Fig. 26**). The PEM electrolyzers can achieve current densities of ~200 mA cm⁻² at least 255 hours by using Ru_{0.5}Ir_{0.5}O₂ catalyst, and the performance of PEM electrolyzer without significant performance degradation.”

Supplementary Fig. 24 | ADT-CV test of Ru_{0.5}Ir_{0.5}O₂. **a**, Cyclic voltammogram curves of Ru_{0.5}Ir_{0.5}O₂ in a potential region of 0.25 ~ 1.3 V vs. RHE with the scan rate of 100 mV s⁻¹. **b**, The OER polarization curves of Ru_{0.5}Ir_{0.5}O₂ before and after 1000 CV cycles. **c**, The Tafel slopes of Ru_{0.5}Ir_{0.5}O₂ before and after the ADT-CV testing.

Supplementary Fig. 26 | PEM device performance. Chronoamperometry stability test of Ru_{0.5}Ir_{0.5}O₂ catalyst (with a mass loading of 1.0 mg cm⁻²) in acidic PEM electrolyser (0.5 M H₂SO₄) at room temperature. And the commercial Pt/C (with a mass loading of 0.7 mg cm⁻²) was used as the cathode catalyst.

2. During the anodic reaction, metal oxides commonly become an amorphous-phase. Please clarify whether the crystallinity of Ru_{0.5}Ir_{0.5}O₂ changes after the ADT-CV test.

[Author's Response]: Thank you for this important suggestion. We characterized Ru_{0.5}Ir_{0.5}O₂ after the ADT-CV testing. The Bragg diffraction peaks in the XRD pattern of Ru_{0.5}Ir_{0.5}O₂ were basically unchanged, and no other peaks corresponding to rutile C-IrO₂ and rutile C-RuO₂ were observed (**Supplementary Fig. 25a**). And the **Supplementary Fig. 25b** shows the original two-dimensional sheet structure of Ru_{0.5}Ir_{0.5}O₂ basically remained after ADT-CV test. Both the SAED pattern (along the [0001] zone axes, **Supplementary Fig. 25c**) and the HRTEM image (**Supplementary Fig. 25d**) illustrate that there is no amorphous area in Ru_{0.5}Ir_{0.5}O₂ catalyst. Furthermore, the STEM-EDX element mapping (**Supplementary Fig. 25e**) shows the uniform distribution of Ir and Ru elements. And the TEM-EDX shows the Ru, Ir and O elements of Ru_{0.5}Ir_{0.5}O₂ after ADT-CV test (**Supplementary Fig. 25f**).

Supplementary Fig. 25 | Characterizations of Ru_{0.5}Ir_{0.5}O₂ after ADT-CV testing. **a**, XRD pattern of Ru_{0.5}Ir_{0.5}O₂ after ADT-CV testing. **b**, TEM image of Ru_{0.5}Ir_{0.5}O₂, **c**, the SAED pattern and **d**, HRTEM image of Ru_{0.5}Ir_{0.5}O₂ after the ADT-CV testing. **e**, STEM-EDX mapping of Ru_{0.5}Ir_{0.5}O₂ and **f**, TEM-EDX spectrum of Ru_{0.5}Ir_{0.5}O₂ after ADT-CV testing.

We have modified the manuscript accordingly:

[Line 221, Page 10] “And the characterization of $\text{Ru}_{0.5}\text{Ir}_{0.5}\text{O}_2$ after 1000 cycles of ADT-CV showed that the change in crystallinity of $\text{Ru}_{0.5}\text{Ir}_{0.5}\text{O}_2$ before and after ADT was negligible (Supplementary Fig. 25).”

Supplementary Fig. 25 | Characterizations of $\text{Ru}_{0.5}\text{Ir}_{0.5}\text{O}_2$ after ADT-CV testing. **a**, XRD pattern of $\text{Ru}_{0.5}\text{Ir}_{0.5}\text{O}_2$ after ADT-CV testing. **b**, TEM image of $\text{Ru}_{0.5}\text{Ir}_{0.5}\text{O}_2$, **c**, the SAED pattern and **d**, HRTEM image of $\text{Ru}_{0.5}\text{Ir}_{0.5}\text{O}_2$ after the ADT-CV testing. **e**, STEM-EDX mapping of $\text{Ru}_{0.5}\text{Ir}_{0.5}\text{O}_2$ and **f**, TEM-EDX spectrum of $\text{Ru}_{0.5}\text{Ir}_{0.5}\text{O}_2$ after ADT-CV testing.

3. The ECSA is also an important factor to evaluate the intrinsic OER catalytic activity. The ECSA

should be tested by Cdl. And the LSV curves normalized by ECSA should be added in this manuscript. [Author's Response]: Following your valuable suggestion, as shown in **Supplementary Fig. 17** and **Supplementary Table. 5**, we have added the electrochemical double-layer capacitance (C_{dl}) test and the electrochemically active surface area (ECSA) was calculated by C_{dl} . The OER activity normalized to the ECSA of the catalysts were also calculated (**Supplementary Fig. 18**), suggesting intrinsically improved OER activity on the $\text{Ru}_{0.5}\text{Ir}_{0.5}\text{O}_2$.

Supplementary Fig. 17 | ECSA characterizations of catalysts. CV curves of (a) $\text{Ru}_{0.5}\text{Ir}_{0.5}\text{O}_2$, (b) C-IrO_2 and (c) C-RuO_2 catalysts in the non-Faradaic region with the scan rates of 5, 10, 15, 20, 25, 30, and 35 mV s^{-1} in 0.5 M H_2SO_4 electrolyte. **d**, C_{dl} plots obtained from CV curves.

Supplementary Fig. 18 | OER performance. Normalized LSV curves to electrochemically active surface area (ECSA) of $\text{Ru}_{0.5}\text{Ir}_{0.5}\text{O}_2$, C-IrO_2 and C-RuO_2 catalysts.

Supplementary Table 5 | The ECSA of $\text{Ru}_{0.5}\text{Ir}_{0.5}\text{O}_2$, C-IrO_2 and C-RuO_2 in this work.

Catalysts	ECSA / cm^2
-----------	----------------------

$\text{Ru}_{0.5}\text{Ir}_{0.5}\text{O}_2$	113.7
C-IrO ₂	23.4
C-RuO ₂	39.4

We have modified the manuscript accordingly:

[Line 171, Page 8] “To better understand the origin of the high OER performance of $\text{Ru}_{0.5}\text{Ir}_{0.5}\text{O}_2$, we explored the electrochemical double-layer capacitance (C_{dl}) test and the electrochemically active surface area (ECSA) was calculated by C_{dl} for activity normalization (**Supplementary Figs. 17, 18** and **Supplementary Fig. Table 5**). The prepared $\text{Ru}_{0.5}\text{Ir}_{0.5}\text{O}_2$ with typical two-dimensional (2D) sheet-like shape structure exhibited higher C_{dl} and ECSA than those of C-IrO₂ and C-RuO₂ (**Supplementary Fig. 17** and **Supplementary Table 5**), suggesting that 2D shape can significantly improve the density of active sites. The OER activity normalized to the ECSA and BET of the catalysts were also calculated (**Supplementary Figs. 18, 19**). The specific activity of $\text{Ru}_{0.5}\text{Ir}_{0.5}\text{O}_2$ was better than those of all the other samples. These results show that $\text{Ru}_{0.5}\text{Ir}_{0.5}\text{O}_2$ has superior intrinsic OER catalytic activity.”

Supplementary Fig. 17 | ECSA characterizations of catalysts. CV curves of (a) $\text{Ru}_{0.5}\text{Ir}_{0.5}\text{O}_2$, (b) C-IrO₂ and (c) C-RuO₂ catalysts in the non-Faradaic region with the scan rates of 5, 10, 15, 20, 25, 30, and 35 mV s⁻¹ in 0.5 M H₂SO₄ electrolyte. **d**, C_{dl} plots obtained from CV curves.

Supplementary Fig. 18 | OER performance. Normalized LSV curves to electrochemically active surface area (ECSA) of Ru_{0.5}Ir_{0.5}O₂, C-IrO₂ and C-RuO₂ catalysts.

Supplementary Fig. 19 | OER performance. Normalized LSV curves to BET-based areas of Ru_{0.5}Ir_{0.5}O₂, C-IrO₂ and C-RuO₂.

Supplementary Table 5 | The ECSA of Ru_{0.5}Ir_{0.5}O₂, C-IrO₂ and C-RuO₂ in this work.

Catalysts	ECSA / cm ²
Ru _{0.5} Ir _{0.5} O ₂	113.7
C-IrO ₂	23.4
C-RuO ₂	39.4

4. It is respectfully recommended that the authors consider including further information pertaining to the simulation model in the manuscript. While it is noted that the combination of Ru and Ir resulted in a substantial improvement in performance, it remains unclear how the simulation accurately mimics the experimental structure in order to allow for a thorough comparison. Although Figure 5 provides a visual representation of the structure, the current description of the simulation model does not provide sufficient detail to enable the reader to fully comprehend the model. Consequently, it is suggested that the authors supplement their explanation of the simulation model with additional information to improve the manuscript's clarity and comprehensibility.

[Author's Response]: Thank you for your valuable recommendation to include further information about the simulation model in our manuscript. We agree with the reviewer in the importance of

providing a clear and detailed description to enable a full comprehension of the model and its relation to the experimental structure. We have now included the details of the model building as follows:

In our DFT models for $\text{Ru}_{0.5}\text{Ir}_{0.5}\text{O}_2$, we approached the problem by expanding the 2D IrO_2 which is obtained from XRD by 4 times along the x and y direction, and substituting Ir atoms with Ru atoms to reach an atomic ratio of $\text{Ru}/\text{Ir} = 1:1$, which aligns closely with the experimental value, we calculated some different structures $\text{Ru}_{0.5}\text{Ir}_{0.5}\text{O}_2$ (01-10) surface, and pick the most stable structure as our calculated model.

We have added above details in revised **Supplementary Information**.

5. Since the low overpotential is also the key figure in this paper, the correctness of this value is very important. I can't find statistics on how many samples have been tested. Because of the potentially high significance of the results reported in this manuscript Statistical data of 3-5 samples should be reported to consolidate this strong claim. It will greatly enhance the reliability and reproducibility of the reported data.

[Author's Response]: Thank you for your valuable suggestion. we add the OER results of $\text{Ru}_{0.5}\text{Ir}_{0.5}\text{O}_2$ by measuring 3 different samples. **Supplementary Fig. 14** shows that the high reproducibility of OER performance of $\text{Ru}_{0.5}\text{Ir}_{0.5}\text{O}_2$ samples. The results show that the OER performance of $\text{Ru}_{0.5}\text{Ir}_{0.5}\text{O}_2$ is repeatable and stable.

Supplementary Fig. 14 | Reproducibility of linear sweep voltammetry (LSV) scans for $\text{Ru}_{0.5}\text{Ir}_{0.5}\text{O}_2$ catalyst. The OER polarization curves of $\text{Ru}_{0.5}\text{Ir}_{0.5}\text{O}_2$ in O_2 -saturated 0.5 M H_2SO_4 electrolyte. Measured on GCE with a mass loading of $283 \mu\text{g cm}^{-2}$. Scan rate: 5 mV s^{-1} .

We have modified the manuscript accordingly:

[Line 153, Page 7] “In addition, **Supplementary Fig. 14** shows that the OER performance achieved by $\text{Ru}_{0.5}\text{Ir}_{0.5}\text{O}_2$ has good reproducibility.”

Supplementary Fig. 14 | Reproducibility of linear sweep voltammetry (LSV) scans for Ru_{0.5}Ir_{0.5}O₂ catalyst. The OER polarization curves of Ru_{0.5}Ir_{0.5}O₂ in O₂-saturated 0.5 M H₂SO₄ electrolyte. Measured on GCE with a mass loading of 283 μg cm⁻². Scan rate: 5 mV s⁻¹.

6. I suggest the authors add more details about the simulation methods. For example, the solvation effect is known to have a significant impact on predictions. Indeed, I notice the authors mentioned the solvation calculations. However, it is not clear to me what method and how did the authors carry out the simulation. Therefore, I suggest the author add more details and the calculated data.

[Author's Response]: Thank you for your suggestion to provide more details about our simulation methods, specifically regarding the solvation effect. We recognize the importance of adequately addressing the solvation effect, as it can indeed have a significant impact on predictions. We apologize for the lack of clarity in our initial submission and appreciate the opportunity to address this issue.

In the revised Supporting Information, we have provided a more comprehensive description of the solvation calculations.

The solvation effect is considered with a linearized Poisson-Boltzmann implicit solvation model as implemented in VASP module of VASP_{sol}, which is freely distributed as an open-source package and hosted on GitHub (<https://github.com/henniggroup/VASPsol>). We consider an electrolyte that consists of an aqueous solution of monovalent anions and cations of 1M concentration. At room temperature, this electrolyte has a relative permittivity of $\epsilon_b = 78.4$ and a Debye length of $\lambda_D = 3.04 \text{ \AA}$. The parameters for the shape function are $n_c = 0.0025 \text{ \AA}^{-1}$ and $\sigma = 0.6$.

We have added above details in revised **Supplementary Information**.

7. As shown in Figure 3b, to exclude the effect of diffusion, the Tafel slope should be calculated by current density as low as possible.

[Author's Response]: Thank you for your valuable suggestion. The Tafel slopes were calculated by a low current density less than 10 mA cm⁻² in this work.

8. There are two common mechanisms including lattice oxygen and adsorbate evolving mechanism, which one is more suitable for the Ru_{0.5}Ir_{0.5}O₂ catalyst system.

[Author's Response]: Thank you for your suggestion. In this work, the adsorbate evolving mechanism (AEM) was more suitable for the Ru_{0.5}Ir_{0.5}O₂ catalyst system [Wen, Y. *et al.* Stabilizing highly active Ru sites by suppressing lattice oxygen participation in acidic water oxidation. *J. Am. Chem. Soc.* **143**, 6482-6490 (2021)], and the AEM was used to explain the origin of high activity in the Ru_{0.5}Ir_{0.5}O₂ catalyst by the theoretical calculation.

Reviewer #3 (Remarks to the Author):

In “Stable and oxidative charged Ru enhance the oxygen evolution reaction activity in two-dimensional ruthenium-iridium oxide” Zhu, Kang and co-authors describe the synthesis and application of a binary Ru_{0.5}Ir_{0.5}O₂ layered oxide and application as an acid-stable water oxidation catalyst. The paper's main arguments are the following: 1) The origin of the high OER activity of Ru_{0.5}Ir_{0.5}O₂ nanosheets are Ru active sites that are more easily oxidized owing to Ir alloying, and 2) Ir incorporation increases the stability of these Ru active sites through strong Ru-O-Ir bonding. These conclusions are based on electron microscopy, X-ray spectroscopy (XPS, XAS), pulse voltage induced current, and DFT calculations. The OER activity and stability are impressive but it is unclear whether the performance alone is technologically relevant given the high Ir content. I'm also not sure, in its current state, that it distinguishes itself enough from previous work (Refs 13, 29). Overall, I think the manuscript needs significant changes, particularly in the computational section, and there are a number of experimental technical errors in the paper that need to be addressed before the manuscript can be considered acceptable for publication. Detailed comments on outstanding issues in the manuscript are provided below:

[Author's Response]: Thank you for your valuable comments. The constructive suggestions lead to further improvement of the quality for our work.

1. We first systematically summarize the differences between this work (Ru_{0.5}Ir_{0.5}O₂ catalyst) and previous published work [SrRuIr, Refs 13: Wen, Y. *et al.* Stabilizing highly active Ru sites by suppressing lattice oxygen participation in acidic water oxidation. *J. Am. Chem. Soc.* **143**, 6482-6490 (2021); 1T-IrO₂, Refs 29: Dang, Q. *et al.* Iridium metallene oxide for acidic oxygen evolution catalysis. *Nat. Commun.* **12**, 6007 (2021)]. As shown in **Table R1**, Ru_{0.5}Ir_{0.5}O₂ is clearly different from the previously reported materials in the synthetic routes.
2. Wen, Y. *et al.* (SrRuIr) mainly explores the electronic structure of active Ru sites in SrRuIr catalyst was modulated by Sr and Ir, optimizing the binding energetics of OER oxo-intermediates and reveals that the participation of lattice oxygen during OER was suppressed by interactions in the Ru-O-Ir local structure; And Dang, Q. *et al.* (1T-IrO₂) mainly explores 1 T phase-iridium dioxide (1T-IrO₂) was prepared by a synthetic strategy combining mechanochemistry and thermal treatment in a strong alkaline medium. The 1T-phase Ir active sites and ultra-thin two-dimensional structure enable 1T-IrO₂ to achieve high performance acidic OER catalysis. However, the main advantages in this work are the following: 1) The origin of the high OER activity of Ru_{0.5}Ir_{0.5}O₂ nanosheets are Ru active sites that are more easily oxidized owing to Ir alloying, and 2) Ir incorporation increases the stability of these Ru active sites through strong Ru-O-Ir bonding. Therefore, the focus of the research discussed in this work is clearly different from the previously reported materials.
3. In terms of OER catalytic activity, the Ru_{0.5}Ir_{0.5}O₂ catalyst shows high OER activity in acid with a lowest overpotential of 151 mV at 10 mA cm⁻², a highest turnover frequency (TOF) of 6.84 s⁻¹ at 1.44 V vs. RHE and outstanding stability (618.3 h operation). **Table R2** shows the OER performance of Ru_{0.5}Ir_{0.5}O₂ is better than that of the SrRuIr and 1T-IrO₂ catalysts. To sum up, the Ru_{0.5}Ir_{0.5}O₂ catalyst distinguishes itself enough from previous work both in terms of research focus and catalytic performance.
4. The Ru_{0.5}Ir_{0.5}O₂ catalytic activity is much better than 1T-IrO₂, indicating the performance alone is not technologically relevant to Ir contents in this work.

Table R1. The comparison of the synthetic method and crystal structure of Ru_{0.5}Ir_{0.5}O₂ and the previously reported catalysts.

Materials	Crystal structure	Unit-cell parameters	Synthetic method	Reference
-----------	-------------------	----------------------	------------------	-----------

Ru _{0.5} Ir _{0.5} O ₂	Trigonal crystal system, 1T structure	$a = b = 3.00 \text{ \AA}$, $c = 6.95 \text{ \AA}$; $\alpha = \beta = 90^\circ$, $\gamma = 120^\circ$	Two-step molten-alkali method	This work
1T-IrO ₂	Trigonal crystal system, 1T structure	$a = b = 3.11 \text{ \AA}$ and $c = 6.91 \text{ \AA}$; $\alpha = \beta = 90^\circ$, $\gamma = 120^\circ$	One-step molten salt template method	Nat. Commun. 2021,12, 6007
SrRuIr	Rutile phase, tetragonal structure	$a = b = 4.50 \text{ \AA}$ and $c = 3.11 \text{ \AA}$; $\alpha = \beta = \gamma = 90^\circ$	Sol-gel method	J. Am. Chem. Soc. 2021,143, 6482-6490

Table R2. The comparison of the OER performances of Ru_{0.5}Ir_{0.5}O₂ and previously reported catalysts.

Material	Electrolyte	Over-potential ^a (mV)	TOF (s ⁻¹)	Mass activity (A g _{Ru + Ir} ⁻¹)	Stability ^b (h)	Reference
Ru _{0.5} Ir _{0.5} O ₂	0.5 M H ₂ SO ₄	151	6.84 @ 1.44 V vs. RHE	730.4 @ 1.44 V vs. RHE	618.3	This work
SrRuIr	0.5 M H ₂ SO ₄	190	0.2 @ 1.53 V vs. RHE	654 @ 1.53 V vs. RHE	1500	J. Am. Chem. Soc. 2021, 143, 6482-6490
1T-IrO ₂	0.1 M HClO ₄	197	4.2 @ 1.5 V vs. RHE	296.8 @ 1.5 V vs. RHE	40	Nat. Commun. 2021, 12, 6007

^a The overpotentials required to achieve a current density of 10 mA cm⁻²;

^b Chronopotentiometric stability test at the current density of 10 mA cm⁻².

1. In the DFT structure, Ru_{0.5}Ir_{0.5}O₂ is represented with edge-sharing octahedra, while the TEM measurements in Figure 1e suggest that the structure is corner sharing with extremely large Ir-Ru spacing (the 100 spacing is on the order of 0.5 nm)? Interestingly, the (10-10) reflection at ~35 degrees in the XRD pattern suggests a spacing of ~0.25 nm. Are the scale bars in the HAADF-STEM images in Figure 1e labeled incorrectly? And how do the authors rationalize the corner-sharing octahedra structure shown? As the structure is shown in the TEM images, the stoichiometry of the sample should be closer to Ru_{0.5}Ir_{0.5}O₃ which suggests oxidation states of the metals near 6+, something that is not supported by the XANES measurements. Comparing Figure 1d to Figure 1e, it is clear that one of the scale bars is labeled incorrectly.

[Author's Response]: Thank you for your insightful comments. You bring up several important points, and we appreciate your careful review of our data.

- The scale bar in **Fig. 1e** in the previous version was wrong due to our carelessness. Now, the bar is corrected in **Fig. 1e** in the revised manuscript. The original digital photo of aberration-corrected HAADF-STEM image of the Ru_{0.5}Ir_{0.5}O₂ catalyst was added (**Fig. R1**). As shown in **Fig. R2**, we have modified the scale bar of the aberration-corrected HAADF-STEM image of the Ru_{0.5}Ir_{0.5}O₂ catalyst, and the distance of {10-10} plane of Ru_{0.5}Ir_{0.5}O₂ was 0.260 nm.

Meanwhile, the diffraction peak at 34.5° corresponds to the $\{10\text{-}10\}$ plane of $\text{Ru}_{0.5}\text{Ir}_{0.5}\text{O}_2$ in the XRD pattern, indicating a distance of 0.260 nm. This result reconfirms the accuracy of aberration-corrected HAADF-STEM image of the $\text{Ru}_{0.5}\text{Ir}_{0.5}\text{O}_2$ catalyst.

2. The XANES measurements of $\text{Ru}_{0.5}\text{Ir}_{0.5}\text{O}_2$ show that the valence state of Ir is higher than +4 and valence state of Ru is between 0 and +4. The EXAFS results indicate that the polyhedrons of Ir-O and Ru-O are octahedra. And the distance between metal atoms is determined to be 0.30 nm on the base of the aberration-corrected HAADF-STEM images (**Fig. 1e**). All the above data suggested that the $\text{Ru}_{0.5}\text{Ir}_{0.5}\text{O}_2$ is represented with edge-sharing octahedra (**Fig. R3**). Thus, it is plausible to employ the edge-sharing octahedra $\text{Ru}_{0.5}\text{Ir}_{0.5}\text{O}_2$ in the DFT calculations.
3. In order to understand the structure of the $\text{Ru}_{0.5}\text{Ir}_{0.5}\text{O}_2$ more intuitively, $\text{Ru}_{0.5}\text{Ir}_{0.5}\text{O}_2$ crystal were modelled as shown in **Fig. 1f**, where the blue, grey and red spheres schematically represent the arrangement of Ru, Ir and O atoms of $\text{Ru}_{0.5}\text{Ir}_{0.5}\text{O}_2$.

Figure R1. The original data of aberration-corrected HAADF-STEM image of the $\text{Ru}_{0.5}\text{Ir}_{0.5}\text{O}_2$ catalyst. The scale bar in the bottom left corner of the image is 2 nm.

Figure R2. a, Aberration-corrected HAADF-STEM image of the $\text{Ru}_{0.5}\text{Ir}_{0.5}\text{O}_2$ catalyst. b, The blue dashed box inset in a shows the corresponding distance of $\{10\text{-}10\}$ in $\text{Ru}_{0.5}\text{Ir}_{0.5}\text{O}_2$ catalyst.

Figure R3. The schematic atom structure of $\text{Ru}_{0.5}\text{Ir}_{0.5}\text{O}_2$, the Ru, Ir, O atoms are represented by blue,

grey and red spheres.

Fig. 1 | Structural and phase characterizations of $\text{Ru}_{0.5}\text{Ir}_{0.5}\text{O}_2$. **a**, SEM and **b**, TEM images of $\text{Ru}_{0.5}\text{Ir}_{0.5}\text{O}_2$. **b1**, The SAED pattern of $\text{Ru}_{0.5}\text{Ir}_{0.5}\text{O}_2$, where the hexagonal pattern shows the [0001] projection. **c**, XRD pattern of $\text{Ru}_{0.5}\text{Ir}_{0.5}\text{O}_2$. The inset is a larger view of the marked area in **c**. **d**, The aberration-corrected HAADF-STEM image for $\text{Ru}_{0.5}\text{Ir}_{0.5}\text{O}_2$. **e**, High-magnification image of the region in **d**. **f**, The schematic atom structure of $\text{Ru}_{0.5}\text{Ir}_{0.5}\text{O}_2$, the Ru, Ir, O atoms are represented by blue, grey and red spheres.

We have modified the manuscript accordingly:

[Line 99, Page 5] “**Fig. 1e** (the high-magnification image of the region in **Fig. 1d**) reveals that the ordered arranged metal atoms have different brightness, arising from a random and even distribution of Ir and Ru atoms of different brightness, which is also consistent with the substitutional solid solution structure of $\text{Ru}_{0.5}\text{Ir}_{0.5}\text{O}_2$. $\text{Ru}_{0.5}\text{Ir}_{0.5}\text{O}_2$ crystal were modelled as shown in **Fig. 1f**, where the blue, grey and red spheres schematically represent the arrangement of Ru, Ir and O atoms of $\text{Ru}_{0.5}\text{Ir}_{0.5}\text{O}_2$.”

Fig. 1 | Structural and phase characterizations of $\text{Ru}_{0.5}\text{Ir}_{0.5}\text{O}_2$. **a**, SEM and **b**, TEM images of $\text{Ru}_{0.5}\text{Ir}_{0.5}\text{O}_2$. **b1**, The SAED pattern of $\text{Ru}_{0.5}\text{Ir}_{0.5}\text{O}_2$, where the hexagonal pattern shows the [0001] projection. **c**, XRD pattern of $\text{Ru}_{0.5}\text{Ir}_{0.5}\text{O}_2$. The inset is a larger view of the marked area in **c**. **d**, The aberration-corrected HAADF-STEM image for $\text{Ru}_{0.5}\text{Ir}_{0.5}\text{O}_2$. **e**, High-magnification image of the region in **d**. **f**, The schematic atom structure of $\text{Ru}_{0.5}\text{Ir}_{0.5}\text{O}_2$, the Ru, Ir, O atoms are represented by blue, grey and red spheres.

2. The authors use double distilled water to prepare their electrolytes. This water source cannot be considered ultrapure and generally contains residual impurities. It is well known that electrolyte impurities, even at ultralow concentrations, can dramatically influence the performance of OER electrocatalysts. Given the 5-10x larger surface area of the $\text{Ru}_{0.5}\text{Ir}_{0.5}\text{O}_2$ material, it is more probable that this material can adsorb these impurities. What is measured resistance of the double distilled water and what residual impurities are present in it? How do the authors rule out the influence of these impurities. Overall, I think it is imperative that fundamental OER studies, such as this one, use ultrapure Type I water for preparation of their electrolytes. Without this, there is a significant degree of uncertainty whether the results accurately reflect the intrinsic activity of $\text{Ru}_{0.5}\text{Ir}_{0.5}\text{O}_2$.

[Author's Response]: Thank you for your comments.

1. The water source of the double distilled water we used is deionized water produced by commercial companies. Moreover, we detected the residual impurities in double distilled water, and used ICP-MS to find the possible impurities in double distilled water, especially the residual impurities that have an effect on OER. As shown in **Table R3**, the double distilled water we used for the experimental test did not contain any residual impurities that could affect OER. And the measurement resistivity of double distilled water at 298 K is 17 MΩ cm.
2. In order to further exclude the influence of solvents (double distilled water) in our experimental tests, we also tested the OER performance of the $\text{Ru}_{0.5}\text{Ir}_{0.5}\text{O}_2$ catalyst in an electrolyte formulated with ultrapure water. The ultrapure water (18.2 MΩ cm) was prepared by using laboratory water purification system (EPED, model no. Plus-E2-20TJ). **Fig. R4** shows almost no difference in OER performance tested by $\text{Ru}_{0.5}\text{Ir}_{0.5}\text{O}_2$ catalysts in different O_2 -saturated 0.5 M H_2SO_4 electrolytes prepared by double distilled water and ultrapure water, which also shows that the effect of double distilled water on OER performance in this work is almost negligible.

Table R3. ICP-MS analyses of the double distilled water used in the experiment.

Element	Content / at. %
Mg	Below limit of detection (LOD)
Ca	Below LOD
Al	Below LOD
Ru	Below LOD
Ir	Below LOD
Mn	Below LOD
Fe	Below LOD
Co	Below LOD
Ni	Below LOD

Figure R4. The comparison of OER performances for $\text{Ru}_{0.5}\text{Ir}_{0.5}\text{O}_2$ catalyst in electrolytes formulated with different aqueous solvents. The OER polarization curves of $\text{Ru}_{0.5}\text{Ir}_{0.5}\text{O}_2$ were measured in O_2 -saturated 0.5 M H_2SO_4 electrolytes prepared by double distilled water and ultrapure water, respectively. Measured on GCE with a mass loading of $283 \mu\text{g cm}^{-2}$. Scan rate: 5 mV s^{-1} .

3. The XPS and XAS measurements seem to contradict each other. The XPS suggests Ir is more oxidized in $\text{Ru}_{0.5}\text{Ir}_{0.5}\text{O}_2$ and Ru is less oxidized as compared to RuO_2 and IrO_2 (assuming these have a $4+$ oxidation state) whereas the Ir whiteline is lower in energy in $\text{Ru}_{0.5}\text{Ir}_{0.5}\text{O}_2$ and the Ir-O bond length is longer in the EXAFS data (suggesting it is reduced compared to IrO_2) and the Ru-O bond is shorter in the EXAFS data (suggesting higher oxidation state). Additionally, Ref 13 shows the opposite trend in observed bond lengths. The authors should discuss this discrepancy and provide a possible explanation. Do the bond lengths in the converged DFT structures of $\text{Ru}_{0.5}\text{Ir}_{0.5}\text{O}_2$, RuO_2 and IrO_2 agree with the EXAFS analysis? Do the magnetic moments/bader charges in the DFT analysis also suggest an oxidation state assignment of $>4+$?

[Author's Response]: Thank you for pointing out the problems of XPS and XAS measurement results in our study. We apologize for not providing a clear explanation for these problems in the manuscript.

1. In this work, the XPS results suggest that $\text{Ru}_{0.5}\text{Ir}_{0.5}\text{O}_2$ exhibits a lower valence state of Ru sites than that in C- RuO_2 (Ru^{4+}) and a higher Ir valence state than that in C- IrO_2 (Ir^{4+}). The Ir L_{3-} edge XANES spectra of C- IrO_2 (Fig. 2d) was corrected in the new version, showing that the white line peak intensity of $\text{Ru}_{0.5}\text{Ir}_{0.5}\text{O}_2$ exhibits significantly higher than that of C- IrO_2 and Ir foil, demonstrating that the valence states of Ir show an order of $\text{Ru}_{0.5}\text{Ir}_{0.5}\text{O}_2 > \text{C-}\text{IrO}_2 > \text{Ir foil}$. Meanwhile, we once again collected the Ir L_{3-} edge XANES spectra in $\text{Ru}_{0.5}\text{Ir}_{0.5}\text{O}_2$, which reconfirm that the valence state of Ir in $\text{Ru}_{0.5}\text{Ir}_{0.5}\text{O}_2$ is higher than that of C- IrO_2 (Fig. R5). The white-line region of Ru K-edge for $\text{Ru}_{0.5}\text{Ir}_{0.5}\text{O}_2$ shows the white-line adsorption energy of which is between that of C- RuO_2 and Ru foil, indicating that the valence state of Ru in

$\text{Ru}_{0.5}\text{Ir}_{0.5}\text{O}_2$ is between 0 and +4. To sum up, the XPS and XAS measurements seem to mutually confirm the valence state of the metal in this work.

2. The updated EXAFS spectra of Ru K-edge spectra for $\text{Ru}_{0.5}\text{Ir}_{0.5}\text{O}_2$, and C-RuO₂ (**Fig. 2e**) were obtained by re-analyzing the XAS data of $\text{Ru}_{0.5}\text{Ir}_{0.5}\text{O}_2$, and C-RuO₂ *via* Athena software.

Fig. 2e indicated that the bond length of Ru-O in $\text{Ru}_{0.5}\text{Ir}_{0.5}\text{O}_2$ was increased compared to that of C-RuO₂ and the length of Ir-O bonds was increased compared to that of C-IrO₂ (**Fig. 2f**). For solid solutions, there are two factors that affect the bond length of metals-oxygen. One is a crystal structure, and the other is a valence state. In the 1T phase structure, the metal-oxygen bond length is longer than that of the rutile phase [Dang, Q. *et al.* Iridium metallene oxide for acidic oxygen evolution catalysis. *Nat. Commun.* **12**, 6007 (2021)]; For valence states, the higher valence state corresponds to the shorter bond length, and the lower valence state corresponds to the longer bond length. In this work, the crystal structure has larger impact on the bond length of Ir-O, so the bond length of the Ir-O in $\text{Ru}_{0.5}\text{Ir}_{0.5}\text{O}_2$ is longer than that of the corresponding rutile phase. At the same time, DFT calculation results (**Supplementary Table 9**) show that the changes of the bond length of Ir-O and Ru-O in $\text{Ru}_{0.5}\text{Ir}_{0.5}\text{O}_2$ were consistent with experimental observations.

3. As depicted in **Supplementary Table 8**, the Bader charge analysis also suggested the oxidation state of Ru in $\text{Ru}_{0.5}\text{Ir}_{0.5}\text{O}_2$ is lower than in C-RuO₂ and that the oxidation state of Ir in $\text{Ru}_{0.5}\text{Ir}_{0.5}\text{O}_2$ is higher than in C-IrO₂.

Figure R5. Repeat XANES characterization of $\text{Ru}_{0.5}\text{Ir}_{0.5}\text{O}_2$ sample. Ir L₃-edge XANES spectra for $\text{Ru}_{0.5}\text{Ir}_{0.5}\text{O}_2$, C-IrO₂ and Ir foil.

Fig. 2 | XPS and XANES characterizations of electrocatalysts. **a**, Ru 3p XPS spectra of Ru_{0.5}Ir_{0.5}O₂ and C-RuO₂. **b**, Ir 4f XPS spectra of Ru_{0.5}Ir_{0.5}O₂ and C-IrO₂. **c**, Ru K-edge XANES spectra of Ru_{0.5}Ir_{0.5}O₂, C-RuO₂ and Ru foil. **d**, Ir L₃-edge XANES spectra for Ru_{0.5}Ir_{0.5}O₂, C-IrO₂ and Ir foil. **e**, Fourier-transformed EXAFS spectra of Ru K-edge spectra for Ru_{0.5}Ir_{0.5}O₂, C-RuO₂ and Ru foil. **f**, Fourier-transformed EXAFS spectra at Ir L₃-edge collected for Ru_{0.5}Ir_{0.5}O₂, C-IrO₂ and Ir foil. **g-i**, Ru K-edge WT-EXAFS of Ru_{0.5}Ir_{0.5}O₂, C-RuO₂ and Ru foil. **j-l**, Ir L₃-edge WT-EXAFS of Ru_{0.5}Ir_{0.5}O₂, C-IrO₂ and Ir foil.

Supplementary Table 8 | The Bader charges of Ru_{0.5}Ir_{0.5}O₂, C-IrO₂ and C-RuO₂ derived from the DFT calculations.

Catalysts	Element	Formal electron loss
Ru _{0.5} Ir _{0.5} O ₂	Ru	1.47
	Ir	1.53
C-IrO ₂	Ru	-
	Ir	1.52
C-RuO ₂	Ru	1.52
	Ir	-

Supplementary Table 9 | The Ru-O and Ir-O bond length (Å) from DFT calculations for various catalysts.

Catalysts	Bond	Bond Length / Å	Bond Length / Å	Bond Length / Å	Bond Length / Å	Bond Length / Å	Bond Length / Å
C-IrO ₂	Ir-O	1.965	1.965	1.958	1.958	1.958	1.958
Ru _{0.5} Ir _{0.5} O ₂	Ir-O	2.008	1.995	1.984	2.035	2.006	2.030
	Ru-O	2.019	2.010	2.022	1.970	2.020	2.004
C-RuO ₂	Ru-O	2.012	2.012	1.926	1.926	1.926	1.926

We have modified the manuscript accordingly:

[Line 122, Page 6] “The valence state results of Ru and Ir species were consistent with those discussed earlier in the XPS results. The Fourier transforms of the extended X-ray absorption fine structure (EXAFS) spectra (**Fig. 2e, f**) at the Ru K-edge and Ir L₃-edge were conducted to investigate the local chemical environment of Ru_{0.5}Ir_{0.5}O₂²⁴. The FT-EXAFS of Ru K-edge and Ir L₃-edge reveal that the bond length of Ru-O in Ru_{0.5}Ir_{0.5}O₂ is slightly increased compared to that of C-RuO₂, and the length of Ir-O bonds is also increased compared to that of C-IrO₂, which may be due to different crystal structure as the metal-oxygen bonds in 1T phase are larger than those in rutile.”

[Line 319, Page 15] “These trends are also supported by the Bader charge analysis as shown in **Supplementary Table 8**. Thus, both the XPS prediction and Bader charge analysis support the consistent oxidation state change compared with experiment. As shown in **Supplementary Table 9**, the bond length of Ru-O in Ru_{0.5}Ir_{0.5}O₂ increased compared to that of C-RuO₂ and the length of Ir-O bonds also increases compared to that of C-IrO₂, which is consistent with experimental observations. For solid solutions, there are two factors that affect the bond length of metals-oxygen. One is the crystal structure, and the other is the valence state. In the 1T phase structure, the metal-oxygen bond length is longer than that of the rutile phase²⁹; For valence states, the higher valence state corresponds to the shorter bond length, while the lower valence state corresponds to the longer bond length. In this work, the crystal structure has larger impact on to the bond length of Ir-O, so the bond length of the Ir-O in Ru_{0.5}Ir_{0.5}O₂ is longer than that of the corresponding rutile phase.”

Fig. 2 | XPS and XANES characterizations of electrocatalysts. **a**, Ru 3p XPS spectra of $\text{Ru}_{0.5}\text{Ir}_{0.5}\text{O}_2$ and C-RuO₂. **b**, Ir 4f XPS spectra of $\text{Ru}_{0.5}\text{Ir}_{0.5}\text{O}_2$ and C-IrO₂. **c**, Ru K-edge XANES spectra of $\text{Ru}_{0.5}\text{Ir}_{0.5}\text{O}_2$, C-RuO₂ and Ru foil. **d**, Ir L₃-edge XANES spectra for $\text{Ru}_{0.5}\text{Ir}_{0.5}\text{O}_2$, C-IrO₂ and Ir foil. **e**, Fourier-transformed EXAFS spectra of Ru K-edge spectra for $\text{Ru}_{0.5}\text{Ir}_{0.5}\text{O}_2$, C-RuO₂ and Ru foil. **f**, Fourier-transformed EXAFS spectra at Ir L₃-edge collected for $\text{Ru}_{0.5}\text{Ir}_{0.5}\text{O}_2$, C-IrO₂ and Ir foil. **g-i**, Ru K-edge WT-EXAFS of $\text{Ru}_{0.5}\text{Ir}_{0.5}\text{O}_2$, C-RuO₂ and Ru foil. **j-l**, Ir L₃-edge WT-EXAFS of $\text{Ru}_{0.5}\text{Ir}_{0.5}\text{O}_2$, C-IrO₂ and Ir foil.

Supplementary Table 8 | The Bader charges of $\text{Ru}_{0.5}\text{Ir}_{0.5}\text{O}_2$, C-IrO₂ and C-RuO₂ derived from the DFT calculations.

Catalysts	Element	Formal electron loss
$\text{Ru}_{0.5}\text{Ir}_{0.5}\text{O}_2$	Ru	1.47
	Ir	1.53
C-IrO ₂	Ru	-
	Ir	1.52
C-RuO ₂	Ru	1.52
	Ir	-

Supplementary Table 9 | The Ru-O and Ir-O bond length (Å) from DFT calculations for various catalysts.

Catalysts	Bond	Bond Length / Å	Bond Length / Å	Bond Length / Å	Bond Length / Å	Bond Length / Å	Bond Length / Å
C-IrO ₂	Ir-O	1.965	1.965	1.958	1.958	1.958	1.958
Ru _{0.5} Ir _{0.5} O ₂	Ir-O	2.008	1.995	1.984	2.035	2.006	2.030
	Ru-O	2.019	2.010	2.022	1.970	2.020	2.004
C-RuO ₂	Ru-O	2.012	2.012	1.926	1.926	1.926	1.926

4. The authors state that the Ru 3p(1/2) XPS peak shifts from 484 eV to 483.6 eV, comparing RuO₂ to Ru_{0.5}Ir_{0.5}O₂. However, the Ru 3p(1/2) XPS peak for the as-synthesized catalyst is not shown in Fig 2a, nor in the SI. The authors should either include this peak in Fig 2a or show it as a separate figure in the SI, like in Fig S15.

[Author's Response]: Thank you for your valuable comments. According to your suggestion, we have modified the Ru 3p XPS spectra in the revised manuscript (Fig. 2a and Supplementary Fig. 21a).

Fig. 2 | XPS and XANES characterizations of electrocatalysts. a, Ru 3p XPS spectra of Ru_{0.5}Ir_{0.5}O₂ and C-RuO₂.

Supplementary Fig. 21 | XPS characterizations of Ru_{0.5}Ir_{0.5}O₂ after long-term stability test at 10 mA cm⁻². a, Ru 3p.

5. In Fig 2a, what is the oxidation state of Ru as measured in XPS? The authors should show their fitting procedure and estimate an oxidation state based on the fit. I have the same question about Fig 2b: what are the fitted peaks and what Ir oxidation states do they correspond to? What is the overall oxidation state of Ir as measured via XPS? The authors should provide all of this information. They

should show peak assignments and provide more information on the XPS fitting process in the SI. Do the surface-sensitive oxidation states of Ir and Ru as measured by XPS match the bulk-sensitive oxidation state information obtained via XANES in Figs 2c&d? If not, the authors should hypothesize why.

[Author's Response]: Thank you for your valuable comments.

1. Following your valuable suggestion, we have modified the Ru 3p XPS spectra and Ir 4f XPS spectra in the revised manuscript (**Fig. 2** and **Supplementary Fig. 21**). As depicted in **Fig. 2**, Ru_{0.5}Ir_{0.5}O₂ exhibits a lower valence state of Ru sites than that in C-RuO₂ (Ru⁴⁺) and a higher Ir valence state than that in C-IrO₂ (Ir⁴⁺).
2. **Supplementary Table 3** shows the fitting parameters used, as well as the details of Ru 3p and Ir 4f fitting for Ru_{0.5}Ir_{0.5}O₂ and other samples.
3. The XANES data in **Fig. 2c&d** show that the Ru valence state is lower than the C-RuO₂ (Ru⁴⁺), and the Ir valence state is slightly higher than the C-IrO₂ (Ir⁴⁺). These results demonstrated the surface-sensitive oxidation states of Ir and Ru as measured by XPS match the bulk-sensitive oxidation state information obtained *via* XANES.

Fig. 2 | XPS and XANES characterizations of electrocatalysts. **a**, Ru 3p XPS spectra of Ru_{0.5}Ir_{0.5}O₂ and C-RuO₂. **b**, Ir 4f XPS spectra of Ru_{0.5}Ir_{0.5}O₂ and C-IrO₂. **c**, Ru K-edge XANES spectra of Ru_{0.5}Ir_{0.5}O₂, C-RuO₂ and Ru foil. **d**, Ir L₃-edge XANES spectra for Ru_{0.5}Ir_{0.5}O₂, C-IrO₂ and Ir foil. **e**, Fourier-transformed EXAFS spectra of Ru K-edge spectra for Ru_{0.5}Ir_{0.5}O₂, C-RuO₂ and Ru foil. **f**, Fourier-transformed EXAFS spectra at Ir L₃-edge collected for Ru_{0.5}Ir_{0.5}O₂, C-IrO₂ and Ir foil. **g-i**, Ru K-edge WT-EXAFS of Ru_{0.5}Ir_{0.5}O₂, C-RuO₂ and Ru foil. **j-l**, Ir L₃-edge WT-EXAFS of Ru_{0.5}Ir_{0.5}O₂, C-IrO₂ and Ir foil.

Supplementary Fig. 21 | XPS characterizations of $\text{Ru}_{0.5}\text{Ir}_{0.5}\text{O}_2$ after long-term stability test at 10 mA cm^{-2} . a, Ru 3p and b, Ir 4f XPS spectra of $\text{Ru}_{0.5}\text{Ir}_{0.5}\text{O}_2$ after the stability test.

Supplementary Table 3 | XPS line shapes, binding energies and fitting parameters of Ru 3p and Ir 4f XPS spectra for Fig. 2b using CasaXPS software.

Catalysts	Peak	Position / eV	Area	FWHM / eV	Lineshape
$\text{Ru}_{0.5}\text{Ir}_{0.5}\text{O}_2$	Ru $3p_{3/2}$	461.6	53801.9	4.4	GL (30)
	Ru $3p_{1/2}$	483.6	25813.6	4.4	GL (30)
	Ir $4f_{7/2}$	62.2	15000.8	1.6	GL (30)
	Ir $4f_{5/2}$	65.2	12282.7	1.6	GL (30)
	Ir $4f_{7/2}$ (satellite 1)	63.6	5439.5	1.7	GL (30)
	Ir $4f_{5/2}$ (satellite 1)	66.6	4100.0	1.7	GL (30)
	Ir $4f_{7/2}$ (satellite 2)	67.5	6344.8	3.4	GL (30)
C-IrO ₂	Ir $4f_{7/2}$	61.8	10762.0	1.26	GL (30)
	Ir $4f_{5/2}$	64.8	8554.1	1.26	GL (30)
	Ir $4f_{7/2}$ (satellite 1)	63.1	4775.2	1.6	GL (30)
	Ir $4f_{5/2}$ (satellite 1)	66.1	3523.8	1.6	GL (30)
	Ir $4f_{7/2}$ (satellite 2)	67.1	4457.2	3.2	GL (30)
C-RuO ₂	Ru $3p_{3/2}$	462	14674.6	3.55	GL (30)

Note: FWHM, full width at half maximum.

We have modified the manuscript accordingly:

[Line 107, Page 5] “As depicted in the XPS spectra, Ru_{0.5}Ir_{0.5}O₂ exhibits a meaningful negative-shift of Ru 3p_{3/2} and Ru 3p_{1/2} peaks (461.6 and 483.6 eV) in comparison with those for C-RuO₂ (462.0 and 484.0 eV) (Fig. 2a), which confirms the a lower valence state of Ru in Ru_{0.5}Ir_{0.5}O₂ than that in C-RuO₂ (Ru⁴⁺)^{18,24}. Ir 4f XPS spectra (Fig. 2b) show the peaks located at 62.2 and 65.2 eV, which are assigned to Ir 4f_{7/2} and Ir 4f_{5/2} of Ir⁴⁺²⁴. Compared with those of C-IrO₂ (61.8 and 64.8 eV), Ru_{0.5}Ir_{0.5}O₂ shows a slight positive-shift. Which indicates that Ru_{0.5}Ir_{0.5}O₂ has the higher Ir valence state than C-IrO₂³⁰. The fitting parameters used of all peaks can be found in **Supplementary Table 3.**”

[Line 122, Page 6] “The valance state results of Ru and Ir species were consistent with those discussed earlier in the XPS results.”

Fig. 2 | XPS and XANES characterizations of electrocatalysts. a, Ru 3p XPS spectra of Ru_{0.5}Ir_{0.5}O₂ and C-RuO₂. **b,** Ir 4f XPS spectra of Ru_{0.5}Ir_{0.5}O₂ and C-IrO₂. **c,** Ru K-edge XANES spectra of Ru_{0.5}Ir_{0.5}O₂, C-RuO₂ and Ru foil. **d,** Ir L₃-edge XANES spectra for Ru_{0.5}Ir_{0.5}O₂, C-IrO₂ and Ir foil. **e,** Fourier-transformed EXAFS spectra of Ru K-edge spectra for Ru_{0.5}Ir_{0.5}O₂, C-RuO₂ and Ru foil. **f,** Fourier-transformed EXAFS spectra at Ir L₃-edge collected for Ru_{0.5}Ir_{0.5}O₂, C-IrO₂ and Ir foil. **g-i,** Ru K-edge WT-EXAFS of Ru_{0.5}Ir_{0.5}O₂, C-RuO₂ and Ru foil. **j-l,** Ir L₃-edge WT-EXAFS of Ru_{0.5}Ir_{0.5}O₂, C-IrO₂ and Ir foil.

Supplementary Table 3 | XPS line shapes, binding energies and fitting parameters of Ru 3p and Ir 4f XPS spectra for Fig. 2b using CasaXPS software.

Catalysts	Peak	Position / eV	Area	FWHM / eV	Lineshape
Ru _{0.5} Ir _{0.5} O ₂	Ru 3p _{3/2}	461.6	53801.9	4.4	GL (30)
	Ru 3p _{1/2}	483.6	25813.6	4.4	GL (30)
	Ir 4f _{7/2}	62.2	15000.8	1.6	GL (30)
	Ir 4f _{5/2}	65.2	12282.7	1.6	GL (30)
	Ir 4f _{7/2} (satellite 1)	63.6	5439.5	1.7	GL (30)
	Ir 4f _{5/2} (satellite 1)	66.6	4100.0	1.7	GL (30)
	Ir 4f _{7/2} (satellite 2)	67.5	6344.8	3.4	GL (30)
	Ir 4f _{7/2}	61.8	10762.0	1.26	GL (30)
C-IrO ₂	Ir 4f _{5/2}	64.8	8554.1	1.26	GL (30)
	Ir 4f _{7/2} (satellite 1)	63.1	4775.2	1.6	GL (30)
	Ir 4f _{5/2} (satellite 1)	66.1	3523.8	1.6	GL (30)
	Ir 4f _{7/2} (satellite 2)	67.1	4457.2	3.2	GL (30)
	Ir 4f _{7/2}	61.8	10762.0	1.26	GL (30)
C-RuO ₂	Ru 3p _{3/2}	462	14674.6	3.55	GL (30)
	Ru 3p _{1/2}	484	7318.4	3.55	GL (30)

Note: FWHM, full width at half maximum.

6. The authors should perform EXAFS and XANES on the Ru_{0.5}Ir_{0.5}O₂ catalyst after the long term stability test to further support their claims that the oxidation state changes after passing significant OER current. Does the bulk oxidation state also change or just the surface?

[Author's Response]: Following your valuable suggestion, as shown in **Supplementary Fig. 22**, we have added the XANES analyses on the Ru_{0.5}Ir_{0.5}O₂ catalyst after the long term stability test in the revised manuscript. The XANES data reveal that the valence state of Ru increased slightly and the oxidation state of Ir decreased slightly after the stability test compared with before the test, which further support our claims that the bulk oxidation state changes after passing significant OER current.

Supplementary Fig. 22 | XAS characterizations of $\text{Ru}_{0.5}\text{Ir}_{0.5}\text{O}_2$ catalyst after the long term stability test. a, Ru XANES spectra at the Ru K-edge of $\text{Ru}_{0.5}\text{Ir}_{0.5}\text{O}_2$ before and after the stability test. b, Ir XANES spectra at Ir L_3 -edge of $\text{Ru}_{0.5}\text{Ir}_{0.5}\text{O}_2$ before and after the stability test.

We have modified the manuscript accordingly:

[Line 197, Page 9] “Furthermore, the XANES analyses also reveal that the valence state of Ru increased slightly and the oxidation state of Ir decreased slightly after the stability test compared with before the test (**Supplementary Fig. 22**).”

Supplementary Fig. 22 | XAS characterizations of $\text{Ru}_{0.5}\text{Ir}_{0.5}\text{O}_2$ catalyst after the long term stability test. a, Ru XANES spectra at the Ru K-edge of $\text{Ru}_{0.5}\text{Ir}_{0.5}\text{O}_2$ before and after the stability test. b, Ir XANES spectra at Ir L_3 -edge of $\text{Ru}_{0.5}\text{Ir}_{0.5}\text{O}_2$ before and after the stability test.

7. The authors should report the electrocatalytic activity and all of the figures of merit normalized by BET surface area rather than the geometric area, since the BET surface area of $\text{Ru}_{0.5}\text{Ir}_{0.5}\text{O}_2$ is 4x higher than RuO_2 and 8x higher than IrO_2 . Reporting these values normalized by the BET surface area will be a better comparison and lend more credibility to the arguments made.

[Author’s Response]: Thank you for your valuable suggestion. We added the BET-normalized OER activity in the revised manuscript. As shown in **Supplementary Fig. 19**, The OER activity of $\text{Ru}_{0.5}\text{Ir}_{0.5}\text{O}_2$ outperformed those of all the other samples, demonstrating the superior intrinsic OER catalytic activity of $\text{Ru}_{0.5}\text{Ir}_{0.5}\text{O}_2$.

Supplementary Fig. 19 | OER performance. Normalized LSV curves to BET-based areas of $\text{Ru}_{0.5}\text{Ir}_{0.5}\text{O}_2$, C-IrO_2 and C-RuO_2 .

We have modified the manuscript accordingly:

[Line 177, Page 8] “The OER activity normalized to the ECSA and BET of the catalysts were also calculated (**Supplementary Figs. 18, 19**). The specific activity of $\text{Ru}_{0.5}\text{Ir}_{0.5}\text{O}_2$ was better than those of all the other samples. These results show that $\text{Ru}_{0.5}\text{Ir}_{0.5}\text{O}_2$ has superior intrinsic OER catalytic activity.”

Supplementary Fig. 19 | OER performance. Normalized LSV curves to BET-based areas of $\text{Ru}_{0.5}\text{Ir}_{0.5}\text{O}_2$, C-IrO_2 and C-RuO_2 .

8. In Fig 5, the authors modeled the active site of $\text{Ru}_{0.5}\text{Ir}_{0.5}\text{O}_2$ as a defect Ru atom in the middle of the slab. However, it is very unlikely this site is the true active site, as its formation requires the breaking of multiple M-O bonds and consequently has a large formation energy. This is why the formation of the *OH adsorbate is 1.79 eV and is calculated in this work to be the potential limiting step despite the potential limiting step on most OER catalysts involving the O^* adsorbate [Man, Rossmeisl, ChemCatChem, 2011].

[Author’s Response]: We appreciate your insight regarding the active site model in **Fig. 5** and the potential implications for the calculated potential limiting step. We understand that our initial model may not accurately represent the true active site, and we are grateful for the opportunity to address this issue.

Indeed, our prediction of overpotentials are much higher than the experiment because our simulation model has no defects. As inspired by the reviewer, we benchmarked many factors that affects the overpotential predictions, and found that the type of surface site is most important. We considered the edges of $\text{Ru}_{0.5}\text{Ir}_{0.5}\text{O}_2$ as active sites, and we obtained significantly lower overpotential of 360 mV on (01-10) edge-site of $\text{Ru}_{0.5}\text{Ir}_{0.5}\text{O}_2$ and the potential limiting step becomes O_2 formation. Comparing with the experimentally observed overpotential of 151 mV, we consider the DFT calculation now well reproduce the experiment within uncertainty.

In this revised reaction pathway, the final step is the potential-determining step (PDT). However, this step is only 0.02 eV higher than the formation of *OOH. Thus, statistically, this is consistent with the findings that the potential limiting step on most OER catalysts involving the O* adsorbate as reported in the existing literature. [Man, I. C. *et al.* Universality in oxygen evolution electrocatalysis on oxide surfaces. *ChemCatChem* **3**, 1159-1165 (2011)]

We have included above discussion and cited the 2011 referene in the revised manuscript.

Fig. 5 | DFT simulation findings of Ru_{0.5}Ir_{0.5}O₂. (a) Atomistic structure and E_{dft} of the Ru_{0.5}Ir_{0.5}O₂ (-331.4 eV). Atomistic structures of C-IrO₂ (b) and C-RuO₂ (c) (blue, Ru; grey, Ir; red, O). d, Schematic illustration of OER mechanism on the Ru_{0.5}Ir_{0.5}O₂ (blue, Ru; grey, Ir; red, O; white, H). e, The reaction paths on Ru_{0.5}Ir_{0.5}O₂ catalyst with the set potential of 0 and 1.23 V. The overpotential (η) is labeled for viewing convenience.

We have modified the manuscript accordingly:

[Line 330, Page 15] “We chose to study the catalytic performance of Ru_{0.5}Ir_{0.5}O₂ (01-10) edge-site^{47,48}, C-IrO₂ (110) surface and C-RuO₂ (110) surface^{29,49,50}(Fig. 5a).”

[Line 337, Page 16] “For Ru_{0.5}Ir_{0.5}O₂, generation of O₂ is the PDS with an overpotential of 0.36 V. Thus, our DFT simulations showed that the Ru_{0.5}Ir_{0.5}O₂ has the lowest OER overpotential ($\eta_{\text{OER}} = 0.36$ V) compared to C-IrO₂ ($\eta_{\text{OER}} = 1.00$ V) and C-RuO₂ ($\eta_{\text{OER}} = 0.81$ V), coincident with our experimental observation that OER stability and catalytic activity of Ru_{0.5}Ir_{0.5}O₂ catalyst were significantly improved.”

Fig. 5 | DFT simulation findings of Ru_{0.5}Ir_{0.5}O₂. (a) Atomistic structure and E_{dft} of the Ru_{0.5}Ir_{0.5}O₂ (-331.4 eV). Atomistic structures of C-IrO₂ (b) and C-RuO₂ (c) (blue, Ru; grey, Ir; red, O). **d**, Schematic illustration of OER mechanism on the Ru_{0.5}Ir_{0.5}O₂ (blue, Ru; grey, Ir; red, O; white, H). **e**, The reaction paths on Ru_{0.5}Ir_{0.5}O₂ catalyst with the set potential of 0 and 1.23 V. The overpotential (η) is labeled for viewing convenience.

9. The calculated theoretical overpotentials in this work are vastly different from the experimental OER onset potentials, both reported in this work and in previous works [Ref 13]. Yet, the authors don't discuss this discrepancy in the manuscript. The authors need to revisit their DFT calculations and explain why they differ so much from experiment. Specifically, the authors need to consider other sites, like the undercoordinated oxygens on the edge of the slab, which have previously been shown to be much more active on Ir-Ru-O alloys [Man (2011). Theoretical study of Electro-catalysts for oxygen evolution, DTU].

[Author's Response]: Thank you for pointing out our omission in discussing the discrepancy between the estimated overpotentials for Ru_{0.5}Ir_{0.5}O₂ and the experimental evidence.

The reviewer rightly pointed out the discrepancy between our calculated theoretical overpotentials and the experimental OER onset potentials. After systematically comparing the available published results (**Supplementary Table 10**), we realize that most of the existing DFT calculations overestimate the OER overpotentials of various reported catalysts. Among the many factors affecting the overpotential predictions, we found the type of surface site is most important.

As inspired by the reviewer, we considered the edges of Ru_{0.5}Ir_{0.5}O₂ as active sites, and recalculated the (01-10) edge-site of Ru_{0.5}Ir_{0.5}O₂. For Ru_{0.5}Ir_{0.5}O₂, the limit-rating step changes to the generation of O₂ with an overpotential of 0.36 V (**Fig. 5**). Comparing with the experimentally observed overpotential of 151 mV, we consider the DFT calculation well reproduce the experiment and the remaining difference is already within the uncertainty that comes from sources such as the choice of functional, simplifications in the theoretical models, inherent DFT approximations, and possible variations in

experimental conditions and measurements.

We are grateful for the reference [Man, I. C. *et al.* Theoretical study of electro-catalysts for oxygen evolution. Technical University of Denmark (2011)] suggested by the reviewer, which provided us with essential insights into the behavior of undercoordinated oxygens on the edge of the slab. Therefore, we have cited this paper in the revised manuscript to acknowledge the importance of considering undercoordinated oxygens edge sites.

Fig. 5 | DFT simulation findings of Ru_{0.5}Ir_{0.5}O₂. (a) Atomistic structure and E_{dft} of the Ru_{0.5}Ir_{0.5}O₂ (-331.4 eV). Atomistic structures of C-IrO₂ (b) and C-RuO₂ (c) (blue, Ru; grey, Ir; red, O). **d**, Schematic illustration of OER mechanism on the Ru_{0.5}Ir_{0.5}O₂ (blue, Ru; grey, Ir; red, O; white, H). **e**, The reaction paths on Ru_{0.5}Ir_{0.5}O₂ catalyst with the set potential of 0 and 1.23 V. The overpotential (η) is labeled for viewing convenience.

Supplementary Table 10 | The comparison of the OER performances (overpotential and estimated overpotential) of Ru_{0.5}Ir_{0.5}O₂ and various reported catalysts.

Electrocatalysts	Electrolyte	Overpotential ^a / mV	Estimated Overpotential / mV	Reference
Ru _{0.5} Ir _{0.5} O ₂	0.5 M H ₂ SO ₄	151	360	This work
Re-RuO ₂	0.1 M HClO ₄	190	790	Nat. Commun. 2023 , 14 , 354

Li _{0.52} RuO ₂	0.5 M H ₂ SO ₄	156	510	Nat. Commun. 2022 , 13, 3784
RuIrO _x	0.5 M H ₂ SO ₄	233	400	Nat. Commun. 2019 , 10, 4875
CaCu ₃ Ru ₄ O ₁₂	0.5 M H ₂ SO ₄	171	660	Nat. Commun. 2019 , 10, 3809
SrRuIr	0.5 M H ₂ SO ₄	190	370	J. Am. Chem. Soc. 2021 , 143, 6482- 6490
Ru ₁ Ir ₁ O _x	0.5 M H ₂ SO ₄	204	650	Adv. Energy Mater. 2021 , 11, 2102883
3R-IrO ₂	0.1 M HClO ₄	188	550	Joule 2021 , 5, 1-14
Ru ₁ -Pt ₃ Cu	0.1 M HClO ₄	220	420	Nat. Catal. 2019 , 2, 304-313
W _{0.2} Er _{0.1} Ru _{0.7} O _{2-δ}	0.5 M H ₂ SO ₄	168	530	Nat. Commun. 2020 , 11, 5368
Ru-N-C	0.5 M H ₂ SO ₄	267	590	Nat. Commun. 2019 , 10, 4849
Cu-doped RuO ₂	0.5 M H ₂ SO ₄	188	660	Adv. Mater. 2018 , 30, 1801351
RuO ₂ NSs	0.5 M H ₂ SO ₄	199	450	Energy Environ. Sci. 2020 , 13, 5143-5151
RuO ₂ nanosheet	0.1 M HClO ₄	255	430	Adv. Energy Mater. 2019 , 9, 1803795
Mn-RuO ₂	0.5 M H ₂ SO ₄	158	1480	ACS Catal. 2020 , 10, 1152-1160

a: The overpotentials required to achieve a current density of 10 mA cm⁻².

We have modified the manuscript accordingly:

[Line 343, Page 16] “After systematically comparing the available published results (**Supplementary Table 10**), we realize that most of the existing DFT calculations overestimate the OER overpotentials of various reported catalysts. Among the many factors affecting the overpotential predictions, we found the type of surface site is most important. We considered the edges of Ru_{0.5}Ir_{0.5}O₂ as active sites. These simulations yielded a considerably lower overpotential of 360 mV, which aligns much more closely with the experimentally observed overpotential of 151 mV. We believe that this now adequately explained the difference between the predicted overpotential of DFT calculation and the experiment

results. Remaining differences can be attributed to various sources of uncertainty inherent in both theoretical and experimental methods.”

Fig. 5 | DFT simulation findings of $\text{Ru}_{0.5}\text{Ir}_{0.5}\text{O}_2$. (a) Atomistic structure and E_{dft} of the $\text{Ru}_{0.5}\text{Ir}_{0.5}\text{O}_2$ (-331.4 eV). Atomistic structures of C-IrO₂ (b) and C-RuO₂ (c) (blue, Ru; grey, Ir; red, O). **d**, Schematic illustration of OER mechanism on the $\text{Ru}_{0.5}\text{Ir}_{0.5}\text{O}_2$ (blue, Ru; grey, Ir; red, O; white, H). **e**, The reaction paths on $\text{Ru}_{0.5}\text{Ir}_{0.5}\text{O}_2$ catalyst with the set potential of 0 and 1.23 V. The overpotential (η) is labeled for viewing convenience.

Supplementary Table 10 | The comparison of the OER performances (overpotential and estimated overpotential) of $\text{Ru}_{0.5}\text{Ir}_{0.5}\text{O}_2$ and various reported catalysts.

Electrocatalysts	Electrolyte	Overpotential ^a / mV	Estimated Overpotential / mV	Reference
$\text{Ru}_{0.5}\text{Ir}_{0.5}\text{O}_2$	0.5 M H_2SO_4	151	360	This work
Re-RuO ₂	0.1 M HClO_4	190	790	Nat. Commun. 2023 , 14, 354
$\text{Li}_{0.52}\text{RuO}_2$	0.5 M H_2SO_4	156	510	Nat. Commun. 2022 , 13, 3784
RuIrO_x	0.5 M H_2SO_4	233	400	Nat. Commun. 2019 , 10, 4875

CaCu ₃ Ru ₄ O ₁₂	0.5 M H ₂ SO ₄	171	660	Nat. Commun. 2019 , 10, 3809
SrRuIr	0.5 M H ₂ SO ₄	190	370	J. Am. Chem. Soc. 2021 , 143, 6482- 6490
Ru ₁ Ir ₁ O _x	0.5 M H ₂ SO ₄	204	650	Adv. Energy Mater. 2021 , 11, 2102883
3R-IrO ₂	0.1 M HClO ₄	188	550	Joule 2021 , 5, 1-14
Ru ₁ -Pt ₃ Cu	0.1 M HClO ₄	220	420	Nat. Catal. 2019 , 2, 304-313
W _{0.2} Er _{0.1} Ru _{0.7} O ₂₋₈	0.5 M H ₂ SO ₄	168	530	Nat. Commun. 2020 , 11, 5368
Ru-N-C	0.5 M H ₂ SO ₄	267	590	Nat. Commun. 2019 , 10, 4849
Cu-doped RuO ₂	0.5 M H ₂ SO ₄	188	660	Adv. Mater. 2018 , 30, 1801351
RuO ₂ NSs	0.5 M H ₂ SO ₄	199	450	Energy Environ. Sci. 2020 , 13, 5143-5151
RuO ₂ nanosheet	0.1 M HClO ₄	255	430	Adv. Energy Mater. 2019 , 9, 1803795
Mn-RuO ₂	0.5 M H ₂ SO ₄	158	1480	ACS Catal. 2020 , 10, 1152-1160

a: The overpotentials required to achieve a current density of 10 mA cm⁻².

10. Why do the authors use an ordered model for DFT calculations but also state that Ru and Ir form a solid solution for Ru_{0.5}Ir_{0.5}O₂ (implying that it is not an ordered structure)? As the EXAFS experiments suggest, the Ru-O and Ir-O bond lengths are different and, if indeed an ordered structure is formed, this should result in superstructure reflections in the SAED data (which was not observed nor discussed). If Ru_{0.5}Ir_{0.5}O₂ is truly a solid-solution material then Ru and Ir atoms wouldn't be expected to order in-plane. The authors should include a DFT model of a slab with a random in-plane distribution of Ru and Ir atoms to compare. In Fig S22, how much more stable are the slabs that have in-plane order of Ru atoms? The authors present a variety of structures and state they use the lowest energy structure, but no information is given about the relative energies.

[Author's Response]: We are grateful for the insightful comments and suggestions made by the reviewer. We have carefully considered each of the issues raised and made necessary revisions to our manuscript accordingly.

We concur with the reviewer's statement regarding the potential confusion arising from the term "solid solution". Our intention was to convey that Ru and Ir are uniformly distributed across the metallic sites.

In our DFT models for $\text{Ru}_{0.5}\text{Ir}_{0.5}\text{O}_2$, we approached the problem by substituting Ir atoms of 2D IrO_2 with Ru atoms to reach an atomic ratio of $\text{Ru}/\text{Ir} = 1:1$, which aligns closely with the experimental value. To address your concern, we constructed $\text{Ru}_{0.5}\text{Ir}_{0.5}\text{O}_2$ structure models with 10 independent configurations of Ir and Ru atoms, which were randomly arranged (as depicted in **Fig. 5a** and **Supplementary Fig. 33**). The $\text{Ru}_{0.5}\text{Ir}_{0.5}\text{O}_2$ structure with the lowest relative energy and highest stability was then selected through DFT optimization (**Fig. 5a**).

Regarding your question about the relative energy levels in **Fig. S22** (**Supplementary Fig. 33** in the revised Supporting Information), we apologize for the oversight. The slabs with in-plane order Ru atoms are -331.4 eV more stable, and all the potential energies are referenced to **Fig. 5a**, which is the most stable structure in DFT calculations. We have updated such information in revised Supporting Information.

Fig. 5 | DFT simulation findings of $\text{Ru}_{0.5}\text{Ir}_{0.5}\text{O}_2$. (a) Atomistic structure and E_{dft} of the $\text{Ru}_{0.5}\text{Ir}_{0.5}\text{O}_2$ (-331.4 eV). Atomistic structures of C-IrO_2 (b) and C-RuO_2 (c) (blue, Ru; grey, Ir; red, O). d, Schematic illustration of OER mechanism on the $\text{Ru}_{0.5}\text{Ir}_{0.5}\text{O}_2$ (blue, Ru; grey, Ir; red, O; white, H). e, The reaction paths on $\text{Ru}_{0.5}\text{Ir}_{0.5}\text{O}_2$ catalyst with the set potential of 0 and 1.23 V. The overpotential (η) is labeled for viewing convenience.

Supplementary Fig. 33 | Calculation of $\text{Ru}_{0.5}\text{Ir}_{0.5}\text{O}_2$ structural stability. (a) to (i) represent the structures and ΔE_{dft} of $\text{Ru}_{0.5}\text{Ir}_{0.5}\text{O}_2$ with different structure types and all the potential energies are referenced to the most stable structure (-331.4 eV) in DFT calculations. The Ir, Ru and O atoms are represented with the grey, blue and red, respectively. In our DFT models for $\text{Ru}_{0.5}\text{Ir}_{0.5}\text{O}_2$, we approached the problem by expanding the 2D IrO_2 which is obtained from XRD by 4 times along the x and y direction, and substituting Ir atoms with Ru atoms to reach an atomic ratio of $\text{Ru}/\text{Ir} = 1:1$, which aligns closely with the experimental value, we calculated some different structures $\text{Ru}_{0.5}\text{Ir}_{0.5}\text{O}_2$ (01-10) surface, and pick the most stable structure as our calculated model.

We have modified the manuscript accordingly:

[Line 310, Page 14] “In order to obtain relatively stable $\text{Ru}_{0.5}\text{Ir}_{0.5}\text{O}_2$ structure, we constructed $\text{Ru}_{0.5}\text{Ir}_{0.5}\text{O}_2$ with different permutations of Ir atoms and Ru atoms, and performed optimization calculations on these models (Fig. 5a and Supplementary Fig. 33), the calculation results of the most stable structure are shown in Fig. 5a.”

Fig. 5 | DFT simulation findings of $\text{Ru}_{0.5}\text{Ir}_{0.5}\text{O}_2$. (a) Atomistic structure and E_{dft} of the $\text{Ru}_{0.5}\text{Ir}_{0.5}\text{O}_2$ (-331.4 eV). Atomistic structures of C-IrO_2 (b) and C-RuO_2 (c) (blue, Ru; grey, Ir; red, O). **d**, Schematic illustration of OER mechanism on the $\text{Ru}_{0.5}\text{Ir}_{0.5}\text{O}_2$ (blue, Ru; grey, Ir; red, O; white, H). **e**, The reaction paths on $\text{Ru}_{0.5}\text{Ir}_{0.5}\text{O}_2$ catalyst with the set potential of 0 and 1.23 V. The overpotential (η) is labeled for viewing convenience.

Supplementary Fig. 33 | Calculation of $\text{Ru}_{0.5}\text{Ir}_{0.5}\text{O}_2$ structural stability. (a) to (i) represent the structures and ΔE_{dft} of $\text{Ru}_{0.5}\text{Ir}_{0.5}\text{O}_2$ with different structure types and all the potential energies are referenced to the most stable structure (-331.4 eV) in DFT calculations. The Ir, Ru and O atoms are represented with the grey, blue and red, respectively. In our DFT models for $\text{Ru}_{0.5}\text{Ir}_{0.5}\text{O}_2$, we approached the problem by expanding the 2D IrO_2 which is obtained from XRD by 4 times along the x and y direction, and substituting Ir atoms with Ru atoms to reach an atomic ratio of $\text{Ru}/\text{Ir} = 1:1$, which aligns closely with the experimental value, we calculated some different structures $\text{Ru}_{0.5}\text{Ir}_{0.5}\text{O}_2$ (01-10) surface, and pick the most stable structure as our calculated model.

11. It would be helpful to include a visualization of the layered crystal structure in Fig 1, similar to Fig 5a/d (maybe as an inset in Fig 1b or 1d)

[Author's Response]: Thank you for your suggestion to include a visualization of the layered crystal structure, and we agree that such a visualization would help readers to better understand the structure of $\text{Ru}_{0.5}\text{Ir}_{0.5}\text{O}_2$. We have added a schematic atom structure of $\text{Ru}_{0.5}\text{Ir}_{0.5}\text{O}_2$ to the revised manuscript as **Fig. 1f**, following the reviewer's suggestion. In this visualization, the blue, grey, and red spheres represent the arrangement of Ru, Ir, and O atoms in $\text{Ru}_{0.5}\text{Ir}_{0.5}\text{O}_2$, respectively. We believe this additional figure will enhance the clarity of the manuscript and provide a more intuitive understanding of the material's structure.

Fig. 1 | Structural and phase characterizations of $\text{Ru}_{0.5}\text{Ir}_{0.5}\text{O}_2$. **a**, SEM and **b**, TEM images of $\text{Ru}_{0.5}\text{Ir}_{0.5}\text{O}_2$. **b1**, The SAED pattern of $\text{Ru}_{0.5}\text{Ir}_{0.5}\text{O}_2$, where the hexagonal pattern shows the [0001] projection. **c**, XRD pattern of $\text{Ru}_{0.5}\text{Ir}_{0.5}\text{O}_2$. The inset is a larger view of the marked area in **c**. **d**, The aberration-corrected HAADF-STEM image for $\text{Ru}_{0.5}\text{Ir}_{0.5}\text{O}_2$. **e**, High-magnification image of the region in **d**. **f**, The schematic atom structure of $\text{Ru}_{0.5}\text{Ir}_{0.5}\text{O}_2$, the Ru, Ir, O atoms are represented by blue, grey and red spheres.

We have modified the manuscript accordingly:

[Line 102, Page 5] “ $\text{Ru}_{0.5}\text{Ir}_{0.5}\text{O}_2$ crystal were modelled as shown in **Fig. 1f**, where the blue, grey and red spheres schematically represent the arrangement of Ru, Ir and O atoms of $\text{Ru}_{0.5}\text{Ir}_{0.5}\text{O}_2$.”

Fig. 1 | Structural and phase characterizations of $\text{Ru}_{0.5}\text{Ir}_{0.5}\text{O}_2$. **a**, SEM and **b**, TEM images of

Ru_{0.5}Ir_{0.5}O₂. **b1**, The SAED pattern of Ru_{0.5}Ir_{0.5}O₂, where the hexagonal pattern shows the [0001] projection. **c**, XRD pattern of Ru_{0.5}Ir_{0.5}O₂. The inset is a larger view of the marked area in **c**. **d**, The aberration-corrected HAADF-STEM image for Ru_{0.5}Ir_{0.5}O₂. **e**, High-magnification image of the region in **d**. **f**, The schematic atom structure of Ru_{0.5}Ir_{0.5}O₂, the Ru, Ir, O atoms are represented by blue, grey and red spheres.

12. Are the catalysts listed in Table S2 results from different syntheses or just replicate ICP-AES measurements of the same sample?

[Author's Response]: Thank you! The results in **Supplementary Table 6** show the replicate ICP-AES measurements of the same sample.

13. I am confused by the statement that “the applied voltage acts on the accumulation of charge on the catalyst and affects the current generated...rather than acting directly on the coordinates of the OER reaction. Meanwhile, the activation free energy of the OER reaction decreases linearly with the storage of oxidative charge stored.” The applied voltage acts on the accumulation of charge, which changes the activation free energy. Similarly, in the DFT-derived free energy diagrams, the overpotential is evaluated based on the reaction coordinates which are shifted based on applied potential. Both of these concepts demonstrate that the reaction coordinate is indeed a function of the applied voltage. How is the accumulated charge on the metal compensated by adsorbed surface charge, what is the identity of the adsorbate, and how is this a function of overpotential?

[Author's Response]: We appreciate your thoughtful question regarding the role of applied voltage in the oxygen evolution reaction (OER) and the effect of charge accumulation on the catalyst.

The statement, “the applied voltage acts on the accumulation of charge on the catalyst and affects the current generated...rather than acting directly on the coordinates of the OER reaction. Meanwhile, the activation free energy of the OER reaction decreases linearly with the storage of oxidative charge stored,” was first discussed by Nong, H. N *et al.* [Nong, H. N. *et al.* Key role of chemistry versus bias in electrocatalytic oxygen evolution. *Nature* **587**, 408-413 (2020)]. According to their work, electrocatalysts expedite the reaction by assisting with the necessary electron transfer and facilitating the formation and breaking of chemical bonds.

However, the bond making/breaking process is slower than the electron transfer, making it the main contributor to the reaction coordinate. As Grimaud, A. *et al.* [Grimaud, A. *et al.* Activation of surface oxygen sites on an iridium-based model catalyst for the oxygen evolution reaction. *Nat. Energy*. **2**, 16189 (2017)] pointed out, the charging of catalyst surfaces under bias also influences bond formation and breakage. The accumulation of charge on the catalyst surface elevates the oxidation degree of the metal active site and the concentration of surface oxidation charge, which directly impacts the electrocatalytically generated current.

Importantly, the bias does not directly influence the reaction coordinates of the OER. And the accumulation of oxidation charge under bias primarily increases the number of high-valence active sites. More high-valence active sites translate to enhanced catalytic activity, which modifies the free energy of the OER. Nong, H. N *et al.* discovered that the activation free energy decreases linearly with the amount of oxidative charge stored, forming the basis of electrocatalytic performance.

The reaction coordinate is not directly influenced by the applied voltage. The charge accumulated on the metal arises from metal oxidation, reflected in the increase of the valence state of the metal species, without requiring compensation by additional adsorbate.

In the DFT calculations, we have considered the voltage effect using the standard hydrogen electrode model (SHE), which has been included in our revisions.

We hope this clarification addresses your concerns.

14. The Tafel data presented in Figure 2b is not over a sufficiently large voltage/current range. At minimum, data should be presented over the full range of reported Tafel slope (i.e. if a Tafel slope of 108 mV/dec is reported then a linear fit over 108 mV and a full decade increase in current should be demonstrated). However, this is an overly simplistic view of the Tafel behavior. Realistically (and as evidenced in Figure 2b) the Tafel slope is variable with overpotential. The authors make a strong claim that the accumulated oxidative charge controls the OER (in reference to the original work in DOI: 10.1038/s41586-020-03101-x). As shown in that work, if this effect is the sole determining factor of reactivity the Tafel behavior should be directly tied to the accumulated charge. The log current as a function of accumulated charge should be reported (the relationship between Figure 3b & 4e).

[Author's Response]: Thank you. Following your suggestion, we modified the Tafel slope (**Fig. 3b**) in the revised manuscript. In order to explore the relationship between the accumulated charge and Tafel behavior. As shown in **Supplementary Fig. 28**, the total charge (integral anodic charge) vs. log OER current densities from PVC measurements is linear throughout the potential range, suggesting that the oxidative charge may directly affect the OER rate. [Nong, H. N. *et al.* Key role of chemistry versus bias in electrocatalytic oxygen evolution. *Nature* **587**, 408-413 (2020)] We have also cited this reference [Jiang, Z. *et al.* Filling metal-organic framework mesopores with TiO₂ for CO₂ photoreduction. *Nature* **586**, 549-554 (2021)] in the revised paper.

Fig. 3 | OER performance of Ru_{0.5}Ir_{0.5}O₂ electrocatalyst and the reference samples. a, The OER polarization curves of Ru_{0.5}Ir_{0.5}O₂, C-IrO₂ and C-RuO₂ in O₂-saturated 0.5 M H₂SO₄ electrolyte with iR -correction (mass loading ~ 283 $\mu\text{g cm}^{-2}$). **b**, Tafel plots of Ru_{0.5}Ir_{0.5}O₂, C-IrO₂ and C-RuO₂. **c**, The comparison of overpotentials at 10 mA cm⁻² and current densities at 1.44 V vs. RHE for different catalysts. **d**, Mass activities and TOFs of Ru_{0.5}Ir_{0.5}O₂, C-IrO₂ and C-RuO₂. **e**, The comparison of

chronopotentiometric measurements for different catalysts. **f**, The Comparison of the required overpotential at 10 mA cm^{-2} and chronopotentiometry durability in acidic media for various reported electrocatalysts (Supplementary Table 7).

Supplementary Fig. 28 | Total charge (integral anodic charge) vs. log OER current densities of Ru_{0.5}Ir_{0.5}O₂ (a), C-IrO₂ (b) and C-RuO₂ (c) from PVC measurements.

We have modified the manuscript accordingly:

[Line 151, Page 7] “Besides, the lowest Tafel slope for Ru_{0.5}Ir_{0.5}O₂ (45 mV dec^{-1}) indicates that the fastest kinetic velocity compared to C-IrO₂ (126 mV dec^{-1}) and C-RuO₂ (108 mV dec^{-1}) (Fig. 3b).”

[Line 265, Page 12] “The applied voltage acts on the accumulation of charge on the catalyst and affects the current generated by the electrocatalysis, rather than acting directly on the coordinates of the OER reaction³⁸(Supplementary Fig. 28).”

Fig. 3 | OER performance of Ru_{0.5}Ir_{0.5}O₂ electrocatalyst and the reference samples. a, The OER polarization curves of Ru_{0.5}Ir_{0.5}O₂, C-IrO₂ and C-RuO₂ in O₂-saturated 0.5 M H₂SO₄ electrolyte with *iR*-correction (mass loading $\sim 283 \mu\text{g cm}^{-2}$). **b**, Tafel plots of Ru_{0.5}Ir_{0.5}O₂, C-IrO₂ and C-RuO₂. **c**, The comparison of overpotentials at 10 mA cm⁻² and current densities at 1.44 V vs. RHE for different catalysts. **d**, Mass activities and TOFs of Ru_{0.5}Ir_{0.5}O₂, C-IrO₂ and C-RuO₂. **e**, The comparison of chronopotentiometric measurements for different catalysts. **f**, The Comparison of the required overpotential at 10 mA cm⁻² and chronopotentiometry durability in acidic media for various reported electrocatalysts (Supplementary Table 7).

Supplementary Fig. 28 | Total charge (integral anodic charge) vs. log OER current densities of Ru_{0.5}Ir_{0.5}O₂ (a), C-IrO₂ (b) and C-RuO₂ (c) from PVC measurements.

15. In the anhydrous acetonitrile electrochemical tests, what evidence is there that the redox peaks are associated with Ru redox? Both Ir (C-IrO₂) and Ru (C-RuO₂) exhibit redox peaks in the general voltage range where the peaks are observed for Ru_{0.5}Ir_{0.5}O₂. In fact, the C-RuO₂ sample shows nearly identical E_{1/2} for its redox peaks to C-IrO₂. Without direct evidence that Ru is the redox active ion for these redox peaks (through for example in-situ spectroscopic measurements), the claim that Ru is the active ion is speculation.

[Author's Response]: Thank you for your valuable questions.

1. Cyclic voltammetry (CV) was tested in anhydrous acetonitrile in order to explore the intrinsic redox properties of the sample without OER reactive species (H₂O). This determines the sample in this potential interval undergoes a Faraday reaction or a redox reaction on the sample itself in the presence of the OER reactive species (H₂O). [Rausch, B. *et al.* Decoupled catalytic hydrogen evolution from a molecular metal oxide redox mediator in water splitting. *Science* **345**, 1326-1330 (2014); Symes, M. D. *et al.* Decoupling hydrogen and oxygen evolution during electrolytic water splitting using an electron-coupled-proton buffer. *Nat. Chem.* **5**, 403-409 (2013); Yu, F. Y. *et al.* Pt-O bond as an active site superior to Pt⁰ in hydrogen evolution reaction. *Nat. Commun.* **11**, 490 (2020)]

As shown in **Fig. 4a**, the redox peak onset potential of Ru_{0.5}Ir_{0.5}O₂ in the CV plot is shifted towards low potential relative to that of C-RuO₂ and C-IrO₂, indicating that the metal species in Ru_{0.5}Ir_{0.5}O₂ were more prone to oxidation. And the XANES data and XPS data show that Ru valence state in Ru_{0.5}Ir_{0.5}O₂ is lower than C-RuO₂ (Ru⁴⁺), and Ir valence state is slightly higher than C-IrO₂ (Ir⁴⁺). The DFT calculations also support the Ru/Ir valence state of the Ru_{0.5}Ir_{0.5}O₂ (**Supplementary Fig. 34** and **Supplementary Table 8**). These results show that Ru species in Ru_{0.5}Ir_{0.5}O₂ are more likely to lose electrons and oxidize than Ir species under the same conditions. Therefore, the redox of Ru is a significant contributor to the redox peaks of Ru_{0.5}Ir_{0.5}O₂ in anhydrous acetonitrile.

Fig. 4 | The CV, PVC response and TPV curves of different electrocatalysts. **a**, CV curves of $\text{Ru}_{0.5}\text{Ir}_{0.5}\text{O}_2$, C-IrO₂ and C-RuO₂ in anhydrous acetonitrile. **b**, PVC protocol of $\text{Ru}_{0.5}\text{Ir}_{0.5}\text{O}_2$, C-IrO₂ and C-RuO₂ between 1.195 V vs. RHE cathodic and 1.245 to 1.645 V vs. RHE anodic non-*i*R corrected potentials in O₂-saturated 0.5 M H₂SO₄ electrolyte. **c**, PVC protocol (black) and showing oxidation and reduction with current response (red). **d**, The anodic and inverted cathodic current decay of $\text{Ru}_{0.5}\text{Ir}_{0.5}\text{O}_2$. **e**, Total charge (integral anodic charge) of $\text{Ru}_{0.5}\text{Ir}_{0.5}\text{O}_2$, C-IrO₂ and C-RuO₂ versus potential from PVC. **f**, Anodic capacitance derived from normalized anodic charge to potential step from the PVC. **g**, The TPV curves of $\text{Ru}_{0.5}\text{Ir}_{0.5}\text{O}_2$, C-IrO₂ and C-RuO₂. **h**, Intensity-Time curves of $\text{Ru}_{0.5}\text{Ir}_{0.5}\text{O}_2$, C-IrO₂ and C-RuO₂ at 10 Hz. (*t*₁, *t*₂ and *t*₃ are the peak time of $\text{Ru}_{0.5}\text{Ir}_{0.5}\text{O}_2$, C-IrO₂ and C-RuO₂, respectively). **i**, Comparison of peak occurrence time of $\text{Ru}_{0.5}\text{Ir}_{0.5}\text{O}_2$ with peak occurrence time of C-IrO₂ and C-RuO₂ at different frequencies (10, 20, 30, 40, 50, 60, 70 and 80 Hz).

Supplementary Fig. 34 | The XPS simulation. **a**, Ru 3p XPS simulation spectrum of $\text{Ru}_{0.5}\text{Ir}_{0.5}\text{O}_2$ and C-RuO₂. **b**, Ir 4f XPS simulation spectrum of $\text{Ru}_{0.5}\text{Ir}_{0.5}\text{O}_2$ and C-IrO₂

Supplementary Table 8 | The Bader charges of $\text{Ru}_{0.5}\text{Ir}_{0.5}\text{O}_2$, C-IrO₂ and C-RuO₂ derived from DFT calculations.

Catalysts	Element	Formal electron loss
Ru _{0.5} Ir _{0.5} O ₂	Ru	1.47
	Ir	1.53
C-IrO ₂	Ru	-
	Ir	1.52
C-RuO ₂	Ru	1.52
	Ir	-

We have modified the manuscript accordingly:

[Line 235, Page 11] “As depicted in Fig. 4a, compared to C-IrO₂ and C-RuO₂, Ru_{0.5}Ir_{0.5}O₂ shows an increase in the oxidation state under a lower bias.”

Fig. 4 | The CV, PVC response and TPV curves of different electrocatalysts. a, CV curves of Ru_{0.5}Ir_{0.5}O₂, C-IrO₂ and C-RuO₂ in anhydrous acetonitrile. **b,** PVC protocol of Ru_{0.5}Ir_{0.5}O₂, C-IrO₂ and C-RuO₂ between 1.195 V vs. RHE cathodic and 1.245 to 1.645 V vs. RHE anodic non-iR corrected potentials in O₂-saturated 0.5 M H₂SO₄ electrolyte. **c,** PVC protocol (black) and showing oxidation and reduction with current response (red). **d,** The anodic and inverted cathodic current decay of

Ru_{0.5}Ir_{0.5}O₂. **e**, Total charge (integral anodic charge) of Ru_{0.5}Ir_{0.5}O₂, C-IrO₂ and C-RuO₂ versus potential from PVC. **f**, Anodic capacitance derived from normalized anodic charge to potential step from the PVC. **g**, The TPV curves of Ru_{0.5}Ir_{0.5}O₂, C-IrO₂ and C-RuO₂. **h**, Intensity-Time curves of Ru_{0.5}Ir_{0.5}O₂, C-IrO₂ and C-RuO₂ at 10 Hz. (t₁, t₂ and t₃ are the peak time of Ru_{0.5}Ir_{0.5}O₂, C-IrO₂ and C-RuO₂, respectively). **i**, Comparison of peak occurrence time of Ru_{0.5}Ir_{0.5}O₂ with peak occurrence time of C-IrO₂ and C-RuO₂ at different frequencies (10, 20, 30 40, 50, 60, 70 and 80 Hz).

16. What are the reactions that occur in the acetonitrile experiments and how are these connected to the OER voltages? Is this only surface adsorption? If so, what ion charge compensates the redox of the catalyst? If it is adsorption, linear current scaling with scan rate should be demonstrated. If it is not adsorption, what reaction defines the reversible redox peaks observed and why is there a clear background oxidative current being driven on Ru_{0.5}Ir_{0.5}O₂. I do not see how this data informs the interpretation of the OER experiments.

[Author's Response]: Thank you for your valuable questions.

1. Cyclic voltammetry (CV) was tested in anhydrous acetonitrile in order to explore the intrinsic redox properties of the sample without OER reactive species (H₂O). [Rausch, B. *et al.* Decoupled catalytic hydrogen evolution from a molecular metal oxide redox mediator in water splitting. *Science* **345**, 1326-1330 (2014); Symes, M. D. *et al.* Decoupling hydrogen and oxygen evolution during electrolytic water splitting using an electron-coupled-proton buffer. *Nat. Chem.* **5**, 403-409 (2013); Yu, F. Y. *et al.* Pt-O bond as an active site superior to Pt⁰ in hydrogen evolution reaction. *Nat. Commun.* **11**, 490 (2020)]
2. The CV tested in anhydrous acetonitrile is not directly related to the OER reaction potential. This is not an adsorption reaction. The CV test in anhydrous acetonitrile can determine whether a redox reaction on the sample itself in this potential interval in the presence of the OER reactive species (H₂O).
3. In fact, we find that Ru_{0.5}Ir_{0.5}O₂ is more prone to oxidation reaction than C-RuO₂ and C-IrO₂ through CV test in anhydrous acetonitrile. Combined with previous XPS, XANES and DFT data, it is speculated that this is mainly due to the oxidation of Ru species in Ru_{0.5}Ir_{0.5}O₂. And the pulse voltage induced current (PVC) test also provided identical oxidation state trends, suggesting that the observed behavior is general. Meanwhile, the activation free energy decreases linearly with the degree of oxidation of the active site in the catalyst, which responds to the OER rate. [Nong, H. N. *et al.* Key role of chemistry versus bias in electrocatalytic oxygen evolution. *Nature* **587**, 408-413 (2020)]

17. The TPV measurements are not performed in electrolyte and not performed under applied potential. Clearly, they are measuring electron transfer rates between the samples and Pt. It is not clear how these measurements have any relevance to electron transfer rates for oxygen evolution which are determined by the relative energetics between redox states of the catalyst surface and adsorbates.

[Author's Response]: Thank you for your detailed examination of our work and your thoughtful questions. We appreciate your thoughtful question regarding the TPV measurements and the effect on OER catalysts.

1. Transient photo-induced voltage (TPV) measurements explore the electron transport behavior of material surfaces induced by transient photo-induced voltage. Photovoltage induces the shift and recombination of the electron on the surface of the material. And the transfer behavior of the electron at this time can also reflect the transfer behavior of electrons at the interface of the

material table during the electrocatalytic process. In addition to OER reactions, TPV can also be used in other reactions to explore electron transfer behavior on material surfaces. In fact, Pt only acts as a conductive and facilitates the collection of electrical signals generated by the sample surface after light excitation. [Zhou, Y. J. *et al.* Amino modified carbon dots with electron sink effect increase interface charge transfer rate of Cu-based electrocatalyst to enhance the CO₂ conversion selectivity to C₂H₄. *Adv. Funct. Mater.*, **32**, 2113335 (2022); Dong, M. *et al.* CO₂ dominated bifunctional catalytic sites for efficient industrial exhaust conversion. *Adv. Funct. Mater.* **32**, 2110136 (2022).]

2. The TPV tests does not reveal the charge consumption process occurring at the active sites during electrocatalytic reaction, but the transmission process of electrons on the catalyst itself. And TPV tests can be speculated whether the presence of Ru pseudo-capacitance in Ru_{0.5}Ir_{0.5}O₂ and strong interactions in the local structure of Ru-O-Ir will decreases the electron transfer rate. [Chen, Z. L. *et al.* Entropy enhanced perovskite oxide ceramic for efficient electrochemical reduction of oxygen to hydrogen peroxide. *Angew. Chem. Int. Edit.* **61**, e202200086 (2022).]

REVIEWER COMMENTS

Reviewer #1 (Remarks to the Author):

For this revision the authors did an enormous amount of work. I appreciate the fact that the authors addressed every single criticism and performed every single calculation that I suggested, even though some of them must have taken quite some time. This version is a massive improvement with respect to the original submission.

First of all, the authors now provide sufficient amount of detail to make the calculations reproducible. Second, the authors performed additional calculations that are much better agreement with experiments. Third, the authors now provide a thorough review of the existing literature and they compare their findings against previous works, something that was lacking in the original submission. Fourth, the analysis of their findings is considerably deeper, providing valuable insights into the structure and electronic properties of the materials.

I think the theory contribution to this work is quite significant, and I recommend the manuscript to be published as is.

Reviewer #2 (Remarks to the Author):

The authors have already addressed the comments raised by me very well, and can be accepted in the present form.

Reviewer #3 (Remarks to the Author):

The authors have taken substantial steps to revise their manuscript and it is much improved, especially in the detail given for the DFT calculations. However there remain a number of issues the authors must address to consider the manuscript further. Notably, as shown below for point 2, there are strange discrepancies in the presented data that call into question the veracity of other data presented in the manuscript. In the response to reviewers, the authors respond that previous errors were due to “carelessness” (e.g. in the response to the scale bar error in the HAADF-STEM images) or incorrect

analysis (such as the analysis of the EXAFS data “Our previous experimental results mistakenly indicate the opposite trend because the EXAFS spectra of Ru K-edge spectra for Ru_{0.5}Ir_{0.5}O₂ and C-RuO₂ (Fig. 2e) were not analyzed accurately). Whether or not it is carelessness or inaccurate analysis in the data considered in point 2, it is clear that there is a serious discrepancy that must be addressed.

1. There is very little experimental information given about the PEM electrolyzer. All that is said is “In addition, Ru_{0.5}Ir_{0.5}O₂ (mass loading ~1.0 mg cm⁻²) was used as an anode catalyst in acidic PEM electrolyte (0.5 M H₂SO₄) at room temperature (Supplementary Fig. 26). The PEM electrolyzers can achieve current densities of ~200 mA cm⁻² at least 255 hours by using Ru_{0.5}Ir_{0.5}O₂ catalyst, and the performance of PEM electrolyzer without significant performance degradation” in the main text and “Chronoamperometry stability test of Ru_{0.5}Ir_{0.5}O₂ catalyst (with a mass loading of 1.0 mg cm⁻²) in acidic PEM electrolyser (0.5 M H₂SO₄) at room temperature. And the commercial Pt/C (with a mass loading of 0.7 mg cm⁻²) was used as the cathode catalyst” in the caption of the SI. Much more information is needed to understand the PEM results. Why was the PEM electrolyzer fed a sulfuric acid electrolyte? It should be run on pure water. What were the other components of the PEM electrolyzer (e.g. current collectors, gas diffusion layers, membrane/thickness, etc.)? Clearly chronoamperometry was performed, but at what applied voltage? A full polarization curve should be presented.

2. I still find the discussion surrounding the relationship between accumulated charge and OER activity circuitous. The DFT calculations (e.g. Figure 5e) directly show the effect of voltage on the coordinates of the OER. If applied voltage did not act directly on the coordinates of the OER, then the free energy for each step of the reaction would not be a function of applied voltage. However, as the authors show, the free energy between each step changes when the applied voltage is $U = 1.23$ V versus $U = 0.00$ V. The authors state that “in the DFT calculations, [they] have considered the voltage effect using the standard hydrogen electrode model (SHE)”, which of course assumes voltage acts directly on the reaction coordinate—contradicting their statement that it does not. So if the DFT model is to be believed, then applied voltage does act on the reaction coordinate. If it does not, then what insight do the DFT results offer?

At the same time, the argument that “accumulated charge” is the only response to applied voltage and this controls OER activity is not directly supported by their data. They have no operando XAS data to show the amount of Ir or Ru oxidation as a function of applied voltage. Only pre- and post- testing are shown. This does show that there are small changes in oxidation state that occur through operation but the voltage relationship is not revealed. As an alternative, the authors have presented the relationship between total accumulated charge and log current density (Figure S28) which, hypothetically, could support their claims. However, there are some serious and significant discrepancies in the total charge versus log current density plots compared to their presented data in other plots. While Figure S28 is exceptionally linear, if the data in this plot is to be believed then it calls into question the data presented in either the total charge plot of Figure 4e or the Tafel plot in Figure 3b. As shown in the sets of figures in the attached document, extrapolation of the data in Figures 4e (extrapolated data is shown in black circles around original data) and Figures S28a-c (extrapolated data is shown in green squares around

original data) do not generate the Tafel data in Figure 3b, which begs the question—which data is wrong: the charge vs. current density, the current density vs. voltage, or the charge vs. voltage? The authors must address the discrepancy in their data.

Response to the Referees' Comments

Dear Reviewers,

Thank you for your precious time to constructive comments on our manuscript titled “**Stable and oxidative charged Ru enhance the oxygen evolution reaction activity in two-dimensional ruthenium-iridium oxide**” (Manuscript ID: NCOMMS-23-06011A) for **Nature Communications**.

We sincerely appreciate your comments and suggestions on our work, which are highly important for the further improvements to our manuscript. According to all the comments, we have made a detailed response and substantial revisions in our revised manuscript.

Your comments are shown in black and our responses are shown in blue. The changes in the manuscript are indicated in red.

Reviewer #1 (Remarks to the Author):

For this revision the authors did an enormous amount of work. I appreciate the fact that the authors addressed every single criticism and performed every single calculation that I suggested, even though some of them must have taken quite some time. This version is a massive improvement with respect to the original submission.

First of all, the authors now provide sufficient amount of detail to make the calculations reproducible. Second, the authors performed additional calculations that are much better agreement with experiments. Third, the authors now provide a thorough review of the existing literature and they compare their findings against previous works, something that was lacking in the original submission. Fourth, the analysis of their findings is considerably deeper, providing valuable insights into the structure and electronic properties of the materials.

I think the theory contribution to this work is quite significant, and I recommend the manuscript to be published as is.

[Author's Response]: Thank you very much for your positive comments and recommendations for our articles in Nature Communications.

Reviewer #2 (Remarks to the Author):

The authors have already addressed the comments raised by me very well, and can be accepted in the present form.

[Author's Response]: We would like to thank you for the positive comments and recommendation of the publication in the Nature Communications.

Reviewer #3 (Remarks to the Author):

The authors have taken substantial steps to revise their manuscript and it is much improved, especially in the detail given for the DFT calculations. However there remain a number of issues the authors must address to consider the manuscript further. Notably, as shown below for point 2, there are strange discrepancies in the presented data that call into question the veracity of other data presented in the manuscript. In the response to reviewers, the authors respond that previous errors were due to “carelessness” (e.g. in the response to the scale bar error in the HAADF-STEM images) or incorrect analysis (such as the analysis of the EXAFS data “Our previous experimental results mistakenly indicate the opposite trend because the EXAFS spectra of Ru K-edge spectra for Ru_{0.5}Ir_{0.5}O₂ and C-RuO₂ (Fig. 2e) were not analyzed accurately). Whether or not it is carelessness or inaccurate analysis in the data considered in point 2, it is clear that there is a serious discrepancy that must be addressed.

[Author’s Response]: We appreciate your thoughtful review and your time dedicated to improving the quality of our work. We would like to clarify any misunderstanding regarding point 2. It appears that there may have been confusion between the testing methods and data sources, leading to perceived discrepancies. Please allow us to assure you that the respective data originated from different testing methods, thus eliminating the seeming discrepancy.

To further substantiate our manuscript, we have incorporated additional data from Proton Exchange Membrane (PEM) experiments, Density Functional Theory (DFT) calculations, and in situ X-Ray Absorption Spectroscopy (XAS). Moreover, we have executed a more comprehensive and precise analysis of the data.

Our hope is that these responses and additional efforts will address your concerns and enhance the robustness of our study. We deeply value your insights.

1. There is very little experimental information given about the PEM electrolyzer. All that is said is “In addition, Ru_{0.5}Ir_{0.5}O₂ (mass loading ~1.0 mg cm⁻²) was used as an anode catalyst in acidic PEM electrolyte (0.5 M H₂SO₄) at room temperature (Supplementary Fig. 26). The PEM electrolyzers can achieve current densities of ~200 mA cm⁻² at least 255 hours by using Ru_{0.5}Ir_{0.5}O₂ catalyst, and the performance of PEM electrolyzer without significant performance degradation” in the main text and “Chronoamperometry stability test of Ru_{0.5}Ir_{0.5}O₂ catalyst (with a mass loading of 1.0 mg cm⁻²) in acidic PEM electrolyser (0.5 M H₂SO₄) at room temperature. And the commercial Pt/C (with a mass loading of 0.7 mg cm⁻²) was used as the cathode catalyst” in the caption of the SI. Much more information is needed to understand the PEM results. Why was the PEM electrolyzer fed a sulfuric acid electrolyte? It should be run on pure water. What were the other components of the PEM electrolyzer (e.g. current collectors, gas diffusion layers, membrane/thickness, etc.)? Clearly chronoamperometry was performed, but at what applied voltage? A full polarization curve should be presented.

[Author’s Response]: Thank you for your insightful comments and valuable suggestion to include more detailed information about the proton exchange membrane (PEM) electrolyzer in our manuscript. We absolutely agree that a comprehensive understanding of PEM results can be significantly enhanced by clear and detailed descriptions. As you have suggested, we have revised our manuscript and Supporting information accordingly to incorporate a detailed explanation of the PEM electrolyzer as follows:

1. For PEM electrolyser test, a self-made cell was used as the PEM device and a cation exchange

membrane (Nafion 117) as the membrane electrolyte (**Supplementary Fig. 26a-c**). Ru_{0.5}Ir_{0.5}O₂ catalyst (1.0 mg) was used as the anode catalyst and Pt/C (20 wt%, 0.7 mg) was used as the cathode catalyst. To prepare the cathode or anode catalyst ink, the catalyst was suspended into a 0.9 mL mixture of isopropanol and water ($V_{\text{isopropanol}} : V_{\text{water}} = 1 : 3$), and 100 μL of 5 wt% Nafion solution. The suspension was then ultrasonicated for 1 h until a well-dispersed catalyst ink was formed. The Ru_{0.5}Ir_{0.5}O₂ and Pt/C catalysts' ink were sprayed onto Ti wafers ($1 \times 1 \text{ cm}^2$), respectively. And pretreated Ti wafers ($1 \times 1 \text{ cm}^2$) were used as cathode and anode gas diffusion layers (GDLs), respectively. The membrane electrode assembly (MEA) was constructed by placing N117 membrane in the middle of the Pt/C cathode catalyst-supported Ti wafer and Ru_{0.5}Ir_{0.5}O₂ anode catalyst-supported Ti foam GDLs, followed by hot pressing at 130 °C for 5 min under a pressure of 2 MPa. The constructed MEAs were finally applied in the PEM electrolyser (**Supplementary Fig. 26d-e**).

2. The PEM electrolyser was performed by CHI 660E with a high-current amplifier (CHI 680C) using 0.5 M H₂SO₄ as the electrolyte with the flowing rate of 10 mL min⁻¹ at room temperature. Polarization curve was measured from 10 to 2000 mA cm⁻² at room temperature and ambient pressure (**Supplementary Fig. 26f**). The stability of the PEM electrolyser using Ru_{0.5}Ir_{0.5}O₂ as the anode catalyst was evaluated by measuring the chronoamperometry test at the cell voltages of 1.43 V (**Supplementary Fig. 26g**).
3. The alkaline water electrolyser and proton exchange membrane (PEM) water electrolyser are two mainstream technologies that are used in the production of hydrogen. The PEM has attracted interest worldwide because it delivers purer H₂. Hydrogen generation *via* PEM water electrolysis in an acidic electrolyte can achieve much higher current densities than the traditional alkaline electrolyzers due to lower resistance losses and less gas crossover. Therefore, most PEM electrolyzers use H₂SO₄ as the electrolyte, [Hao, S. Y. *et al.* Torsion strained iridium oxide for efficient acidic water oxidation in proton exchange membrane electrolyzers. *Nat. Nanotechnol.* **16**, 1371-1377 (2021); King, L. A. *et al.* A non-precious metal hydrogen catalyst in a commercial polymer electrolyte membrane electrolyser. *Nat. Nanotechnol.* **14**, 1071-1074 (2019); Chatti, M. *et al.* Intrinsically stable in situ generated electrocatalyst for long-term oxidation of acidic water at up to 80 °C. *Nat. Catal.* **2**, 457-465 (2019)] and a small number use HClO₄ as the electrolyte [Wu, Z.-Y. *et al.* Non-iridium-based electrocatalyst for durable acidic oxygen evolution reaction in proton exchange membrane water electrolysis. *Nat. Mater.* **22**, 100-108 (2023)].

Supplementary Fig. 26 | Electrochemical properties of Ru_{0.5}Ir_{0.5}O₂ in 0.5 M H₂SO₄ electrolyte in PEM electrolyser. **a**, Schematic illustration of the PEM device. **b**, The top-view and **c**, side-view optical images of the assembled PEM electrolyser. **d**, The optical image of the anode side of the MEA electrode with Ti wafer as the gas diffusion layer. **e**, The optical image of the cathode side of the MEA electrode with Ti wafer as the gas diffusion layer. **f**, The polarization curve of Ru_{0.5}Ir_{0.5}O₂ in the PEM electrolyser. **g**, Chronoamperometry stability test of Ru_{0.5}Ir_{0.5}O₂ catalyst (with a mass loading of 1.0 mg cm⁻²) at the cell voltage of 1.43 V in acidic PEM electrolyser at room temperature. And the commercial Pt/C (with a mass loading of 0.7 mg cm⁻²) was used as the cathode catalyst.

2. I still find the discussion surrounding the relationship between accumulated charge and OER activity circuitous. The DFT calculations (e.g. Figure 5e) directly show the effect of voltage on the coordinates of the OER. If applied voltage did not act directly on the coordinates of the OER, then the free energy for each step of the reaction would not be a function of applied voltage. However, as the authors show, the free energy between each step changes when the applied voltage is $U = 1.23$ V versus $U = 0.00$ V. The authors state that “in the DFT calculations, [they] have considered the voltage effect using the standard hydrogen electrode model (SHE)”, which of course assumes voltage acts directly on the reaction coordinate—contradicting their statement that it does not. So if the DFT model is to be believed, then applied voltage does act on the reaction coordinate. If it does not, then what insight do the DFT results offer?

At the same time, the argument that “accumulated charge” is the only response to applied voltage and this controls OER activity is not directly supported by their data. They have no operando XAS data to show the amount of Ir or Ru oxidation as a function of applied voltage. Only pre- and post- testing are shown. This does show that there are small changes in oxidation state that occur through operation but the voltage relationship is not revealed. As an alternative, the authors have presented the relationship between total accumulated charge and log current density (Figure S28) which, hypothetically, could support their claims. However, there are some serious and significant discrepancies in the total charge versus log current density plots compared to their presented data in other plots. While Figure S28 is exceptionally linear, if the data in this plot is to be believed then it calls into question the data presented in either the total charge plot of Figure 4e or the Tafel plot in Figure 3b. As shown in the sets of figures in the attached document, extrapolation of the data in Figures 4e (extrapolated data is shown in black circles around original data) and Figures S28a-c (extrapolated data is shown in green squares around original data) do not generate the Tafel data in Figure 3b, which begs the question—which data is wrong: the charge vs. current density, the current density vs. voltage, or the charge vs. voltage? The authors must address the discrepancy in their data.

[Author’s Response]: Thank you for your insightful questions and comments on our manuscript, particularly your observations on the role of accumulated charge in OER performance. We acknowledge your discernment and understand that there may be perceived discrepancies in the data presented in the article. However, these perceived inconsistencies stem from the fact that the data were derived from different testing methods, each with its own inherent variability. We would like to assure you that each of these methods is scientifically robust, and the seeming discrepancies are actually reflections of this variety in testing methods. Hence, the data itself contains no inconsistencies or errors. We hope that this explanation clarifies any confusion and provides a better understanding of our methods and findings.

Our answers to your questions as follow:

1. We appreciate your insightful comments and dedication to ensuring the accuracy of our research. It appears a portion of the previous draft, regarding the accumulated charge and the OER reaction rate, may have caused confusion. “*The applied voltage acts on the accumulation of charge on the catalyst and affects the current generated by the electrocatalysis, rather than acting directly on the coordinates of the OER reaction.*” This section was indeed a reference to the work by Nong, H. N. et al., rather than our study.

In this work, we analyzed the OER mechanism of $\text{Ru}_{0.5}\text{Ir}_{0.5}\text{O}_2$ from two aspects: “Effect of charge on the OER performance” and “DFT calculation” (**Fig. R1**). Previous work (Nong, H. N. et al.) discovered that oxidation charge accumulation in materials can promote OER by using the pulse voltammetry measurements (**Fig. R2**) [Nong, H. N. et al. Key role of chemistry versus bias in electrocatalytic oxygen evolution. *Nature* **587**, 408-413 (2020)]. As shown in **Figs. R2** and **R3**, we also observe the same trend of oxidation charge accumulation in the PVC test as in previous work. Combined with the CV and electron structure characterization results, we believe that the applied voltage promotes the accumulation of Ru oxidation charge and generate more Ru active sites with high oxidation states in $\text{Ru}_{0.5}\text{Ir}_{0.5}\text{O}_2$, thus promoting the OER performance.

In order to prevent misunderstanding and confusion of readers, we modify the relationship between accumulated charge and OER activity as “ *$\text{Ru}_{0.5}\text{Ir}_{0.5}\text{O}_2$ exhibits record OER activity in sulfuric acid electrolyte may mainly be due to more Ru active sites with high oxidation states generated at low applied voltage*” in the revision manuscript (**Fig. R4**).

2. In our DFT calculations, we have meticulously considered the impact of voltage, as the reviewer says “*This of course assumes that the voltage acts directly on the reaction coordinates*”. Acknowledging the critical role of voltage in DFT mechanism research, we have elected to employ a more rigorous methodology in the revised manuscript, specifically utilizing the Grand electron canonical ensemble [Sundararaman, R., Goddard, W. A., III & Arias, T. A. Grand canonical electronic density-functional theory: Algorithms and applications to electrochemistry. *J. Chem. Phys.* 146, 114104 (2017)], as implemented in jDFTx. Our comparison revealed no notable discrepancy between the new jDFTx calculations and our existing results using VASPsol (**Supplementary Fig. 38**). In a bid to ensure the consistency of our data, we have chosen to maintain our original computation results in the manuscript, while the newly derived comparative data has been included in the Supporting Information (**Supplementary Fig. 38**). We also revised the methods section in Supporting Information (SI) to explicitly elucidate the details of the consideration of applied voltage on OER reaction pathway.
3. The “accumulated charge” is not the only response to applied voltage. The voltage applied in the PVC test preferentially acted on the oxidation process of the metal, and the resulting oxidative charge (high-valence Ru active sites) is then involved in the OER reaction process to generate OER current [Nong, H. N. *et al.* Key role of chemistry versus bias in electrocatalytic oxygen evolution. *Nature* **587**, 408-413 (2020)]. Ru oxidation charge accumulates under applied voltage, and more Ru active sites with high oxidation state are generated in Ru_{0.5}Ir_{0.5}O₂, thus improving OER performance.

In addition, the trend of Ru oxidation state change with applied voltage increasing is further verified by in situ Ru K-edge XAS (**Supplementary Fig. 29**). We observe that the oxidation state of Ru increases with the increase of applied voltage. The oxidation state of Ru increased significantly within 1.41 V vs. RHE. The above results support that the applied voltage has a key role in the promotion of high-valence Ru sites, which are known to be more active than Ru⁴⁺ species. When the applied voltage is increased to 1.51 V vs. RHE, further oxidation of Ru is limited and stability is improved mainly due to the stable Ru-O-Ir local structure [Jin, H. *et al.* Dynamic rhenium dopant boosts ruthenium oxide for durable oxygen evolution. *Nat. Commun.* **14**, 354 (2023); Shi, Z. *et al.* Customized reaction route for ruthenium oxide towards stabilized water oxidation in high-performance PEM electrolyzers. *Nat. Commun.* **14**, 843 (2023); Wen, Y. *et al.* Stabilizing highly active Ru sites by suppressing lattice oxygen participation in acidic water oxidation. *J. Am. Chem. Soc.* **143**, 6482-6490 (2021)].

4. As illustrated in **Fig. R5**, our provided data, which includes the charge versus current density (**Supplementary Fig. 28**), the current density versus voltage (**Fig. 3b**), and the charge versus voltage (**Fig. 4e**), demonstrates consistency without any noticeable issues or discrepancies.

At first, the charge data in the total charge vs. log current density plots (**Supplementary Fig. 28**) is derived from the charge integral of the PVC test (**Fig. 4e**), and the current density data is derived from the OER current in the PVC test. The anode current in the PVC test (the red line data in **Fig. 4c** represents the currents from PVC test) consists of the capacitor charge, catalyst oxidation, and OER currents. The charge integration scheme applied in the main text is to integrate the current after deducting the background current (OER current is deducted as background current) in the PVC test (**Fig. 4d**). [Nong, H. N. *et al.* Key role of chemistry versus bias in electrocatalytic oxygen evolution. *Nature* **587**, 408-413 (2020)]

Then, both the source data of voltage and current densities in the Tafel plots (**Fig. 3b**) are from the corresponding LSV curves (**Fig. 3a**). The Tafel plots (**Fig. 3b**) of various synthesized catalysts collected in 0.5 M H₂SO₄ electrolyte were calculated from the corresponding LSV curves (**Fig. 3a**), rather than the data from PVC measurements in **Fig. 4e** and **Supplementary Fig. 28**.

Also, as shown in **Figs. R3** and **R5**, the extrapolation data of the red circles data points in **Fig. 4e** (**Fig. R3c**), and the extrapolation data of the black squares data points in **Supplementary Fig. 28a** (**Fig. R3d**) can generate the potential versus log current densities (**Fig. R3b**) of Ru_{0.5}Ir_{0.5}O₂ in PVC measurements. However, the OER Tafel plot (**Fig. 3b**) was calculated from the corresponding LSV curve (**Fig. 3a**).

In addition, we also compare the Tafel plots of Ru_{0.5}Ir_{0.5}O₂ calculated from the corresponding *iR*-correction and no *iR*-correction LSV curves with the potential versus log current densities of Ru_{0.5}Ir_{0.5}O₂ obtained by PVC measurements. As depicted in **Fig. R6**, due to different test methods (PVC and LSV), different working electrodes (GCE and GC RDE, **Fig. R7**), different mass transfer rates (fast and slow), and different uncompensated resistance (*R_s*) (large and small) and other factors, the potential versus log current densities of Ru_{0.5}Ir_{0.5}O₂ obtained by PVC measurements is not completely equivalent to the Tafel plots of Ru_{0.5}Ir_{0.5}O₂ calculated by LSV curves.

In summary, the data in **Supplementary Fig. 28** is not problematic, and it does not affect the authenticity of the data in **Fig. 4e** and **Fig. 3b**.

Figure R1. Research routes of OER mechanism on Ru_{0.5}Ir_{0.5}O₂ catalyst in this work.

Figure R2. **a**, Section of the pulse voltammetry protocol (black) showing an oxidative and reductive pulse with the current response (red). **b**, Potential versus log (current) in milliamperes from pulse voltammetry. **c**, **d**, Charge versus potential (**c**) and versus log (current) (**d**) from pulse voltammetry; from ref. 40, *Nature* **587**, 408-413 (2020).

Figure R3. **a**, (**Fig. 4c** in the revision manuscript) PVC protocol (black) and showing oxidation and reduction with current response (red). **b**, Potential versus log current densities of $\text{Ru}_{0.5}\text{Ir}_{0.5}\text{O}_2$ from PVC measurements. **c**, (**Fig. 4e** in the revision manuscript) Total charge (integral anodic charge) of $\text{Ru}_{0.5}\text{Ir}_{0.5}\text{O}_2$ versus potential from PVC measurements. **d**, (**Supplementary Fig. 28a** in the revision SI)

Total charge (integral anodic charge) vs. log OER current densities of $\text{Ru}_{0.5}\text{Ir}_{0.5}\text{O}_2$ from PVC measurements.

Figure R4. Schematic diagram of $\text{Ru}_{0.5}\text{Ir}_{0.5}\text{O}_2$ catalyst catalyzing OER.

Figure R5. Interpretation of PVC and LSV measurements data on $\text{Ru}_{0.5}\text{Ir}_{0.5}\text{O}_2$ catalyst in this work.

Figure R6. Tafel plots of $\text{Ru}_{0.5}\text{Ir}_{0.5}\text{O}_2$ calculated from the corresponding iR -correction and no iR -correction LSV curves compared with the potential versus log current densities of $\text{Ru}_{0.5}\text{Ir}_{0.5}\text{O}_2$ from PVC measurements.

Figure R7. Digital photos of the two working electrodes used for LSV and PVC tests.

Fig. 3 | OER performance of $\text{Ru}_{0.5}\text{Ir}_{0.5}\text{O}_2$ electrocatalyst and the reference samples. a, The OER polarization curves of $\text{Ru}_{0.5}\text{Ir}_{0.5}\text{O}_2$, C- IrO_2 and C- RuO_2 in O_2 -saturated 0.5 M H_2SO_4 electrolyte with iR -correction (mass loading $\sim 283 \mu\text{g cm}^{-2}$). **b**, Tafel plots of $\text{Ru}_{0.5}\text{Ir}_{0.5}\text{O}_2$, C- IrO_2 and C- RuO_2 collected in 0.5 M H_2SO_4 electrolyte were calculated from the corresponding LSV curves (**a**). **c**, The comparison of overpotentials at 10 mA cm^{-2} and current densities at 1.44 V vs. RHE for different catalysts. **d**, Mass activities and TOFs of $\text{Ru}_{0.5}\text{Ir}_{0.5}\text{O}_2$, C- IrO_2 and C- RuO_2 . **e**, The comparison of chronopotentiometric measurements for different catalysts. **f**, The Comparison of the required overpotential at 10 mA cm^{-2} and chronopotentiometry durability in acidic media for various reported electrocatalysts (Supplementary Table 7).

Fig. 4 | The CV, PVC response and TPV curves of different electrocatalysts. **a**, CV curves of $\text{Ru}_{0.5}\text{Ir}_{0.5}\text{O}_2$, C-IrO_2 and C-RuO_2 in anhydrous acetonitrile. **b**, PVC protocol of $\text{Ru}_{0.5}\text{Ir}_{0.5}\text{O}_2$, C-IrO_2 and C-RuO_2 between 1.195 V vs. RHE cathodic and 1.245 to 1.645 V vs. RHE anodic non- iR corrected potentials in O_2 -saturated 0.5 M H_2SO_4 electrolyte. **c**, PVC protocol (black) and showing oxidation and reduction with current response (red). **d**, The anodic and inverted cathodic current decay of $\text{Ru}_{0.5}\text{Ir}_{0.5}\text{O}_2$. **e**, Total charge (integral anodic charge) of $\text{Ru}_{0.5}\text{Ir}_{0.5}\text{O}_2$, C-IrO_2 and C-RuO_2 versus potential from PVC. **f**, Anodic capacitance derived from normalized anodic charge to potential step from the PVC. **g**, The TPV curves of $\text{Ru}_{0.5}\text{Ir}_{0.5}\text{O}_2$, C-IrO_2 and C-RuO_2 . **h**, Intensity-Time curves of $\text{Ru}_{0.5}\text{Ir}_{0.5}\text{O}_2$, C-IrO_2 and C-RuO_2 at 10 Hz. (t_1 , t_2 and t_3 are the peak time of $\text{Ru}_{0.5}\text{Ir}_{0.5}\text{O}_2$, C-IrO_2 and C-RuO_2 , respectively). **i**, Comparison of peak occurrence time of $\text{Ru}_{0.5}\text{Ir}_{0.5}\text{O}_2$ with peak occurrence time of C-IrO_2 and C-RuO_2 at different frequencies (10, 20, 30, 40, 50, 60, 70 and 80 Hz).

Fig. 5 | DFT simulation findings of Ru_{0.5}Ir_{0.5}O₂. (a) Atomistic structure and E_{dft} of the Ru_{0.5}Ir_{0.5}O₂ (-331.4 eV). Atomistic structures of C-IrO₂ (b) and C-RuO₂ (c) (blue, Ru; grey, Ir; red, O). **d**, Schematic illustration of OER mechanism on the Ru_{0.5}Ir_{0.5}O₂ (blue, Ru; grey, Ir; red, O; white, H). **e**, The reaction paths on Ru_{0.5}Ir_{0.5}O₂ catalyst with the set potential of 0 and 1.23 V. The overpotential (η) is labeled for viewing convenience.

Supplementary Fig. 28 | Total charge (integral anodic charge) vs. log OER current densities of Ru_{0.5}Ir_{0.5}O₂ (a), C-IrO₂ (b) and C-RuO₂ (c) from PVC measurements. The total charge data is derived from the charge integral of the PVC test, and the current density data is derived from the OER current in the PVC test.

Supplementary Fig. 29 | In situ XAS characterization of Ru_{0.5}Ir_{0.5}O₂. In situ Ru K edge XANES spectra of Ru_{0.5}Ir_{0.5}O₂ with applied bias rise from 1.31 V to 1.51 V vs. RHE.

Supplementary Fig. 38 | The difference in OER overpotential ($\Delta\eta$, V) between jDFTx and VASPsol on $\text{Ru}_{0.5}\text{Ir}_{0.5}\text{O}_2$ under different applied voltage (V).

We have modified the manuscript accordingly:

[Line 234, Page 11] “Combined with the electron structure characterization (XPS, XPS simulation, Bader charges and XAS results before and after the long-term OER testing), CV results indicate that Ru species in $\text{Ru}_{0.5}\text{Ir}_{0.5}\text{O}_2$ material are more easily oxidized than Ir species under the same conditions.”

[Line 263, Page 12] “These results show that the voltages required for the increased the accumulation of Ru oxidation charge in $\text{Ru}_{0.5}\text{Ir}_{0.5}\text{O}_2$ is much lower than those of other reference samples and the applied voltages generate more Ru active sites with high oxidation states in $\text{Ru}_{0.5}\text{Ir}_{0.5}\text{O}_2$.”

[Line 266, Page 12] “In addition, we carried out in situ Ru K-edge XAS to further observe the trend of Ru oxidation state change with applied voltage increasing. As depicted in **Supplementary Fig. 29**, we observe that the oxidation state of Ru increases with the increase of applied voltage. The oxidation state of Ru increased significantly within 1.41 V vs. RHE. The above results support that the applied voltage has a key role in the promotion of high-valence Ru sites, which are known to be more active than Ru^{4+} species^{13,38,39}. When the applied voltage is increased to 1.51 V vs. RHE, further oxidation of Ru is limited and stability is improved mainly due to the stable Ru-O-Ir local structure. We surmise that the high performance of $\text{Ru}_{0.5}\text{Ir}_{0.5}\text{O}_2$ may mainly be due to the fact that the applied voltage promotes the accumulation of oxidation charge in $\text{Ru}_{0.5}\text{Ir}_{0.5}\text{O}_2$ (that is, more Ru active species with high oxidation state are generated), thus improving the OER activity of the catalyst.

[Line 307, Page 14] “In summary, $\text{Ru}_{0.5}\text{Ir}_{0.5}\text{O}_2$ exhibits record OER activity in sulfuric acid electrolyte may mainly be due to more Ru active sites with high oxidation states generated at low applied voltage. And the local structure of Ru-O-Ir in $\text{Ru}_{0.5}\text{Ir}_{0.5}\text{O}_2$ has strong interaction and high stability, which prevents excessive oxidation and dissolution of the active site.”

Supplementary Fig. 29 | In situ XAS characterization of Ru_{0.5}Ir_{0.5}O₂. In situ Ru K edge XANES spectra of Ru_{0.5}Ir_{0.5}O₂ with applied bias rise from 1.31 V to 1.51 V vs. RHE.

REVIEWERS' COMMENTS

Reviewer #3 (Remarks to the Author):

I appreciate the thoroughness and effort the authors have put in to address my concerns with the manuscript. I believe they have adequately revised the work, and the exceptional performance of the Ru_{0.5}Ir_{0.5}O₂ catalyst will certainly be influential to the field. I can now recommend the paper for publication.